# FAST AND INTERPRETABLE PROTEIN SUBSTRUCTURE ALIGNMENT VIA OPTIMAL TRANSPORT

**Zhiyu Wang**[1,2] *  **Bingxin Zhou**[1] * †  **Jing Wang**[1]  **Yang Tan**[1]  **Weishu Zhao**[1]
**Pietro Liò**[2]  **Liang Hong**[1]†
[1] Shanghai Jiao Tong University.  [2] University of Cambridge.

## ABSTRACT

Proteins are essential biological macromolecules that execute life functions. Local structural motifs, such as active sites, are the most critical components for linking structure to function and are key to understanding protein evolution and enabling protein engineering. Existing computational methods struggle to identify and compare these local structures, which leaves a significant gap in understanding protein structures and harnessing their functions. This study presents PLASMA, a deep-learning-based framework for efficient and interpretable residue-level local structural alignment. We reformulate the problem as a regularized optimal transport task and leverage differentiable Sinkhorn iterations. For a pair of input protein structures, PLASMA outputs a clear alignment matrix with an interpretable overall similarity score. Through extensive quantitative evaluations and three biological case studies, we demonstrate that PLASMA achieves accurate, lightweight, and interpretable residue-level alignment. Additionally, we introduce PLASMA-PF, a training-free variant that provides a practical alternative when training data are unavailable. Our method addresses a critical gap in protein structure analysis tools and offers new opportunities for functional annotation, evolutionary studies, and structure-based drug design. Reproducibility is ensured via our official implementation at https://github.com/ZW471/PLASMA-Protein-Local-Alignment.git.

## 1 INTRODUCTION

Proteins are essential macromolecules responsible for life functions, from catalysis and signal transduction to structural support and transport. Local structural motifs (*e.g.*, catalytic residues, binding pockets, metal-binding sites) are critical for understanding mechanisms, designing therapeutics, and guiding protein engineering (Mills et al., 2018). Structural conservation is three to ten times stronger than sequence conservation across evolution, suggesting that local structural comparison can reveal functional relationships invisible to sequence-based methods (Hvidsten et al., 2009).

Despite their importance, existing computational methods primarily emphasize global structure comparison or sequence alignment. The inability to detect local structural motifs, *i.e.*, compact three-dimensional residue arrangements that often concentrate around catalytic pockets or interaction sites, prevents researchers from understanding protein evolution, predicting functions of uncharacterized proteins, and rationally designing proteins with desired properties. While large-scale resources like AFDB (Jumper et al., 2021; Varadi et al., 2022) open a unique opportunity to uncover conserved motifs across the protein universe, active sites often comprise spatially proximate residues that may be widely separated in sequence or embedded within different overall fold architectures (Liu et al., 2018). Addressing this gap is key to advancing our understanding of protein function and evolution.

The development of robust local structure alignment methods specifically targeting local structural motifs is not merely a technical challenge but a fundamental requirement for advancing multiple areas of biological research and application. Existing methods for protein substructure alignment

---

*Equal contribution first authors.
†Corresponding authors (bingxin.zhou@sjtu.edu.cn; hongl3liang@sjtu.edu.cn).

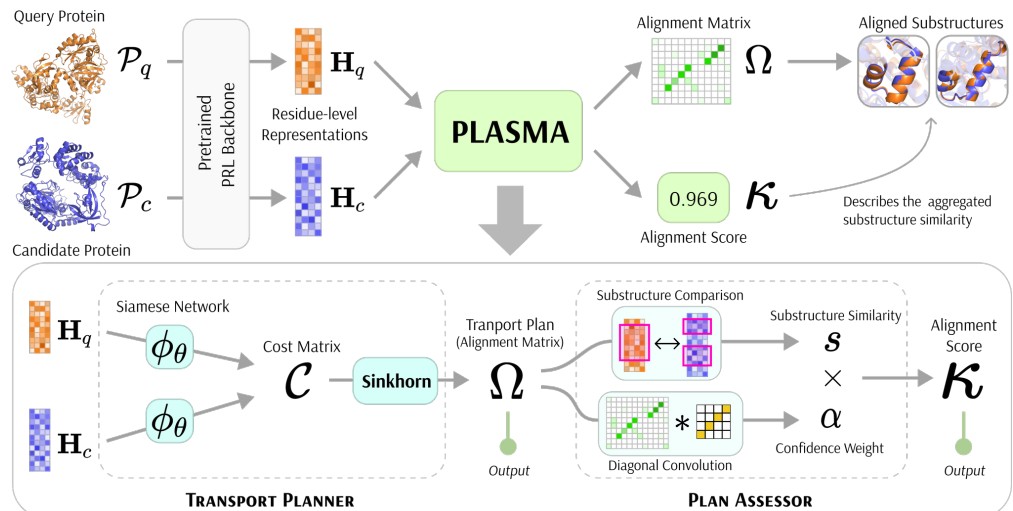

Figure 1: PLASMA Overview. PLASMA converts residue-level protein embeddings into substructure alignments using optimal transport. A *Transport Planner* learns cost matrices with Sinkhorn iterations, and a *Plan Assessor* produces similarity scores. The framework provides alignment matrices and quantitative scores without requiring model-specific designs.

can be broadly divided into three categories. The first relies on template-based searches, where predefined motifs are used to identify similar substructures (Bittrich et al., 2020; Kim et al., 2025). These approaches are effective for detecting well-characterized patterns but cannot uncover novel similarities, making them **unsuitable for pairing novel structural motifs**. The second category estimates substructure similarity based on the global similarity of entire protein structures. Several studies leverage structural superposition (Zhang, 2005) or structural tokenization (Holm, 2020) to produce residue-level matches with sequence alignment, but they are **computationally demanding and difficult to scale to large datasets**. More recent embedding-based methods (Hamamsy et al., 2024) are enabled by advances in protein representation learning, which make alignment faster and competitive for whole-protein comparison. However, they compress residue-level information into coarse embeddings, which causes **problems in producing interpretable local alignments**. The third category directly addresses substructure alignment by constructing pairwise similarity matrices and using dynamic programming to find matching regions. This approach captures local similarities more accurately than global methods and produces scores that reflect substructure correspondence (Kaminski et al., 2023; Liu et al., 2024; Pantolini et al., 2024). However, the results can be influenced by overall structural patterns, and **alignment matrices have limited interpretability** since they are optimized for algorithmic performance rather than clarity. Additionally, these methods are typically untrainable and cannot adapt to specific alignment tasks or incorporate domain knowledge, limiting their ability to improve through experience or be customized for particular biological contexts.

The challenges above point to the need for a novel protein substructure alignment method that combines accuracy, efficiency, and clarity. To this end, we explore optimal transport (OT), a mathematical framework proven effective in alignment problems (Mena et al., 2018). In particular, the differentiable Sinkhorn algorithm (Sinkhorn & Knopp, 1967; Cuturi, 2013) has shown strong ability to uncover meaningful correspondences in 3D shape analysis (Eisenberger et al., 2020) and subgraph matching (Ramachandran et al., 2024). Notably, these OT-based alignment methods assume strict one-to-one correspondences between all residues or that one set of residues is fully contained within the other. These constraints do not hold for protein substructure alignment, as functionally similar regions may only partially overlap and vary in length across proteins.

To address the aforementioned limitations, we reframe protein substructure alignment as an OT problem and introduce PLASMA (**P**luggable **L**ocal **A**lignment via **S**inkhorn **MA**trix). As illustrated in Figure 1, PLASMA operates on residue-level embeddings from a pre-trained protein representation model and identifies the residue-level alignment between protein pairs. The *Transport Planner* computes the pairwise matching using a learnable cost matrix and differentiable Sinkhorn iterations (Section 3), and the *Plan Assessor* then summarizes the resulting alignment matrix into a single similarity score reflecting the overall similarity of the matched substructures (Section 4). PLASMA

functions as a lightweight, plug-and-play module for protein representation models. It is capable of efficiently aligning partial and variable-length matches between local structural regions.

Our work addresses these limitations through three contributions. First, we introduce a formulation of residue-level local structural alignment based on regularized optimal transport with a learnable geometric cost, which provides a principled and flexible way to define correspondence and enables efficient, fully parallel implementation. Second, this formulation enables clear and interpretable residue–residue correspondences and naturally supports partial, variable-length, and non-sequential motif alignments, resolving the difficulty of obtaining reliable local alignments. Third, PLASMA produces a normalized and interpretable similarity score through its OT-based objective, overcoming the limitations of existing approaches whose alignment matrices or similarity measures lack a consistent probabilistic meaning. Our experiments show strong generalization to low-homology structures, and the case studies demonstrate the biological interpretability and practical utility of the resulting alignments.

## 2 PROTEIN SUBSTRUCTURE ALIGNMENT VIA OPTIMAL TRANSPORT

**Problem Formulation**   Consider a query protein $\mathcal{P}_q = \{r_{q,1}, \ldots, r_{q,N}\}$ of $N$ residues and a candidate protein $\mathcal{P}_c = \{r_{c,1}, \ldots, r_{c,M}\}$ of $M$ residues. Suppose the two proteins contain local structural motifs $\mathcal{F}_q = \{f_{q,1}, \ldots, f_{q,n}\} \subseteq \mathcal{P}_q$ and $\mathcal{F}_c = \{f_{c,1}, \ldots, f_{c,m}\} \subseteq \mathcal{P}_c$, where $n \leq N$ and $m \leq M$. The objective of protein substructure alignment is: (1) to identify the corresponding fragments $\mathcal{F}_q$ and $\mathcal{F}_c$ within $\mathcal{P}_q$ and $\mathcal{P}_c$, and (2) to score their level of similarity.

The task is challenging for several reasons: the overall structures of $\mathcal{P}_q$ and $\mathcal{P}_c$ may differ substantially, the fragments $\mathcal{F}_q$ and $\mathcal{F}_c$ may vary in sequence length or composition, and alignments require remaining meaningful in a biological context. In particular, biologically relevant alignments should capture functional similarities, such as common enzymatic activities or conserved structural roles.

**Optimal Transport Reformulation**   To address the protein substructure alignment problem, we reformulate it as an entropy-regularized OT problem between the residues of two proteins $\mathcal{P}_q$ and $\mathcal{P}_c$. Each protein is represented as a set of residue embeddings that capture local biochemical and structural context. The OT solver then computes a soft alignment matrix $\Omega \in \mathbb{R}^{N \times M}$ by assigning weights between residues so as to minimize the overall transport cost $\mathcal{C}$. This formulation bypasses explicit fragment enumeration, naturally accommodates partial and variable-length matches, and produces interpretable alignment matrices that highlight the underlying substructures (Appendix A).

**Overview of PLASMA**   We implement entropy-regularized OT and propose **PLASMA**, a module that transforms $\boldsymbol{H}_q \in \mathbb{R}^{N \times d}$ and $\boldsymbol{H}_c \in \mathbb{R}^{M \times d}$, residue-level $d$-dimensional hidden representations of $\mathcal{P}_q$ and $\mathcal{P}_c$ (*e.g.*, from pre-trained protein language models), into a soft alignment matrix $\Omega \in \mathbb{R}^{N \times M}$ and a similarity score $\kappa \in [0, 1]$. In our experiments, we instantiate $H_q$ and $H_c$ with seven diverse protein representation backbones (Section 6), and observe consistent alignment behavior across them, indicating that PLASMA is not tied to a particular choice of encoder. Formally,

$$(\Omega, \kappa) = \text{PLASMA}(\boldsymbol{H}_q, \boldsymbol{H}_c). \tag{1}$$

PLASMA consists of two complementary components (visualized in Figure 1, with details introduced in the next two sections). The first component, the *Transport Planner*, produces $\Omega$ to highlight local correspondences between $\mathcal{P}_q$ and $\mathcal{P}_c$. The second component, the *Plan Assessor*, summarizes this alignment matrix into a similarity score $\kappa \in [0, 1]$, providing a quantitative measure of alignment quality. The framework achieves a computational complexity of $O(N^2)$ (Appendix B).

## 3 TRANSPORT PLANNER

The Transport Planner module handles the core OT computation. It defines cost functions between residue pairs and solves the regularized OT problem to produce an $\Omega$ that captures residue-level matching between query and candidate proteins $(\mathcal{P}_q, \mathcal{P}_c)$.

**Cost Matrix**   We formulate a learnable cost matrix with a siamese network architecture to capture complex residue-level similarities. This approach enables PLASMA to learn task-specific represen-

tations that optimize alignment quality through end-to-end training. The cost from $r_{q,i}$ to $r_{c,j}$ is denoted by $\mathcal{C}_{ij}$ in the learnable cost matrix, defined as

$$\mathcal{C}_{ij} = \left\| \left[ \phi_\theta(\mathrm{LN}(\boldsymbol{h}_{q,i})) - \phi_\theta(\mathrm{LN}(\boldsymbol{h}_{c,j})) \right]_+ \right\|_1. \tag{2}$$

Here $\boldsymbol{h}_{q,i}$ and $\boldsymbol{h}_{c,j}$ denote the hidden representations of residues $r_{q,i}$ and $r_{c,j}$, respectively. The operator $[\cdot]_+$ applies a hinge non-linearity, shown to outperform dot-product similarity in subgraph matching tasks (Raj et al., 2025). The layer normalization $\mathrm{LN}(\cdot)$ facilitates robust optimization dynamics with numerical stability and scale-invariant representations. The siamese network $\phi_\theta(\cdot)$ processes query and candidate residues using a twin architecture with shared parameters $\theta$.

**Learnable and Parameter-Free Implementations**  The siamese network architecture can be chosen flexibly, ranging from Transformer-based (Hamamsy et al., 2024) models to graph neural networks (Jamasb et al., 2024), depending on the inductive bias of the input data and the computational budget. Here we also provide a simple implementation using fully connected layers:

$$\phi_\theta(\boldsymbol{h}) = \mathrm{ReLU}(\boldsymbol{h} \cdot \boldsymbol{W}_1) \cdot \boldsymbol{W}_2, \tag{3}$$

where $\boldsymbol{W}_1 \in \mathbb{R}^{d \times d'}$ and $\boldsymbol{W}_2 \in \mathbb{R}^{d' \times d'}$ are learnable transformation matrices with $d'$ hidden dimension. For simplicity, we omit the subscript of $\boldsymbol{H}$ as the siamese network applies the same set of parameters to both the query and candidate proteins. This lightweight design serves as an effective default while allowing more sophisticated architectures to be substituted without modifying the overall PLASMA architecture. In addition, for scenarios with a lack of labeled data, we introduce a parameter-free variant, **PLASMA-PF**, which bypasses the siamese network and operates directly on residue embeddings. The cost used in the OT objective follows (2) with no architectural components removed other than the encoder. PLASMA-PF preserves the fundamental alignment functionality and offers a fast baseline for substructure similarity evaluation. Notably, the learnable version remains preferable for improved stability and extrapolation (See Section 6.3 and Figure 4).

**Sinkhorn Alignment Matrix**  Based on the cost matrix $\mathcal{C}$ defined in (2), we formulate the corresponding OT problem (Appendix A) and solve it using the Sinkhorn algorithm (Cuturi, 2013). The algorithm approximates the OT plan by iteratively scaling the matrix to satisfy the marginal constraints with row and column normalizations, ensuring that the total alignment weights of each residue are properly distributed across residues of the other protein:

$$\Omega_{ij}^{(t+1)} = \frac{\boldsymbol{Z}_{ij}^{(t)}}{\sum_{v=1}^{M} \boldsymbol{Z}_{iv}^{(t)}}, \quad \text{where} \ \ \boldsymbol{Z}_{ij}^{(t)} = \frac{\Omega_{ij}^{(t)}}{\sum_{u=1}^{N} \Omega_{uj}^{(t)}}. \tag{4}$$

The iteration is initialized as $\Omega^{(0)} = \exp(-\mathcal{C}/\tau)$, where $\tau$ is a temperature parameter controlling the alignment sharpness (Appendix J). The optimal $\Omega^\star = \Omega^{(T)}$ after $T$ iterations serves as the Sinkhorn alignment matrix. For simplicity, we denote it as $\Omega$ in the subsequent discussions.

The original Sinkhorn algorithm converges to a fully doubly stochastic matrix, forcing each query residue to distribute across all candidate residues (and vice versa). This strict matching is often biologically meaningless, as most residues lack relevant counterparts. PLASMA achieves implicit partial alignments via two mechanisms. First, *early termination* preserves sparsity by limiting Sinkhorn iterations, letting poorly matching residues retain low weights. Second, the *temperature parameter* $\tau$ controls alignment mass, with lower values producing sparser, focused alignments. Together, these mechanisms emphasize biologically relevant correspondences while avoiding forced matches, without hard constraints on the transport budget (Caffarelli & McCann, 2010; Figalli, 2010). Representative alignment matrices demonstrating these patterns are shown in Appendix I.

## 4  PLAN ASSESSOR

The Plan Assessor receives the alignment matrix $\Omega$ from the Transport Planner and transforms it into an interpretable single similarity score $\kappa \in [0, 1]$ that quantifies the existence and degree of similarity of the aligned substructures. This is computed by first calculating a substructure similarity score for the aligned regions, then adjusting it with a confidence weight to correct potential bias.

**Substructure Similarity** We calculate the alignment score on *matched substructure*. With a threshold $\rho$, a residue pair $r_{q,i} \in \mathcal{P}_q$ and $r_{c,j} \in \mathcal{P}_c$ is treated as matched if $\Omega_{ij} > \rho$. The matched residues then form two sets, $\mathcal{R}_q = \{r_{q,i} \mid \forall j, \Omega_{ij} > \rho\}$ and $\mathcal{R}_c = \{r_{c,j} \mid \forall i, \Omega_{ij} > \rho\}$. A matched substructure is a subset of these residues. The representation of the matched substructure can be approximated by summing the embeddings of residues from $\mathcal{R}_q$ and $\mathcal{R}_c$. Therefore, the *substructure similarity score* $s \in [-1, 1]$ is defined as the cosine similarity between the summed representations:

$$s = \frac{\sum_{i \in \mathcal{R}_q} \boldsymbol{h}_{q,i} \cdot \sum_{j \in \mathcal{R}_c} \boldsymbol{h}_{c,j}}{\| \sum_{i \in \mathcal{R}_q} \boldsymbol{h}_{q,i} \| \cdot \| \sum_{j \in \mathcal{R}_c} \boldsymbol{h}_{c,j} \|}. \tag{5}$$

This substructure similarity score is effective when a sufficient number of residues are matched between the two proteins. However, it becomes less reliable when only a few residues are aligned or when the matched residues are dispersed along the sequence rather than forming a continuous region. In such cases, the score reduces to a residue-level similarity measure, which may appear deceptively high even though the aligned residues do not cluster into a structurally interpretable substructure. We thus introduce a *confidence weight* to adjust the initial similarity score.

**Alignment Score with Confidence Weight Correction** The *confidence weight* $\alpha \in [0, 1]$ is derived from $\Omega$ using a 2D convolution with an identity kernel $K = \mathbb{I}_k \in \mathbb{R}^{k \times k}$ of size $k$:

$$\alpha_{ij} = \sum_{u=0}^{k-1} \sum_{v=0}^{k-1} \Omega_{i+u,j+v} \cdot K_{uv} = \sum_{u=0}^{k-1} \Omega_{i+u,j+u}. \tag{6}$$

This convolution operation highlights continuous diagonal segments in $\Omega$ and emphasizes core regions where consecutive residues in the query align with consecutive residues in the candidate. A max-pooling layer then produces a scalar confidence weight $\alpha = \max_{i,j} \alpha_{ij}$, summarizing the strongest local alignment signal used to weight the similarity score and obtain the final *alignment score* $\kappa = \alpha \cdot s_+ \in [0, 1]$. Here $s_+$ is the non-negative substructure similarity score. This formulation provides an intuitive and interpretable measure: $\kappa = 0$ indicates no residue matches and $\kappa = 1$ represents perfect substructure alignment. We follow the convention of established alignment methods (*e.g.*, TM-align (Zhang, 2005)) and exclude negative similarity values, since matched substructures with opposite orientations in the representation space lack meaningful biological interpretation. See Appendix I for visual examples of alignment matrices with different similarity scores.

## 5 MODEL OPTIMIZATION

PLASMA is trained with two complementary objectives: predicting the presence of aligned substructures via the alignment score $\kappa$ and recovering precise residue-level matches via the alignment matrix $\Omega$. Training data consists of protein pairs $(\mathcal{P}_q, \mathcal{P}_c)$, where a subset of pairs contains matched substructures with shared functions. For each input protein pair, two mask vectors $\mathcal{M}_q \in \{0, 1\}^N$ and $\mathcal{M}_c \in \{0, 1\}^M$ are respectively defined to indicate the position of target substructures $\mathcal{F}_q$ and $\mathcal{F}_c$, where 1 marks the residues that belong to the substructure of interest.

**Alignment Score Optimization** The alignment score $\kappa$ serves as the model's prediction on whether the input protein pair contains aligned substructures. We define the ground truth $y = 1$ if the pair contains matched substructures and $y = 0$ otherwise. The prediction is optimized by $\mathcal{L}_{\text{BCE}} = -y \log(\sigma(\kappa)) - (1 - y) \log(1 - \sigma(\kappa))$, where $\sigma(\cdot)$ is the sigmoid function.

**Alignment Matrix Optimization** Unlike the alignment score, optimizing the alignment matrix is challenging because unlabeled residues may correspond to valid but unannotated matches. Treating these residues as negative examples would impose inappropriate penalties on the model. To address this, we propose the *Label Match Loss* (LML) to focus exclusively on the labeled substructures. Specifically, when $\|\mathcal{M}_c\|_1 > 0$ and $\|\mathcal{M}_q\|_1 > 0$, the LML for protein pairs is defined as

$$\mathcal{L}_{\text{LML}} = \|[\mathcal{M}_c - \Omega^\top \mathcal{M}_q]_+\|_1 / \|\mathcal{M}_c\|_1, \tag{7}$$

where $[\cdot]_+$ retains only non-negative elements, and $\|\cdot\|_1$ denotes the $\ell_1$ norm. This loss evaluates how well the constructed alignment matrix $\Omega$ aligns the labeled substructures $(\mathcal{F}_q, \mathcal{F}_c)$ in $(\mathcal{P}_q, \mathcal{P}_c)$. For

each residue $r_j \in \mathcal{P}_c$, $(\Omega^\top \mathcal{M}_q)_j$ gives the alignment weight with respect to labeled residues in $\mathcal{P}_q$. The non-negative contributions by $[\mathcal{M}_c - \Omega^\top \mathcal{M}_q]_+$ are normalized by $\|\mathcal{M}_c\|_1$ across all labeled residues. When no labeled substructures exist, $\mathcal{L}_{\text{LML}} = 0$, which allows the model to focus on known substructures without penalizing unlabeled but potentially valid matches. This loss provides an optional bias toward annotated local structural motifs when such labels exist. These regions are typically small and structurally meaningful (*e.g.*, catalytic or binding motifs), and emphasizing them helps the model avoid being dominated by background alignments.

The final $\mathcal{L} = \mathcal{L}_{\text{BCE}} + \mathcal{L}_{\text{LML}}$ jointly detects substructure existence by $\kappa$ and localizes known substructures by $\Omega$, while staying robust to missing or incomplete labels in the training data.

## 6 EMPIRICAL ANALYSIS

We conduct extensive quantitative and qualitative evaluations to comprehensively assess the validity and advancement of PLASMA in local structural motif alignment tasks. All experiments are programmed with PyTorch v2.5.1 and run on NVIDIA RTX 4090 32 GB GPU.

### 6.1 EXPERIMENTAL SETUP

**Prediction Tasks and Benchmark Datasets**    Our experiments are based on a residue-level functional alignment benchmark, **VenusX** (Tan et al., 2025a). We consider three common classes of functional substructures: activation sites, binding sites, and motifs. Across all test sets, the sequence identity between training and test proteins is kept below $50\%$. For quantitative evaluation, we design two levels of difficulty: (i) interpolation (test_inter), where the test set contains proteins from InterPro families already present in training; and (ii) extrapolation (test_extra), where the test set only includes novel substructures from unseen families. Further details are in Appendix C.1.

**Baseline Methods**    We compare PLASMA with popular baselines in protein structure alignment, including structure-based methods (FOLDSEEK (Van Kempen et al., 2024), TM-ALIGN (Zhang, 2005), and TM-VEC (Hamamsy et al., 2024)) and embedding-based methods (EBA (Pantolini et al., 2024) and COSINESIM, a cosine similarity over protein embeddings). For all embedding-based methods, we implement seven popular pre-trained models to extract residue-level sequence and structure representations, including PROTT5 (Elnaggar et al., 2021), PROSTT5 (Heinzinger et al., 2024), ANKH (Elnaggar et al., 2023), ESM2 (Lin et al., 2023), PROTBERT (Brandes et al., 2022), TM-VEC (Hamamsy et al., 2024), and PROTSSN (Tan et al., 2025b). All baselines use the authors' official code and checkpoints (see Appendices D for details).

**Evaluation Metrics**    To assess the ability to detect the existence of local structural motifs, we use standard binary classification metrics, including ROC-AUC, PR-AUC, and F1-Max. Additionally, to evaluate alignment quality, we introduce the Label Match Score (LMS) by (7) with $\text{LMS} = 1 - \text{LML}$ to measure correspondence between predicted alignments and annotated functional regions.

### 6.2 QUANTITATIVE PERFORMANCE EVALUATION

Table 1 reports performance on test_extra, which contains functional substructures from protein families not seen during training. This setting evaluates the generalizability of the alignment framework, which is essential in practice because new functional substructures are continuously discovered. Full results on seven backbone models are provided in Appendix F, and all hyperparameter and dataset details are summarized in Appendix C.2. Corresponding interpolation results on test_inter are reported in Appendix E.

Across all three substructure detection tasks and all evaluation metrics, PLASMA achieves consistent top performance, highlighting its robustness in capturing fundamental local structural similarities for novel substructures beyond the training distribution. PLASMA-PF also performs strongly and remains competitive without task-specific training. However, unlike in the interpolation setting, PLASMA-PF does not surpass the learnable PLASMA variant on test_extra; this emphasizes the value of supervised examples in improving alignment accuracy for entirely new functional substructures. In contrast, baseline methods show large performance variation across backbone models.

Table 1: Model performance on `test_extra` (mean ± std over three independent seeds). Colors indicate relative performance versus TM-ALIGN.

| Metrics | Methods | Motif | | | Binding Site | | | Active Site | | |
|---|---|---|---|---|---|---|---|---|---|---|
| | | ANKH | ESM2 | PROTSSN | ANKH | ESM2 | PROTSSN | ANKH | ESM2 | PROTSSN |
| ROC-AUC | PLASMA | $.98_{\pm.008}$ | $.97_{\pm.013}$ | $.96_{\pm.016}$ | $.99_{\pm.008}$ | $.98_{\pm.013}$ | $.98_{\pm.014}$ | $.98_{\pm.012}$ | $.98_{\pm.010}$ | $.97_{\pm.011}$ |
| | PLASMA-PF | $.98_{\pm.009}$ | $.93_{\pm.004}$ | $.90_{\pm.005}$ | $.99_{\pm.006}$ | $.92_{\pm.052}$ | $.96_{\pm.012}$ | $.97_{\pm.015}$ | $.96_{\pm.006}$ | $.97_{\pm.008}$ |
| | EBA | $.90_{\pm.033}$ | $.92_{\pm.021}$ | $.32_{\pm.043}$ | $.99_{\pm.007}$ | $.97_{\pm.021}$ | $.30_{\pm.060}$ | $.97_{\pm.013}$ | $.97_{\pm.012}$ | $.43_{\pm.066}$ |
| | Backbone | $.85_{\pm.019}$ | $.74_{\pm.033}$ | $.79_{\pm.018}$ | $.98_{\pm.010}$ | $.72_{\pm.060}$ | $.70_{\pm.070}$ | $.96_{\pm.012}$ | $.79_{\pm.068}$ | $.76_{\pm.033}$ |
| | Foldseek | | $.89_{\pm.033}$ | | | $.90_{\pm.013}$ | | | $.87_{\pm.022}$ | |
| | TM-Align | | $.81_{\pm.014}$ | | | $.91_{\pm.040}$ | | | $.93_{\pm.009}$ | |
| PR-AUC | PLASMA | $.98_{\pm.011}$ | $.97_{\pm.014}$ | $.96_{\pm.017}$ | $.98_{\pm.011}$ | $.97_{\pm.019}$ | $.97_{\pm.019}$ | $.97_{\pm.014}$ | $.98_{\pm.011}$ | $.97_{\pm.012}$ |
| | PLASMA-PF | $.98_{\pm.010}$ | $.95_{\pm.005}$ | $.92_{\pm.007}$ | $.98_{\pm.012}$ | $.90_{\pm.079}$ | $.95_{\pm.026}$ | $.97_{\pm.015}$ | $.96_{\pm.006}$ | $.97_{\pm.009}$ |
| | EBA | $.91_{\pm.035}$ | $.93_{\pm.019}$ | $.38_{\pm.014}$ | $.98_{\pm.012}$ | $.96_{\pm.035}$ | $.28_{\pm.063}$ | $.97_{\pm.012}$ | $.97_{\pm.012}$ | $.43_{\pm.032}$ |
| | Backbone | $.86_{\pm.023}$ | $.77_{\pm.041}$ | $.82_{\pm.027}$ | $.96_{\pm.023}$ | $.67_{\pm.093}$ | $.65_{\pm.118}$ | $.96_{\pm.016}$ | $.84_{\pm.059}$ | $.80_{\pm.038}$ |
| | Foldseek | | $.84_{\pm.031}$ | | | $.76_{\pm.065}$ | | | $.81_{\pm.026}$ | |
| | TM-Align | | $.86_{\pm.020}$ | | | $.89_{\pm.064}$ | | | $.94_{\pm.012}$ | |
| F1-MAX | PLASMA | $.97_{\pm.009}$ | $.95_{\pm.018}$ | $.92_{\pm.022}$ | $.96_{\pm.022}$ | $.95_{\pm.030}$ | $.93_{\pm.026}$ | $.98_{\pm.013}$ | $.97_{\pm.011}$ | $.97_{\pm.011}$ |
| | PLASMA-PF | $.96_{\pm.013}$ | $.90_{\pm.006}$ | $.84_{\pm.008}$ | $.96_{\pm.027}$ | $.85_{\pm.082}$ | $.90_{\pm.031}$ | $.97_{\pm.018}$ | $.94_{\pm.016}$ | $.95_{\pm.012}$ |
| | EBA | $.86_{\pm.035}$ | $.87_{\pm.024}$ | $.00_{\pm.000}$ | $.97_{\pm.021}$ | $.93_{\pm.049}$ | $.00_{\pm.000}$ | $.97_{\pm.013}$ | $.97_{\pm.008}$ | $.00_{\pm.000}$ |
| | Backbone | $.79_{\pm.008}$ | $.70_{\pm.014}$ | $.73_{\pm.013}$ | $.91_{\pm.034}$ | $.62_{\pm.087}$ | $.60_{\pm.107}$ | $.92_{\pm.020}$ | $.75_{\pm.044}$ | $.71_{\pm.018}$ |
| | Foldseek | | $.91_{\pm.046}$ | | | $.97_{\pm.014}$ | | | $.96_{\pm.015}$ | |
| | TM-Align | | $.76_{\pm.015}$ | | | $.87_{\pm.063}$ | | | $.90_{\pm.014}$ | |
| LMS | PLASMA | $.75_{\pm.045}$ | $.69_{\pm.019}$ | $.52_{\pm.046}$ | $.82_{\pm.062}$ | $.77_{\pm.105}$ | $.65_{\pm.088}$ | $.90_{\pm.034}$ | $.87_{\pm.038}$ | $.67_{\pm.044}$ |
| | PLASMA-PF | $.78_{\pm.055}$ | $.48_{\pm.074}$ | $.23_{\pm.021}$ | $.85_{\pm.058}$ | $.49_{\pm.082}$ | $.36_{\pm.055}$ | $.94_{\pm.029}$ | $.68_{\pm.067}$ | $.43_{\pm.032}$ |

| Best | Baseline (TM-Align) | Worst |
|---|---|---|

EBA performs reasonably well with sequence-based ANKH and ESM2 yet drops substantially with structure-based PROTSSN, especially under the extrapolation split. FOLDSEEK and TM-ALIGN remain consistently below PLASMA across nearly all conditions, reflecting the limited usefulness of global structural similarity for residue-level motif detection.

Beyond accuracy, PLASMA demonstrates exceptional computational efficiency. As shown in Figure 2, PLASMA achieves the best performance while requiring minimal time per protein pair—approximately 10ms for PLASMA and 7ms for PLASMA-PF. This represents a roughly 50 times speedup over global structure alignment methods like TM-Align and Foldseek, which require costly structural superposition, and about 3 times faster than EBA due to PLASMA's fully differentiable OT formulation that is efficiently accelerated on GPUs, compared to EBA's inherently sequential dynamic programming approach.

## 6.3 QUALITY OF PREDICTED ALIGNMENTS

Beyond quantitative metrics, we assess PLASMA's robustness in identifying biologically meaningful substructures by examining both alignment scores and alignment matrices.

PLASMA effectively distinguishes proteins with shared local functional substructures even when overall structural similarity is low. Figure 3 provides evidence from two perspectives, with all embedding-based methods obtaining protein representations from ANKH. Figure 3A compares similarity score distributions for protein pairs from `test_inter`, where PLASMA and PLASMA-PF clearly separate positive and negative pairs. This advantage comes from the OT framework, which emphasizes local correspondences independent of overall similarity. In contrast, EBA and COSINESIM show substantial overlap between positive and negative distributions. EBA in particular lacks an upper bound on its scores, making them difficult to interpret and subject to calibration problems (*i.e.*, scores cannot be directly used as probabilities and lead to unstable thresholds). Figure 3B further groups test-set alignment scores by TM-score to assess performance under different levels of global similarity for protein pairs. Although all methods degrade as TM-score decreases, PLASMA and PLASMA-PF consistently maintain high ROC-AUC values above 0.9, whereas baseline EBA, COSINESIM, Foldseek, and TM-align deteriorate sharply on low-similarity samples when TM-score is sufficiently small (*e.g.*, lower than 0.5).

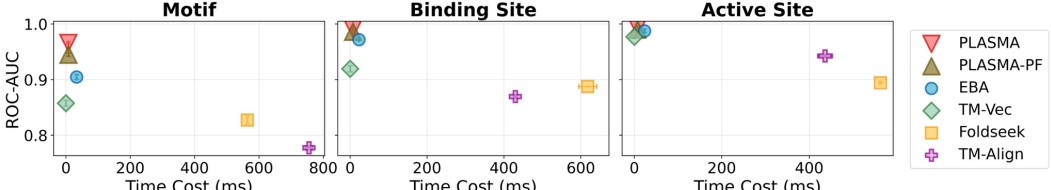

Figure 2: Performance versus computational efficiency comparison. ROC-AUC scores plotted against inference time (milliseconds) for motif and binding/active site detection using PROSTT5 embeddings. Points represent averages across three splits with standard error bars on both axes.

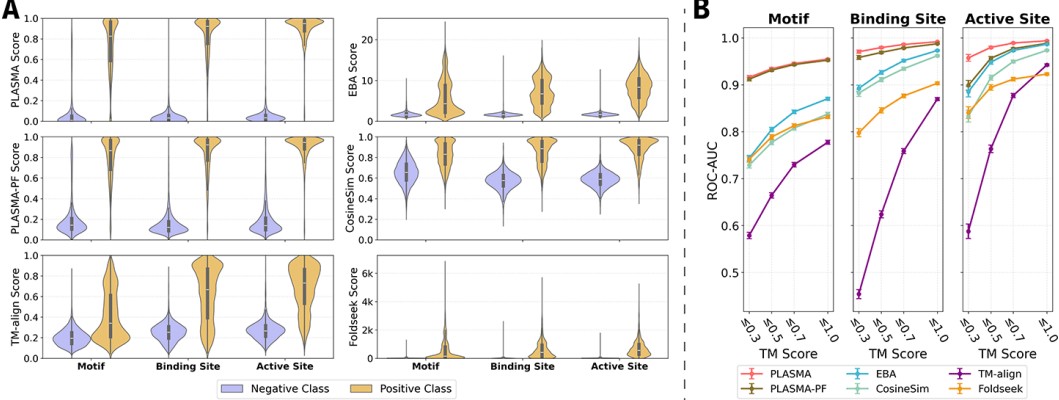

Figure 3: Alignment quality analysis across different approaches. **A**. Distribution of alignment scores for positive and negative protein pairs. **B**. ROC-AUC score trend at different global structural similarity levels.

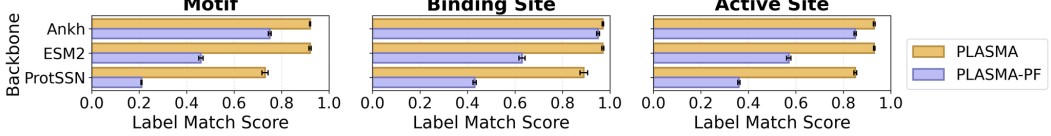

Figure 4: Label Match Score comparison between PLASMA and PLASMA-PF across different substructure types, demonstrating the improved alignment quality achieved through training.

While both PLASMA variants demonstrate strong performance in score-based discrimination, their alignment quality differs. This is evident in Figure 4, which compares their performance using the LMS score to evaluate correspondence between predicted alignments and annotated regions. PLASMA consistently outperforms PLASMA-PF across motifs, binding sites, and active sites, demonstrating that learning improves the prediction of local structural motifs. By contrast, while EBA also produces alignment matrices, it cannot be meaningfully assessed with LMS: its unconstrained formulation yields a maximal LMS of 1.0 regardless of true alignment accuracy.

## 6.4 REPRESENTATIVE ALIGNMENT EXAMPLES

The next experiment evaluates PLASMA's utility in real biological applications. We examine three protein pairs of different substructure sizes (independent of the training set), including simple local motifs, complex cofactor-binding domains, and extended multi-element substructures. In each case, we provide UniProt identifiers, functional descriptions, alignment results, and visualizations from PLASMA and EBA, and corresponding analyses. Appendix N provides additional visualizations that further illustrate the generality of these conclusions. Collectively, these cases show PLASMA detects biologically meaningful local similarities across diverse sequences, structures, and functions.

**Conserved Small Helical Motifs Across Functionally Diverse Protein Structures** The first case matches local structures between P40343 (Vps27, a yeast ESCRT-0 complex component) and

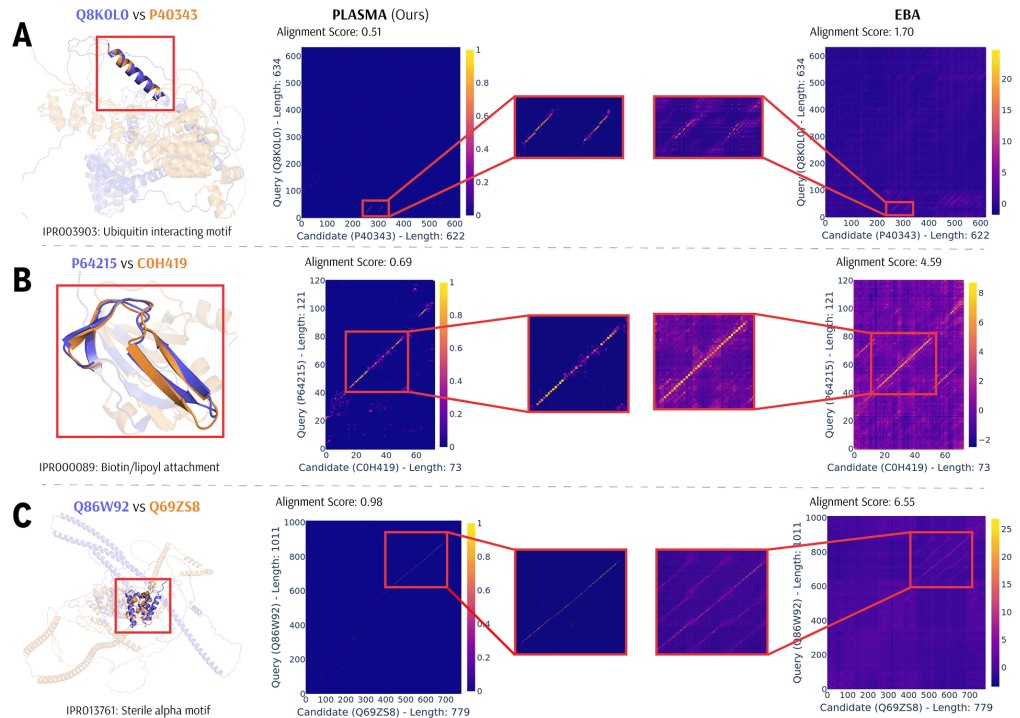

Figure 5: Representative alignment examples across three protein pairs. **A**, P40343 vs Q8K0L0. **B**, P64215 vs C0H419. **C**, Q69ZS8 vs Q86W92. Left: 3D structures with highlighted aligned regions. Center and right: alignment matrices from PLASMA and EBA with zoomed insets. A higher resolution version of this figure can be found at Appendix H.

Q8K0L0 (ASB2, a mouse E3 ubiquitin ligase substrate-recognition component). The two proteins share no apparent sequence homology (21.0% identity) and participate in distinct cellular processes (endosomal sorting versus proteasomal degradation), yet both use analogous helical arrangements for protein-protein interactions: Vps27's GAT domain forms coiled-coils for ESCRT-I recruitment (Curtiss et al., 2007), whereas ASB2 employs ankyrin repeat helices for substrate recognition in the E3 ligase complex. PLASMA assigns high-confidence scores to residues mediating these interactions (Figure 5A). The 3D structure visualization also confirms the alignment of the conserved Leu-X-X-Leu-Leu motif for both proteins (Ren et al., 2008), with an aligned RMSD of 0.18 Å. This finding suggests potential convergent evolution of helical protein-binding interfaces across distinct cellular machineries. By contrast, EBA identifies multiple helices, but most correspond to nonfunctional scaffold regions rather than the relevant interaction motifs.

**Structurally and Functionally Relevant motifs of Different Sizes and Metabolic Contexts**    The second case examines P64215 (GcvH, glycine cleavage system H protein from Mycobacterium tuberculosis) and C0H419 (YngHB, biotin/lipoyl attachment protein from Bacillus subtilis) (Cui et al., 2006). These proteins have different overall sequences (25.2% sequence identity) and metabolic functions: GcvH shuttles methylamine groups in glycine catabolism, while YngHB accommodates both biotin and lipoic acid in a single-domain architecture. Despite these differences, both bind similar cofactors and exhibit conserved $\beta$-sheet arrangements necessary for post-translational modification. As shown in Figure 5B, PLASMA successfully aligns the four-stranded $\beta$-barrel architectures, highlighting the critical lysine-containing $\beta$-turns with an overall alignment score of 0.69 and RMSD of 0.83, whereas the baseline EBA misaligns nonfunctional regions. The alignment of complex conserved structural motifs across protein families demonstrates the potential of PLASMA in revealing modular evolution and conserved cofactor-binding architectures.

**Extended Multi-Element Substructures in Cell Adhesion Regulators**    The third case investigates Q69ZS8 (Kazrin, a scaffold protein in Mus musculus) and Q86W92 (Liprin-$\beta$1/PPFIBP1, a human focal adhesion regulator). Despite their different cellular localizations and interaction part-

ners, they regulate distinct but mechanistically related aspects of cell-cell adhesion: Kazrin organizes desmosomal components in keratinocytes, and Liprin-$\beta 1$ modulates focal adhesion disassembly and cell migration. Yet both proteins rely on extended $\alpha$-helical regions for protein-protein interactions (Groot et al., 2004). As in Figure 5C, PLASMA successfully aligns complex multi-coil substructures spanning multiple helical segments interspersed with flexible linkers, with an overall alignment score of $0.98$ and RMSD $0.82$ Å. The alignment highlights conserved leucine-rich motifs and hinge regions that stabilize oligomerization interfaces, revealing analogous scaffolding strategies. In contrast, EBA identifies plausible structures but often misaligns helices or matches nonfunctional scaffold regions, failing to capture more than just biologically meaningful substructures.

## 7 RELATED WORKS

**Protein Global Structure Alignment** Global structure alignment methods evaluate overall protein similarity. Classic approaches like TM-ALIGN (Zhang, 2005) are foundational, while modern methods increase efficiency by abstracting structures into 1D sequences (FOLDSEEK (Van Kempen et al., 2024)), representing them as fixed vectors for rapid search (TM-VEC (Hamamsy et al., 2024)), or using advanced spatial indexing (GTALIGN (Margelevičius, 2024)). The field has also expanded to align multiple structures (MTM-ALIGN (Dong et al., 2018)), multi-chain complexes (MM-ALIGN (Mukherjee & Zhang, 2009)), and diverse macromolecules universally (US-ALIGN (Zhang et al., 2022)). However, their global nature limits the detection of conserved motifs in dissimilar proteins.

**Substructure and Sequence-based Alignment** To find local similarities, substructure-based methods use graph-based residue embeddings (Tan et al., 2024), focus on active-site environments (Castillo & Ollila, 2025), or apply linear-assignment formulations (Zhang et al., 2025). PLM-based residue representations are also widely used from raw embedding similarity scoring (Kaminski et al., 2023; Liu et al., 2024) to learned alignment models and embedding-aware dynamic programming (Llinares-López et al., 2023; Iovino & Ye, 2024). OT-based differentiable graph matching has been used to learn structure/function-aware substitution matrices (Pellizzoni et al., 2024), with a primary focus on learning matching costs. PLASMA instead targets residue-level local substructure alignment, producing explicit mappings with practical speed and interpretability. Meanwhile, embedding-score-based alignment methods remain hard to interpret quantitatively, as their scores are essentially unbounded (Pantolini et al., 2024).

## 8 CONCLUSION AND DISCUSSION

This work presents PLASMA, a local structural motif alignment framework leveraging regularized optimal transport to detect biologically meaningful local similarities across proteins with diverse sequences, structures, and functions. PLASMA consistently outperforms baseline methods in accuracy, efficiency, and interpretability, capturing subtle structural correspondences often invisible to global alignments. Its trainable variant benefits from supervision to improve alignment precision, while the training-free variant achieves robust performance without task-specific labels.

Beyond quantitative performance, PLASMA provides clear, residue-level alignment matrices that support mechanistic insights into protein function, evolutionary relationships, and structure-guided protein engineering. Its ability to handle varying substructure sizes and complexities (*e.g.*, from short helices to extended multi-element domains) demonstrates versatility and practical relevance. Overall, PLASMA establishes a new standard for accurate, efficient, interpretable, and practically applicable protein local structural motif alignment.

ACKNOWLEDGMENTS

This work was supported by the grants from National Science Foundation of China (Grant Number 92451301; 62302291), the AI for Science Program by Shanghai Municipal Commission of Economy and Informatization (2025-GZL-RGZN-BTBX-02009), the National Key Research and Development Program of China (2024YFA0917603), and Computational Biology Key Program of Shanghai Science and Technology Commission (23JS1400600).

**Reproducibility Statement**  To promote reproducibility, we release all source code and trained models under an open-source license, which is available at `https://github.com/ZW471/PLASMA-Protein-Local-Alignment.git`. Details of data sources are provided in Appendix C.1. Task definitions, evaluation protocols, and hyperparameter settings are described in Sections 6.1 and Appendices C.2. Implementation details and instructions for reproducing experiments are included in the project repository to facilitate independent verification.

**Ethics Statement**  All experiments are conducted on publicly available protein sequence and structure databases. We follow established ethical guidelines in data usage and acknowledge that historical biases present in these resources may be reflected in our results, which is independent of model development.

**The Use of Large Language Models (LLM)**  In the preparation of this manuscript, GPT-5 and GPT-4o were utilized as writing assistants. The usage was strictly limited to improving grammar, clarity, and overall readability. All scientific ideas, experimental results, and conclusions were conceived and formulated exclusively by the authors. All text polished or modified by the LLM was subsequently reviewed and edited by the authors to ensure that the original scientific meaning was accurately preserved.

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
