This appendix provides additional details, analyses, and results that complement the main paper.

- Appendix A gives the full derivation of our OT objective.

- Appendix B presents a more precise discussion of computational cost.

- Appendix C describes the benchmark datasets (**VenusX**) and the hyperparameter configuration.

- Appendix D summarizes all comparison methods, including global structure alignment, global embedding-based alignment, local embedding-based alignment, and the backbone models.

- Appendix E and Appendix F report complete quantitative results for all backbones on `test_inter` and `test_extra` split.

- Appendix G provides further insight into the contribution of individual components.

- Appendices H-I contains additional visualizations of alignment matrices and case studies.

- Appendices J-M offer more detailed quantitative analyses of model behaviour under different settings.

## A  OPTIMAL TRANSPORT FORMULATION FOR PROTEIN ALIGNMENT

To circumvent the computational bottleneck of explicit fragment enumeration, we reframe the alignment problem as finding optimal correspondences between individual residues rather than predefined fragments. This approach leverages optimal transport theory, which provides a principled framework for finding the most efficient assignment between two sets of points based on their similarity and a transportation cost function.

Specifically, we model protein substructure alignment as an entropy regularized optimal transport problem that determines how to optimally redistribute alignment weights from query residues to candidate residues. Instead of relying solely on explicit structural coordinates, this formulation operates on learned residue representations that encode local neighborhood properties, biochemical characteristics, and structural context. The optimal transport solver then identifies which residues should be matched by minimizing the total transportation cost—effectively the sum of dissimilarities between matched residue pairs—across the embedding space.

This approach naturally produces soft, many-to-many alignments where functionally and structurally similar residues are preferentially matched, while simultaneously identifying the corresponding aligned fragments without explicit enumeration. Mathematically, we formulate this as the following optimal transport problem with entropic constraints:

$$\min_{\Omega} \quad \sum_{i=1}^{N}\sum_{j=1}^{M} \Omega_{ij}\mathcal{C}_{ij} - \lambda \sum_{i=1}^{N}\sum_{j=1}^{M} \Omega_{ij}\log(\Omega_{ij}) \tag{8}$$

$$\text{subject to:} \quad \sum_{j=1}^{M} \Omega_{ij} = 1, \quad \forall i \in \{1,\ldots,N\} \tag{9}$$

$$\sum_{i=1}^{N} \Omega_{ij} = 1, \quad \forall j \in \{1,\ldots,M\} \tag{10}$$

$$\Omega_{ij} \geq 0, \quad \forall i,j \tag{11}$$

Here, $\Omega \in \mathbb{R}^{N \times M}$ is the transport plan (alignment matrix), $\mathcal{C}_{ij}$ represents the cost of aligning query residue $i$ to candidate residue $j$, and $\lambda$ is the entropic regularization parameter that controls the smoothness of the alignment. This optimization seeks to find the optimal transport plan that minimizes the total alignment cost while the entropic regularization term ($-\lambda$ term) encourages smooth, distributed assignments rather than hard one-to-one mappings. The equality constraints ensure each query residue distributes 1 total weight and each candidate residue receives 1 total weight.

## B    COMPLEXITY ANALYSIS

PLASMA achieves optimal $O(N^2)$ complexity while maintaining full differentiability. The cost matrix computation dominates computational requirements, requiring $O(N \cdot M \cdot D) = O(N^2 \cdot D)$ operations for the hinge non-linearity between proteins of lengths $N$ and $M$, where $D$ represents the embedding dimension. The siamese network contributes $O(N \cdot D^2)$ operations per protein (if using a two-layer MLP), yielding total $O(N \cdot D^2)$ since $D \ll N$ in practice. The Sinkhorn algorithm requires $O(T \cdot N^2)$ operations where $T$ represents the number of iterations (typically $T \ll N$). The Plan Assessor contributes $O(N^2)$ for substructure similarity computation and $O(K^2 \cdot N^2)$ for confidence weight calculation via diagonal convolution with kernel size $K \ll N$. The overall complexity remains $O(N^2)$, matching the best achievable complexity of the methods based on dynamic programming.

## C    DETAILED EXPERIMENTAL SETUP

### C.1    BENCHMARK DATASETS: VENUSX

We construct our evaluation datasets from the **VenusX** (Tan et al., 2025a) benchmark (https://github.com/ai4protein/VenusX), which provides protein pairs with annotated biologically important substructures curated from the InterPro (Blum et al., 2025) database. We focus on three substructure types: activation sites, binding sites, and motifs, corresponding to the `VenusX_Res_{Act/BindI/Motif}_MP50` datasets where protein pairs share less than 50% sequence similarity. These datasets present increasing difficulty due to their substructure sizes: active sites ($18.7 \pm 7.0$ residues), binding sites ($26.6 \pm 21.7$ residues), and motifs ($80.23 \pm 73.8$ residues). From each VenusX dataset, we generate 20,000 protein pairs with balanced labels: half sharing the same InterPro family ID (positive pairs, $y = 1$) and half from different families (negative pairs, $y = 0$). Each sample is represented as $(\mathcal{P}_q, \mathcal{P}_c, \mathbf{l}_q, \mathbf{l}_c, y)$, where $\mathcal{P}_q$ and $\mathcal{P}_c$ are the protein pair, $\mathbf{l}_q$ and $\mathbf{l}_c$ are their respective substructure annotations, and $y$ indicates family membership.

To evaluate all the embedding based methods' generalization capability across different evolutionary contexts, we create two complementary test scenarios using three different random seeds for robust evaluation. This dual evaluation is crucial for protein analysis since biological systems constantly encounter both familiar protein families with slight variations and entirely novel protein architectures through evolution, horizontal gene transfer, and structural convergence. First, we randomly exclude 10% of InterPro family IDs and split the remaining data into training (75%), validation (5%), and `test_inter` (20%). `test_inter` evaluates *interpolation* performance—the model's ability to recognize substructure similarities within the distribution of known protein families, mimicking scenarios where researchers analyze variants of well-characterized proteins. Second, we create `test_extra` by sampling an equivalent number of protein pairs exclusively from the excluded InterPro families (maintaining the same 50–50 balance between positive and negative pairs). `test_extra` evaluates *extrapolation* performance—the model's ability to identify functional similarities in completely novel protein families, which is critical for annotating newly discovered proteins, understanding convergent evolution, and predicting function in understudied organisms. For each test scenario, the data exclusion and splitting procedure is repeated across three different seeds (1, 42, and 100) to ensure statistical reliability.

### C.2    HYPERPARAMETER CONFIGURATION

For both PLASMA and PLASMA-PF variants, we employ the following hyperparameters: the siamese network uses a hidden dimension of 512 to balance expressiveness with computational efficiency. To ensure computational feasibility while maintaining statistical significance, our training sets only use 1500 protein pairs by sampling 10% of the full training set. The Sinkhorn temperature parameter $\tau$ is set to 0.1 to encourage sparse, focused alignments that highlight the most relevant correspondences. The diagonal convolution kernel size $K = 10$ captures sequential patterns in alignment matrices, while the residue matching threshold $\rho = 0.5$ defines when transport weights indicate meaningful correspondences between residue pairs. See Appendix M for detailed sensitivity analysis and justification of these choices.

# D BASELINES

## D.1 GLOBAL STRUCTURE ALIGNMENT METHODS

Traditional structural biology approaches rely on atomic coordinates to identify protein similarities:

- TM-ALIGN (Zhang, 2005) represents the gold standard for protein structure alignment based on Template Modeling scores. This method performs geometric alignment of protein backbones to identify structurally similar regions.

- FOLDSEEK (Van Kempen et al., 2024) performs structural alignment using 3Di tokenizations, converting 3D structural information into sequence-like representations for comparison.

## D.2 GLOBAL EMBEDDING-BASED ALIGNMENT

COSINESIM methods employ direct cosine similarity between globally aggregated protein embeddings from the backbone models discussed in Appendix D.4, similar to the approach used in TM-Vec (Hamamsy et al., 2024). This approach provides a baseline for embedding-based similarity without explicit residue-level alignment, representing proteins as single vectors and measuring their similarity through cosine distance.

## D.3 LOCAL EMBEDDING-BASED ALIGNMENT

EBA (Pantolini et al., 2024) represents the current state-of-the-art in local embedding-based alignment, combining statistical alignment with neural embeddings to identify similar substructures. This method performs local alignment at the residue level using learned representations.

## D.4 BACKBONES

We evaluate PLASMA with seven popular protein sequence and structure representation models, using the following specific versions and configurations:

- ANKH (Elnaggar et al., 2023): We employ the `base` model variant, which is a compact encoder-decoder architecture optimized for protein sequences with 110 million parameters. This model was trained on protein sequences using a masked language modeling objective and represents one of the most parameter-efficient protein language models. *Available at:* `https://huggingface.co/ElnaggarLab/ankh-base`

- ESM2 (Lin et al., 2023): We utilize the `t33_650M_UR50D` variant, a 650-million parameter encoder-only transformer model with 33 layers. This model was trained on the UniRef50 database and represents one of the largest and most comprehensive protein language models available, providing rich contextual representations for protein analysis. *Available at:* `https://huggingface.co/facebook/esm2_t33_650M_UR50D`

- PROSTT5 (Heinzinger et al., 2024): We use the `AA2fold` checkpoint, which is specifically fine-tuned for protein folding applications. This bilingual language model can process both amino acid sequences and structural information, making it particularly well-suited for structure-aware protein analysis tasks. *Available at:* `https://huggingface.co/Rostlab/ProstT5`

- PROTT5 (Elnaggar et al., 2021): We employ the `xl_half_uniref50-enc` model, which uses only the encoder component of the T5 architecture. This variant was trained on UniRef50 (Suzek et al., 2007) sequences and provides balanced performance between computational efficiency and representation quality with approximately 3 billion parameters. *Available at:* `https://huggingface.co/Rostlab/prot_t5_xl_half_uniref50-enc`

- PROTSSN (Tan et al., 2025b): We utilize the `k20_h512` configuration, which combines sequence and structural information through a hybrid architecture. The model uses $k = 20$ nearest neighbors for structural context and hidden dimensions of $512$, enabling it to capture both sequential and geometric protein properties. *Available at:* `https://github.com/tyang816/ProtSSN`

- TM-VEC (Hamamsy et al., 2024): We employ the `cath_model_large` variant, which was specifically trained on the CATH structural classification database (Knudsen & Wiuf, 2010). This

model specializes in learning structure-aware representations and is particularly effective for detecting remote homology relationships based on structural similarity. *Available at:* `https://figshare.com/articles/dataset/TMvec_DeepBLAST_models/25810099`

- PROTBERT (Brandes et al., 2022): We use the `bfd` checkpoint, which was trained on the Big Fantastic Database (Jumper et al., 2021) containing over 2.1 billion protein sequences. This BERT-based model provides robust protein representations through bidirectional context modeling and large-scale pretraining. *Available at:* `https://huggingface.co/Rostlab/prot_bert_bfd`

## E  FULL INTERPOLATION PERFORMANCE COMPARISON

This section presents comprehensive experimental results using seven backbone protein representation learning models (PROSTT5, PROTT5, ANKH, ESM2, PROTSSN, TM-VEC, and PROTBERT) across three substructure alignment tasks (motifs, binding sites, and active sites) on the `test_inter` dataset. The key findings demonstrate that both PLASMA and PLASMA-PF consistently achieve superior performance across all backbone-task combinations, highlighting the robustness of our optimal transport framework regardless of the underlying protein representation model. Additionally, the Label Match Score (LMS) results show that the trainable PLASMA variant significantly outperforms the parameter-free PLASMA-PF in predicting precise locations of aligned substructures, validating the benefits of supervised learning for accurate residue-level alignment localization.

Table 2: Comprehensive motif detection results on `test_inter` dataset across seven protein representation models.

| Metrics | Methods | Motif | | | | | | |
|---|---|---|---|---|---|---|---|---|
| | | PROSTT5 | PROTT5 | ANKH | ESM2 | PROTSSN | TM-VEC | PROTBERT |
| ROC-AUC | PLASMA | $.97_{\pm.002}$ | $.97_{\pm.002}$ | $.95_{\pm.002}$ | $.96_{\pm.002}$ | $.96_{\pm.001}$ | $.92_{\pm.004}$ | $.87_{\pm.004}$ |
| | PLASMA-PF | $.94_{\pm.003}$ | $.96_{\pm.002}$ | $.95_{\pm.003}$ | $.93_{\pm.004}$ | $.91_{\pm.003}$ | $.87_{\pm.001}$ | $.85_{\pm.004}$ |
| | EBA | $.90_{\pm.004}$ | $.91_{\pm.004}$ | $.87_{\pm.005}$ | $.88_{\pm.003}$ | $.44_{\pm.002}$ | $.88_{\pm.004}$ | $.73_{\pm.006}$ |
| | COSINESIM | $.82_{\pm.008}$ | $.87_{\pm.003}$ | $.84_{\pm.006}$ | $.73_{\pm.009}$ | $.75_{\pm.006}$ | $.86_{\pm.005}$ | $.57_{\pm.014}$ |
| | FOLDSEEK | $.83_{\pm.007}$ | | | | | | |
| | TM-ALIGN | $.78_{\pm.003}$ | | | | | | |
| PR-AUC | PLASMA | $.96_{\pm.002}$ | $.96_{\pm.003}$ | $.95_{\pm.002}$ | $.97_{\pm.001}$ | $.96_{\pm.001}$ | $.93_{\pm.004}$ | $.89_{\pm.003}$ |
| | PLASMA-PF | $.95_{\pm.003}$ | $.96_{\pm.002}$ | $.95_{\pm.003}$ | $.94_{\pm.002}$ | $.92_{\pm.001}$ | $.88_{\pm.002}$ | $.87_{\pm.003}$ |
| | EBA | $.92_{\pm.004}$ | $.93_{\pm.004}$ | $.90_{\pm.004}$ | $.90_{\pm.004}$ | $.45_{\pm.004}$ | $.91_{\pm.004}$ | $.78_{\pm.005}$ |
| | COSINESIM | $.85_{\pm.005}$ | $.88_{\pm.002}$ | $.86_{\pm.005}$ | $.76_{\pm.008}$ | $.78_{\pm.006}$ | $.88_{\pm.002}$ | $.63_{\pm.016}$ |
| | FOLDSEEK | $.78_{\pm.008}$ | | | | | | |
| | TM-ALIGN | $.83_{\pm.004}$ | | | | | | |
| F1-MAX | PLASMA | $.92_{\pm.001}$ | $.93_{\pm.001}$ | $.93_{\pm.002}$ | $.93_{\pm.004}$ | $.91_{\pm.000}$ | $.88_{\pm.004}$ | $.80_{\pm.001}$ |
| | PLASMA-PF | $.90_{\pm.005}$ | $.93_{\pm.002}$ | $.93_{\pm.004}$ | $.89_{\pm.004}$ | $.84_{\pm.004}$ | $.84_{\pm.002}$ | $.77_{\pm.002}$ |
| | EBA | $.84_{\pm.006}$ | $.86_{\pm.003}$ | $.80_{\pm.005}$ | $.81_{\pm.003}$ | $.00_{\pm.000}$ | $.82_{\pm.003}$ | $.69_{\pm.006}$ |
| | COSINESIM | $.74_{\pm.007}$ | $.79_{\pm.004}$ | $.76_{\pm.003}$ | $.69_{\pm.001}$ | $.70_{\pm.002}$ | $.78_{\pm.005}$ | $.67_{\pm.003}$ |
| | FOLDSEEK | $.84_{\pm.007}$ | | | | | | |
| | TM-ALIGN | $.70_{\pm.002}$ | | | | | | |
| LMS | PLASMA | $.91_{\pm.007}$ | $.92_{\pm.001}$ | $.92_{\pm.002}$ | $.92_{\pm.005}$ | $.73_{\pm.013}$ | $.76_{\pm.005}$ | $.71_{\pm.007}$ |
| | PLASMA-PF | $.57_{\pm.003}$ | $.37_{\pm.006}$ | $.75_{\pm.006}$ | $.46_{\pm.009}$ | $.21_{\pm.002}$ | $.45_{\pm.001}$ | $.39_{\pm.009}$ |

Table 3: Comprehensive binding site detection results on test_inter dataset across seven protein representation models.

| Metrics | Methods | Binding Site | | | | | | |
|---|---|---|---|---|---|---|---|---|
| | | PROSTT5 | PROTT5 | ANKH | ESM2 | PROTSSN | TM-VEC | PROTBERT |
| ROC-AUC | PLASMA | $.99_{\pm.001}$ | $.99_{\pm.000}$ | $.99_{\pm.000}$ | $.99_{\pm.001}$ | $.99_{\pm.001}$ | $.96_{\pm.003}$ | $.98_{\pm.001}$ |
| | PLASMA-PF | $.99_{\pm.001}$ | $.99_{\pm.001}$ | $.99_{\pm.000}$ | $.96_{\pm.003}$ | $.97_{\pm.001}$ | $.92_{\pm.004}$ | $.90_{\pm.003}$ |
| | EBA | $.97_{\pm.001}$ | $.97_{\pm.001}$ | $.97_{\pm.001}$ | $.97_{\pm.002}$ | $.40_{\pm.005}$ | $.95_{\pm.000}$ | $.84_{\pm.006}$ |
| | COSINESIM | $.87_{\pm.005}$ | $.88_{\pm.004}$ | $.96_{\pm.002}$ | $.79_{\pm.009}$ | $.75_{\pm.008}$ | $.92_{\pm.006}$ | $.66_{\pm.008}$ |
| | FOLDSEEK | $.89_{\pm.001}$ | | | | | | |
| | TM-ALIGN | $.87_{\pm.003}$ | | | | | | |
| PR-AUC | PLASMA | $.99_{\pm.001}$ | $.99_{\pm.001}$ | $.99_{\pm.000}$ | $.99_{\pm.001}$ | $.99_{\pm.001}$ | $.97_{\pm.002}$ | $.98_{\pm.001}$ |
| | PLASMA-PF | $.99_{\pm.001}$ | $.99_{\pm.001}$ | $.99_{\pm.000}$ | $.97_{\pm.002}$ | $.98_{\pm.001}$ | $.93_{\pm.004}$ | $.93_{\pm.001}$ |
| | EBA | $.98_{\pm.000}$ | $.98_{\pm.001}$ | $.98_{\pm.001}$ | $.98_{\pm.001}$ | $.42_{\pm.004}$ | $.96_{\pm.001}$ | $.87_{\pm.003}$ |
| | COSINESIM | $.90_{\pm.005}$ | $.90_{\pm.003}$ | $.97_{\pm.002}$ | $.83_{\pm.007}$ | $.78_{\pm.005}$ | $.94_{\pm.004}$ | $.70_{\pm.006}$ |
| | FOLDSEEK | $.83_{\pm.002}$ | | | | | | |
| | TM-ALIGN | $.91_{\pm.002}$ | | | | | | |
| F1-MAX | PLASMA | $.98_{\pm.002}$ | $.98_{\pm.001}$ | $.98_{\pm.001}$ | $.98_{\pm.002}$ | $.97_{\pm.002}$ | $.95_{\pm.002}$ | $.94_{\pm.002}$ |
| | PLASMA-PF | $.96_{\pm.001}$ | $.97_{\pm.001}$ | $.97_{\pm.001}$ | $.92_{\pm.003}$ | $.94_{\pm.001}$ | $.91_{\pm.005}$ | $.83_{\pm.002}$ |
| | EBA | $.94_{\pm.001}$ | $.94_{\pm.001}$ | $.94_{\pm.001}$ | $.93_{\pm.002}$ | $.00_{\pm.000}$ | $.93_{\pm.001}$ | $.78_{\pm.007}$ |
| | COSINESIM | $.80_{\pm.008}$ | $.80_{\pm.005}$ | $.91_{\pm.005}$ | $.73_{\pm.006}$ | $.69_{\pm.006}$ | $.86_{\pm.007}$ | $.67_{\pm.001}$ |
| | FOLDSEEK | $.94_{\pm.001}$ | | | | | | |
| | TM-ALIGN | $.84_{\pm.005}$ | | | | | | |
| LMS | PLASMA | $.93_{\pm.002}$ | $.93_{\pm.003}$ | $.93_{\pm.004}$ | $.93_{\pm.003}$ | $.85_{\pm.006}$ | $.86_{\pm.002}$ | $.84_{\pm.003}$ |
| | PLASMA-PF | $.80_{\pm.008}$ | $.59_{\pm.008}$ | $.85_{\pm.005}$ | $.57_{\pm.009}$ | $.36_{\pm.005}$ | $.60_{\pm.008}$ | $.44_{\pm.004}$ |

Table 4: Comprehensive active site detection results on test_inter dataset across seven protein representation models.

| Metrics | Methods | Active Site | | | | | | |
|---|---|---|---|---|---|---|---|---|
| | | PROSTT5 | PROTT5 | ANKH | ESM2 | PROTSSN | TM-VEC | PROTBERT |
| ROC-AUC | PLASMA | $.99_{\pm.001}$ | $.99_{\pm.001}$ | $.99_{\pm.001}$ | $.99_{\pm.001}$ | $.99_{\pm.002}$ | $.99_{\pm.003}$ | $.99_{\pm.004}$ |
| | PLASMA-PF | $.99_{\pm.002}$ | $.99_{\pm.003}$ | $.99_{\pm.003}$ | $.96_{\pm.002}$ | $.98_{\pm.002}$ | $.98_{\pm.003}$ | $.94_{\pm.006}$ |
| | EBA | $.99_{\pm.003}$ | $.99_{\pm.003}$ | $.99_{\pm.003}$ | $.99_{\pm.003}$ | $.43_{\pm.005}$ | $.99_{\pm.003}$ | $.90_{\pm.005}$ |
| | COSINESIM | $.91_{\pm.004}$ | $.91_{\pm.003}$ | $.97_{\pm.002}$ | $.78_{\pm.009}$ | $.74_{\pm.006}$ | $.98_{\pm.002}$ | $.66_{\pm.003}$ |
| | FOLDSEEK | $.89_{\pm.001}$ | | | | | | |
| | TM-ALIGN | $.94_{\pm.003}$ | | | | | | |
| PR-AUC | PLASMA | $.99_{\pm.000}$ | $.99_{\pm.001}$ | $.99_{\pm.001}$ | $.99_{\pm.000}$ | $.99_{\pm.001}$ | $.99_{\pm.002}$ | $.99_{\pm.003}$ |
| | PLASMA-PF | $.99_{\pm.001}$ | $.99_{\pm.002}$ | $.99_{\pm.002}$ | $.97_{\pm.001}$ | $.99_{\pm.001}$ | $.98_{\pm.003}$ | $.95_{\pm.004}$ |
| | EBA | $.99_{\pm.003}$ | $.99_{\pm.002}$ | $.99_{\pm.002}$ | $.99_{\pm.002}$ | $.43_{\pm.006}$ | $.99_{\pm.003}$ | $.92_{\pm.003}$ |
| | COSINESIM | $.93_{\pm.002}$ | $.92_{\pm.001}$ | $.98_{\pm.001}$ | $.83_{\pm.004}$ | $.79_{\pm.002}$ | $.98_{\pm.001}$ | $.70_{\pm.007}$ |
| | FOLDSEEK | $.83_{\pm.006}$ | | | | | | |
| | TM-ALIGN | $.96_{\pm.001}$ | | | | | | |
| F1-MAX | PLASMA | $.98_{\pm.003}$ | $.98_{\pm.003}$ | $.99_{\pm.003}$ | $.98_{\pm.001}$ | $.99_{\pm.002}$ | $.98_{\pm.003}$ | $.96_{\pm.004}$ |
| | PLASMA-PF | $.98_{\pm.003}$ | $.98_{\pm.004}$ | $.98_{\pm.003}$ | $.93_{\pm.004}$ | $.96_{\pm.003}$ | $.97_{\pm.004}$ | $.89_{\pm.005}$ |
| | EBA | $.97_{\pm.005}$ | $.98_{\pm.004}$ | $.97_{\pm.003}$ | $.97_{\pm.003}$ | $.00_{\pm.000}$ | $.97_{\pm.005}$ | $.84_{\pm.004}$ |
| | COSINESIM | $.85_{\pm.004}$ | $.83_{\pm.002}$ | $.94_{\pm.003}$ | $.71_{\pm.006}$ | $.68_{\pm.001}$ | $.93_{\pm.002}$ | $.67_{\pm.006}$ |
| | FOLDSEEK | $.97_{\pm.005}$ | | | | | | |
| | TM-ALIGN | $.90_{\pm.003}$ | | | | | | |
| LMS | PLASMA | $.97_{\pm.004}$ | $.97_{\pm.004}$ | $.97_{\pm.003}$ | $.97_{\pm.004}$ | $.89_{\pm.016}$ | $.93_{\pm.006}$ | $.89_{\pm.008}$ |
| | PLASMA-PF | $.91_{\pm.010}$ | $.68_{\pm.003}$ | $.95_{\pm.006}$ | $.63_{\pm.013}$ | $.43_{\pm.007}$ | $.77_{\pm.011}$ | $.52_{\pm.004}$ |

Table 5: Model performance on `test_inter` (mean ± std over three independent seeds). Colors indicate relative performance versus TM-Align, percentage values report the associated specific relative performance difference.

| Metrics | Methods | Motif | | | Binding Site | | | Active Site | | |
|---|---|---|---|---|---|---|---|---|---|---|
| | | ANKH | ESM2 | PROTSSN | ANKH | ESM2 | PROTSSN | ANKH | ESM2 | PROTSSN |
| ROC-AUC | PLASMA | $.95^{\uparrow 21.8\%}_{\pm.002}$ | $.96^{\uparrow 23.1\%}_{\pm.002}$ | $.96^{\uparrow 23.1\%}_{\pm.001}$ | $.99^{\uparrow 13.8\%}_{\pm.000}$ | $.99^{\uparrow 13.8\%}_{\pm.001}$ | $.99^{\uparrow 13.8\%}_{\pm.001}$ | $.99^{\uparrow 5.3\%}_{\pm.001}$ | $.99^{\uparrow 5.3\%}_{\pm.001}$ | $.99^{\uparrow 5.3\%}_{\pm.002}$ |
| | PLASMA-PF | $.95^{\uparrow 21.8\%}_{\pm.003}$ | $.93^{\uparrow 19.2\%}_{\pm.004}$ | $.91^{\uparrow 16.7\%}_{\pm.003}$ | $.99^{\uparrow 13.8\%}_{\pm.000}$ | $.96^{\uparrow 10.3\%}_{\pm.003}$ | $.97^{\uparrow 11.5\%}_{\pm.001}$ | $.99^{\uparrow 5.3\%}_{\pm.003}$ | $.96^{\uparrow 2.1\%}_{\pm.002}$ | $.98^{\uparrow 4.3\%}_{\pm.002}$ |
| | EBA | $.87^{\uparrow 11.5\%}_{\pm.005}$ | $.88^{\uparrow 12.8\%}_{\pm.003}$ | $.44^{\downarrow 43.6\%}_{\pm.002}$ | $.97^{\uparrow 11.5\%}_{\pm.001}$ | $.97^{\uparrow 11.5\%}_{\pm.002}$ | $.40^{\downarrow 54.0\%}_{\pm.005}$ | $.99^{\uparrow 5.3\%}_{\pm.003}$ | $.99^{\uparrow 5.3\%}_{\pm.003}$ | $.43^{\downarrow 54.3\%}_{\pm.005}$ |
| | Backbone | $.84^{\uparrow 7.7\%}_{\pm.006}$ | $.73^{\downarrow 6.4\%}_{\pm.009}$ | $.75^{\downarrow 3.8\%}_{\pm.006}$ | $.96^{\uparrow 10.3\%}_{\pm.002}$ | $.79^{\downarrow 9.2\%}_{\pm.009}$ | $.75^{\downarrow 13.8\%}_{\pm.008}$ | $.97^{\uparrow 3.2\%}_{\pm.002}$ | $.78^{\downarrow 17.0\%}_{\pm.009}$ | $.74^{\downarrow 21.3\%}_{\pm.006}$ |
| | Foldseek | | $.83^{\uparrow 6.4\%}_{\pm.007}$ | | | $.89^{\uparrow 2.3\%}_{\pm.001}$ | | | $.89^{\downarrow 5.3\%}_{\pm.001}$ | |
| | TM-Align | | $.78_{\pm.003}$ | | | $.87_{\pm.003}$ | | | $.94_{\pm.003}$ | |
| PR-AUC | PLASMA | $.95^{\uparrow 14.5\%}_{\pm.002}$ | $.97^{\uparrow 16.9\%}_{\pm.001}$ | $.96^{\uparrow 15.7\%}_{\pm.001}$ | $.99^{\uparrow 8.8\%}_{\pm.000}$ | $.99^{\uparrow 8.8\%}_{\pm.001}$ | $.99^{\uparrow 8.8\%}_{\pm.001}$ | $.99^{\uparrow 3.1\%}_{\pm.001}$ | $.99^{\uparrow 3.1\%}_{\pm.000}$ | $.99^{\uparrow 3.1\%}_{\pm.001}$ |
| | PLASMA-PF | $.95^{\uparrow 14.5\%}_{\pm.003}$ | $.94^{\uparrow 13.3\%}_{\pm.002}$ | $.92^{\uparrow 10.8\%}_{\pm.001}$ | $.99^{\uparrow 8.8\%}_{\pm.000}$ | $.97^{\uparrow 6.6\%}_{\pm.002}$ | $.98^{\uparrow 7.7\%}_{\pm.001}$ | $.99^{\uparrow 3.1\%}_{\pm.002}$ | $.97^{\uparrow 1.0\%}_{\pm.001}$ | $.99^{\uparrow 3.1\%}_{\pm.001}$ |
| | EBA | $.90^{\uparrow 8.4\%}_{\pm.004}$ | $.90^{\uparrow 8.4\%}_{\pm.004}$ | $.45^{\downarrow 45.8\%}_{\pm.004}$ | $.98^{\uparrow 7.7\%}_{\pm.001}$ | $.98^{\uparrow 7.7\%}_{\pm.001}$ | $.42^{\downarrow 53.8\%}_{\pm.004}$ | $.99^{\uparrow 3.1\%}_{\pm.002}$ | $.99^{\uparrow 3.1\%}_{\pm.002}$ | $.43^{\downarrow 55.2\%}_{\pm.006}$ |
| | Backbone | $.86^{\uparrow 3.6\%}_{\pm.005}$ | $.76^{\downarrow 8.4\%}_{\pm.008}$ | $.78^{\downarrow 6.0\%}_{\pm.006}$ | $.97^{\uparrow 6.6\%}_{\pm.002}$ | $.83^{\downarrow 8.8\%}_{\pm.007}$ | $.78^{\downarrow 14.3\%}_{\pm.005}$ | $.98^{\uparrow 2.1\%}_{\pm.001}$ | $.83^{\downarrow 13.5\%}_{\pm.004}$ | $.79^{\downarrow 17.7\%}_{\pm.002}$ |
| | Foldseek | | $.78^{\downarrow 6.0\%}_{\pm.008}$ | | | $.83^{\downarrow 8.8\%}_{\pm.002}$ | | | $.83^{\downarrow 13.5\%}_{\pm.006}$ | |
| | TM-Align | | $.83_{\pm.004}$ | | | $.91_{\pm.002}$ | | | $.96_{\pm.001}$ | |
| F1-MAX | PLASMA | $.93^{\uparrow 32.9\%}_{\pm.002}$ | $.93^{\uparrow 32.9\%}_{\pm.004}$ | $.91^{\uparrow 30.0\%}_{\pm.000}$ | $.98^{\uparrow 16.7\%}_{\pm.001}$ | $.98^{\uparrow 16.7\%}_{\pm.002}$ | $.97^{\uparrow 15.5\%}_{\pm.002}$ | $.99^{\uparrow 10.0\%}_{\pm.003}$ | $.98^{\uparrow 8.9\%}_{\pm.001}$ | $.99^{\uparrow 10.0\%}_{\pm.002}$ |
| | PLASMA-PF | $.93^{\uparrow 32.9\%}_{\pm.004}$ | $.89^{\uparrow 27.1\%}_{\pm.004}$ | $.84^{\uparrow 20.0\%}_{\pm.004}$ | $.97^{\uparrow 15.5\%}_{\pm.001}$ | $.92^{\uparrow 9.5\%}_{\pm.003}$ | $.94^{\uparrow 11.9\%}_{\pm.001}$ | $.98^{\uparrow 8.9\%}_{\pm.003}$ | $.93^{\uparrow 3.3\%}_{\pm.004}$ | $.96^{\uparrow 6.7\%}_{\pm.003}$ |
| | EBA | $.80^{\uparrow 14.3\%}_{\pm.005}$ | $.81^{\uparrow 15.7\%}_{\pm.003}$ | $.00^{\downarrow 100.0\%}_{\pm.000}$ | $.94^{\uparrow 11.9\%}_{\pm.001}$ | $.93^{\uparrow 10.7\%}_{\pm.002}$ | $.00^{\downarrow 100.0\%}_{\pm.000}$ | $.97^{\uparrow 7.8\%}_{\pm.003}$ | $.97^{\uparrow 7.8\%}_{\pm.003}$ | $.00^{\downarrow 100.0\%}_{\pm.000}$ |
| | Backbone | $.76^{\uparrow 8.6\%}_{\pm.003}$ | $.69^{\downarrow 1.4\%}_{\pm.001}$ | $.70^{\downarrow 0.0\%}_{\pm.002}$ | $.91^{\uparrow 8.3\%}_{\pm.005}$ | $.73^{\downarrow 13.1\%}_{\pm.006}$ | $.69^{\downarrow 17.9\%}_{\pm.006}$ | $.94^{\uparrow 4.4\%}_{\pm.003}$ | $.71^{\downarrow 21.1\%}_{\pm.006}$ | $.68^{\downarrow 24.4\%}_{\pm.001}$ |
| | Foldseek | | $.84^{\uparrow 20.0\%}_{\pm.007}$ | | | $.94^{\uparrow 11.9\%}_{\pm.001}$ | | | $.97^{\uparrow 7.8\%}_{\pm.005}$ | |
| | TM-Align | | $.70_{\pm.002}$ | | | $.84_{\pm.005}$ | | | $.90_{\pm.003}$ | |

## F  FULL EXTRAPOLATION PERFORMANCE COMPARISON

This section evaluates PLASMA's generalization capability on the `test_extra` dataset, which contains substructures never encountered during training. These experiments are crucial for assessing applicability in detecting unknown substructures. The results demonstrate that PLASMA maintains superior performance even when confronted with completely unseen substructures, achieving the highest scores for both detecting the existence of similar substructures and accurately localizing their positions for most of the cases. This robust extrapolation performance further validates that our optimal transport framework captures fundamental protein substructure similarity patterns that transcend specific training examples, making it highly valuable for analyzing newly discovered proteins and understudied organisms.

Table 6: Comprehensive motif detection results on `test_extra` dataset across seven protein representation models.

| Metrics | Methods | Motif | | | | | | |
|---|---|---|---|---|---|---|---|---|
| | | PROSTT5 | PROTT5 | ANKH | ESM2 | PROTSSN | TM-VEC | PROTBERT |
| ROC-AUC | PLASMA | $.97_{\pm.015}$ | $.98_{\pm.012}$ | $.98_{\pm.008}$ | $.97_{\pm.013}$ | $.96_{\pm.016}$ | $.95_{\pm.023}$ | $.79_{\pm.022}$ |
| | PLASMA-PF | $.97_{\pm.014}$ | $.98_{\pm.010}$ | $.98_{\pm.009}$ | $.93_{\pm.004}$ | $.90_{\pm.005}$ | $.88_{\pm.039}$ | $.82_{\pm.016}$ |
| | EBA | $.94_{\pm.017}$ | $.95_{\pm.009}$ | $.90_{\pm.033}$ | $.92_{\pm.021}$ | $.32_{\pm.043}$ | $.94_{\pm.016}$ | $.76_{\pm.025}$ |
| | COSINESIM | $.84_{\pm.029}$ | $.89_{\pm.024}$ | $.85_{\pm.019}$ | $.74_{\pm.033}$ | $.79_{\pm.018}$ | $.83_{\pm.050}$ | $.62_{\pm.080}$ |
| | FOLDSEEK | $.89_{\pm.033}$ | | | | | | |
| | TM-ALIGN | $.81_{\pm.014}$ | | | | | | |
| PR-AUC | PLASMA | $.97_{\pm.017}$ | $.97_{\pm.018}$ | $.98_{\pm.011}$ | $.97_{\pm.014}$ | $.96_{\pm.017}$ | $.95_{\pm.025}$ | $.84_{\pm.014}$ |
| | PLASMA-PF | $.97_{\pm.015}$ | $.97_{\pm.016}$ | $.98_{\pm.010}$ | $.95_{\pm.005}$ | $.92_{\pm.007}$ | $.88_{\pm.040}$ | $.86_{\pm.012}$ |
| | EBA | $.94_{\pm.018}$ | $.96_{\pm.010}$ | $.91_{\pm.035}$ | $.93_{\pm.019}$ | $.38_{\pm.014}$ | $.95_{\pm.014}$ | $.80_{\pm.029}$ |
| | COSINESIM | $.85_{\pm.028}$ | $.90_{\pm.017}$ | $.86_{\pm.023}$ | $.77_{\pm.041}$ | $.82_{\pm.027}$ | $.86_{\pm.036}$ | $.66_{\pm.090}$ |
| | FOLDSEEK | $.84_{\pm.031}$ | | | | | | |
| | TM-ALIGN | $.86_{\pm.020}$ | | | | | | |
| F1-MAX | PLASMA | $.95_{\pm.011}$ | $.96_{\pm.010}$ | $.97_{\pm.009}$ | $.95_{\pm.018}$ | $.92_{\pm.022}$ | $.92_{\pm.022}$ | $.72_{\pm.017}$ |
| | PLASMA-PF | $.93_{\pm.019}$ | $.96_{\pm.006}$ | $.96_{\pm.013}$ | $.90_{\pm.006}$ | $.84_{\pm.008}$ | $.85_{\pm.041}$ | $.75_{\pm.017}$ |
| | EBA | $.88_{\pm.027}$ | $.90_{\pm.014}$ | $.86_{\pm.035}$ | $.87_{\pm.024}$ | $.00_{\pm.000}$ | $.87_{\pm.019}$ | $.73_{\pm.008}$ |
| | COSINESIM | $.77_{\pm.020}$ | $.82_{\pm.025}$ | $.79_{\pm.008}$ | $.70_{\pm.014}$ | $.73_{\pm.013}$ | $.77_{\pm.040}$ | $.68_{\pm.015}$ |
| | FOLDSEEK | $.91_{\pm.046}$ | | | | | | |
| | TM-ALIGN | $.76_{\pm.015}$ | | | | | | |
| LMS | PLASMA | $.72_{\pm.022}$ | $.70_{\pm.022}$ | $.75_{\pm.045}$ | $.69_{\pm.019}$ | $.52_{\pm.046}$ | $.60_{\pm.021}$ | $.48_{\pm.052}$ |
| | PLASMA-PF | $.62_{\pm.042}$ | $.38_{\pm.057}$ | $.78_{\pm.055}$ | $.48_{\pm.074}$ | $.23_{\pm.021}$ | $.44_{\pm.026}$ | $.41_{\pm.066}$ |

Table 7: Comprehensive binding site detection results on `test_extra` dataset across seven protein representation models.

| Metrics | Methods | Binding Site | | | | | | |
|---|---|---|---|---|---|---|---|---|
| | | PROSTT5 | PROTT5 | ANKH | ESM2 | PROTSSN | TM-VEC | PROTBERT |
| ROC-AUC | PLASMA | $.98_{\pm.009}$ | $.98_{\pm.009}$ | $.99_{\pm.008}$ | $.98_{\pm.013}$ | $.98_{\pm.014}$ | $.98_{\pm.008}$ | $.92_{\pm.019}$ |
| | PLASMA-PF | $.98_{\pm.008}$ | $.98_{\pm.010}$ | $.99_{\pm.006}$ | $.92_{\pm.052}$ | $.96_{\pm.012}$ | $.95_{\pm.019}$ | $.87_{\pm.032}$ |
| | EBA | $.98_{\pm.013}$ | $.99_{\pm.009}$ | $.99_{\pm.007}$ | $.97_{\pm.021}$ | $.30_{\pm.060}$ | $.98_{\pm.014}$ | $.83_{\pm.072}$ |
| | COSINESIM | $.89_{\pm.038}$ | $.86_{\pm.059}$ | $.98_{\pm.010}$ | $.72_{\pm.060}$ | $.70_{\pm.070}$ | $.94_{\pm.021}$ | $.56_{\pm.029}$ |
| | FOLDSEEK | $.90_{\pm.013}$ | | | | | | |
| | TM-ALIGN | $.91_{\pm.040}$ | | | | | | |
| PR-AUC | PLASMA | $.98_{\pm.011}$ | $.98_{\pm.010}$ | $.98_{\pm.011}$ | $.97_{\pm.019}$ | $.97_{\pm.019}$ | $.97_{\pm.012}$ | $.90_{\pm.043}$ |
| | PLASMA-PF | $.98_{\pm.013}$ | $.98_{\pm.014}$ | $.98_{\pm.012}$ | $.90_{\pm.079}$ | $.95_{\pm.026}$ | $.93_{\pm.022}$ | $.84_{\pm.078}$ |
| | EBA | $.98_{\pm.014}$ | $.98_{\pm.014}$ | $.98_{\pm.012}$ | $.96_{\pm.035}$ | $.28_{\pm.063}$ | $.97_{\pm.020}$ | $.79_{\pm.115}$ |
| | COSINESIM | $.86_{\pm.076}$ | $.82_{\pm.099}$ | $.96_{\pm.023}$ | $.67_{\pm.093}$ | $.65_{\pm.118}$ | $.93_{\pm.029}$ | $.49_{\pm.076}$ |
| | FOLDSEEK | $.76_{\pm.065}$ | | | | | | |
| | TM-ALIGN | $.89_{\pm.064}$ | | | | | | |
| F1-MAX | PLASMA | $.97_{\pm.016}$ | $.97_{\pm.011}$ | $.96_{\pm.022}$ | $.95_{\pm.030}$ | $.93_{\pm.026}$ | $.96_{\pm.014}$ | $.83_{\pm.046}$ |
| | PLASMA-PF | $.96_{\pm.023}$ | $.97_{\pm.017}$ | $.96_{\pm.027}$ | $.85_{\pm.082}$ | $.90_{\pm.031}$ | $.93_{\pm.018}$ | $.76_{\pm.073}$ |
| | EBA | $.96_{\pm.021}$ | $.96_{\pm.026}$ | $.97_{\pm.021}$ | $.93_{\pm.049}$ | $.00_{\pm.000}$ | $.94_{\pm.034}$ | $.73_{\pm.108}$ |
| | COSINESIM | $.78_{\pm.081}$ | $.76_{\pm.089}$ | $.91_{\pm.034}$ | $.62_{\pm.087}$ | $.60_{\pm.107}$ | $.86_{\pm.046}$ | $.55_{\pm.092}$ |
| | FOLDSEEK | $.97_{\pm.014}$ | | | | | | |
| | TM-ALIGN | $.87_{\pm.063}$ | | | | | | |
| LMS | PLASMA | $.84_{\pm.050}$ | $.83_{\pm.051}$ | $.82_{\pm.062}$ | $.77_{\pm.105}$ | $.65_{\pm.088}$ | $.75_{\pm.071}$ | $.56_{\pm.075}$ |
| | PLASMA-PF | $.79_{\pm.098}$ | $.55_{\pm.079}$ | $.85_{\pm.058}$ | $.49_{\pm.082}$ | $.36_{\pm.055}$ | $.65_{\pm.070}$ | $.43_{\pm.038}$ |

Table 8: Comprehensive active site detection results on `test_extra` dataset across seven protein representation models.

| Metrics | Methods | Active Site | | | | | | |
|---|---|---|---|---|---|---|---|---|
| | | PROSTT5 | PROTT5 | ANKH | ESM2 | PROTSSN | TM-VEC | PROTBERT |
| ROC-AUC | PLASMA | $.98_{\pm.011}$ | $.98_{\pm.010}$ | $.98_{\pm.012}$ | $.98_{\pm.010}$ | $.97_{\pm.011}$ | $.97_{\pm.013}$ | $.95_{\pm.026}$ |
| | PLASMA-PF | $.98_{\pm.010}$ | $.98_{\pm.011}$ | $.97_{\pm.015}$ | $.96_{\pm.006}$ | $.97_{\pm.008}$ | $.97_{\pm.014}$ | $.93_{\pm.024}$ |
| | EBA | $.98_{\pm.012}$ | $.98_{\pm.012}$ | $.97_{\pm.013}$ | $.97_{\pm.012}$ | $.43_{\pm.066}$ | $.97_{\pm.013}$ | $.91_{\pm.027}$ |
| | COSINESIM | $.87_{\pm.032}$ | $.91_{\pm.011}$ | $.96_{\pm.012}$ | $.79_{\pm.068}$ | $.76_{\pm.033}$ | $.96_{\pm.013}$ | $.71_{\pm.012}$ |
| | FOLDSEEK | $.87_{\pm.022}$ | | | | | | |
| | TM-ALIGN | $.93_{\pm.009}$ | | | | | | |
| PR-AUC | PLASMA | $.97_{\pm.014}$ | $.98_{\pm.010}$ | $.97_{\pm.014}$ | $.98_{\pm.011}$ | $.97_{\pm.012}$ | $.97_{\pm.016}$ | $.96_{\pm.019}$ |
| | PLASMA-PF | $.98_{\pm.013}$ | $.98_{\pm.011}$ | $.97_{\pm.015}$ | $.96_{\pm.006}$ | $.97_{\pm.009}$ | $.96_{\pm.017}$ | $.95_{\pm.017}$ |
| | EBA | $.97_{\pm.013}$ | $.97_{\pm.014}$ | $.97_{\pm.012}$ | $.97_{\pm.012}$ | $.43_{\pm.032}$ | $.97_{\pm.014}$ | $.93_{\pm.019}$ |
| | COSINESIM | $.90_{\pm.031}$ | $.92_{\pm.017}$ | $.96_{\pm.016}$ | $.84_{\pm.059}$ | $.80_{\pm.038}$ | $.96_{\pm.015}$ | $.75_{\pm.010}$ |
| | FOLDSEEK | $.81_{\pm.026}$ | | | | | | |
| | TM-ALIGN | $.94_{\pm.012}$ | | | | | | |
| F1-MAX | PLASMA | $.97_{\pm.012}$ | $.98_{\pm.013}$ | $.98_{\pm.013}$ | $.97_{\pm.011}$ | $.97_{\pm.011}$ | $.97_{\pm.015}$ | $.92_{\pm.036}$ |
| | PLASMA-PF | $.97_{\pm.015}$ | $.97_{\pm.020}$ | $.97_{\pm.018}$ | $.94_{\pm.016}$ | $.95_{\pm.012}$ | $.96_{\pm.011}$ | $.89_{\pm.032}$ |
| | EBA | $.97_{\pm.014}$ | $.97_{\pm.013}$ | $.97_{\pm.013}$ | $.97_{\pm.008}$ | $.00_{\pm.000}$ | $.97_{\pm.020}$ | $.87_{\pm.026}$ |
| | COSINESIM | $.83_{\pm.033}$ | $.84_{\pm.013}$ | $.92_{\pm.020}$ | $.75_{\pm.044}$ | $.71_{\pm.018}$ | $.92_{\pm.010}$ | $.68_{\pm.008}$ |
| | FOLDSEEK | $.96_{\pm.015}$ | | | | | | |
| | TM-ALIGN | $.90_{\pm.014}$ | | | | | | |
| LMS | PLASMA | $.89_{\pm.044}$ | $.83_{\pm.030}$ | $.90_{\pm.034}$ | $.87_{\pm.038}$ | $.67_{\pm.044}$ | $.84_{\pm.053}$ | $.60_{\pm.024}$ |
| | PLASMA-PF | $.90_{\pm.043}$ | $.70_{\pm.014}$ | $.94_{\pm.029}$ | $.68_{\pm.067}$ | $.43_{\pm.032}$ | $.78_{\pm.048}$ | $.50_{\pm.021}$ |

Table 9: Model performance on `test_extra` (mean ± std over three independent seeds). Colors indicate relative performance versus TM-Align, percentage values report the associated specific relative performance difference.

| Metrics | Methods | Motif | | | Binding Site | | | Active Site | | |
|---|---|---|---|---|---|---|---|---|---|---|
| | | ANKH | ESM2 | PROTSSN | ANKH | ESM2 | PROTSSN | ANKH | ESM2 | PROTSSN |
| ROC-AUC | PLASMA | $.98^{\uparrow 21.0\%}_{\pm.008}$ | $.97^{\uparrow 19.8\%}_{\pm.013}$ | $.96^{\uparrow 18.5\%}_{\pm.016}$ | $.99^{\uparrow 8.8\%}_{\pm.008}$ | $.98^{\uparrow 7.7\%}_{\pm.013}$ | $.98^{\uparrow 7.7\%}_{\pm.014}$ | $.98^{\uparrow 5.4\%}_{\pm.012}$ | $.98^{\uparrow 5.4\%}_{\pm.010}$ | $.97^{\uparrow 4.3\%}_{\pm.011}$ |
| | PLASMA-PF | $.98^{\uparrow 21.0\%}_{\pm.009}$ | $.93^{\uparrow 14.8\%}_{\pm.004}$ | $.90^{\uparrow 11.1\%}_{\pm.005}$ | $.99^{\uparrow 8.8\%}_{\pm.006}$ | $.92^{\uparrow 1.1\%}_{\pm.052}$ | $.96^{\uparrow 5.5\%}_{\pm.012}$ | $.97^{\uparrow 4.3\%}_{\pm.015}$ | $.96^{\uparrow 3.2\%}_{\pm.006}$ | $.97^{\uparrow 4.3\%}_{\pm.008}$ |
| | EBA | $.90^{\uparrow 11.1\%}_{\pm.033}$ | $.92^{\uparrow 13.6\%}_{\pm.021}$ | $.32^{\downarrow 60.5\%}_{\pm.043}$ | $.99^{\uparrow 8.8\%}_{\pm.007}$ | $.97^{\uparrow 6.6\%}_{\pm.021}$ | $.30^{\downarrow 67.0\%}_{\pm.060}$ | $.97^{\uparrow 4.3\%}_{\pm.013}$ | $.97^{\uparrow 4.3\%}_{\pm.012}$ | $.43^{\downarrow 53.8\%}_{\pm.066}$ |
| | Backbone | $.85^{\uparrow 4.9\%}_{\pm.019}$ | $.74^{\downarrow 8.6\%}_{\pm.033}$ | $.79^{\downarrow 2.5\%}_{\pm.018}$ | $.98^{\uparrow 7.7\%}_{\pm.010}$ | $.72^{\downarrow 20.9\%}_{\pm.060}$ | $.70^{\downarrow 23.1\%}_{\pm.070}$ | $.96^{\uparrow 3.2\%}_{\pm.012}$ | $.79^{\downarrow 15.1\%}_{\pm.068}$ | $.76^{\downarrow 18.3\%}_{\pm.033}$ |
| | Foldseek | | $.89^{\uparrow 9.9\%}_{\pm.033}$ | | | $.90^{\uparrow 1.1\%}_{\pm.013}$ | | | $.87^{\downarrow 6.5\%}_{\pm.022}$ | |
| | TM-Align | | $.81_{\pm.014}$ | | | $.91_{\pm.040}$ | | | $.93_{\pm.009}$ | |
| PR-AUC | PLASMA | $.98^{\uparrow 14.0\%}_{\pm.011}$ | $.97^{\uparrow 12.8\%}_{\pm.014}$ | $.96^{\uparrow 11.6\%}_{\pm.017}$ | $.98^{\uparrow 10.1\%}_{\pm.011}$ | $.97^{\uparrow 9.0\%}_{\pm.019}$ | $.97^{\uparrow 9.0\%}_{\pm.019}$ | $.97^{\uparrow 3.2\%}_{\pm.014}$ | $.98^{\uparrow 4.3\%}_{\pm.011}$ | $.97^{\uparrow 3.2\%}_{\pm.012}$ |
| | PLASMA-PF | $.98^{\uparrow 14.0\%}_{\pm.010}$ | $.95^{\uparrow 10.5\%}_{\pm.005}$ | $.92^{\uparrow 7.0\%}_{\pm.007}$ | $.98^{\uparrow 10.1\%}_{\pm.012}$ | $.90^{\uparrow 1.1\%}_{\pm.079}$ | $.95^{\uparrow 6.7\%}_{\pm.026}$ | $.97^{\uparrow 3.2\%}_{\pm.015}$ | $.96^{\uparrow 2.1\%}_{\pm.006}$ | $.97^{\uparrow 3.2\%}_{\pm.009}$ |
| | EBA | $.91^{\uparrow 5.8\%}_{\pm.035}$ | $.93^{\uparrow 8.1\%}_{\pm.019}$ | $.38^{\downarrow 55.8\%}_{\pm.014}$ | $.98^{\uparrow 10.1\%}_{\pm.012}$ | $.96^{\uparrow 7.9\%}_{\pm.035}$ | $.28^{\downarrow 68.5\%}_{\pm.063}$ | $.97^{\uparrow 3.2\%}_{\pm.012}$ | $.97^{\uparrow 3.2\%}_{\pm.012}$ | $.43^{\downarrow 54.3\%}_{\pm.032}$ |
| | Backbone | $.86^{\downarrow 0.0\%}_{\pm.023}$ | $.77^{\downarrow 10.5\%}_{\pm.041}$ | $.82^{\downarrow 4.7\%}_{\pm.027}$ | $.96^{\uparrow 7.9\%}_{\pm.023}$ | $.67^{\downarrow 24.7\%}_{\pm.093}$ | $.65^{\downarrow 27.0\%}_{\pm.118}$ | $.96^{\uparrow 2.1\%}_{\pm.016}$ | $.84^{\downarrow 10.6\%}_{\pm.059}$ | $.80^{\downarrow 14.9\%}_{\pm.038}$ |
| | Foldseek | | $.84^{\downarrow 2.3\%}_{\pm.031}$ | | | $.76^{\downarrow 14.6\%}_{\pm.065}$ | | | $.81^{\downarrow 13.8\%}_{\pm.026}$ | |
| | TM-Align | | $.86_{\pm.020}$ | | | $.89_{\pm.064}$ | | | $.94_{\pm.012}$ | |
| F1-MAX | PLASMA | $.97^{\uparrow 27.6\%}_{\pm.009}$ | $.95^{\uparrow 25.0\%}_{\pm.018}$ | $.92^{\uparrow 21.1\%}_{\pm.022}$ | $.96^{\uparrow 10.3\%}_{\pm.022}$ | $.95^{\uparrow 9.2\%}_{\pm.030}$ | $.93^{\uparrow 6.9\%}_{\pm.026}$ | $.98^{\uparrow 8.9\%}_{\pm.013}$ | $.97^{\uparrow 7.8\%}_{\pm.011}$ | $.97^{\uparrow 7.8\%}_{\pm.011}$ |
| | PLASMA-PF | $.96^{\uparrow 26.3\%}_{\pm.013}$ | $.90^{\uparrow 18.4\%}_{\pm.006}$ | $.84^{\uparrow 10.5\%}_{\pm.008}$ | $.96^{\uparrow 10.3\%}_{\pm.027}$ | $.85^{\downarrow 2.3\%}_{\pm.082}$ | $.90^{\uparrow 3.4\%}_{\pm.031}$ | $.97^{\uparrow 7.8\%}_{\pm.018}$ | $.94^{\uparrow 4.4\%}_{\pm.016}$ | $.95^{\uparrow 5.6\%}_{\pm.012}$ |
| | EBA | $.86^{\uparrow 13.2\%}_{\pm.035}$ | $.87^{\uparrow 14.5\%}_{\pm.024}$ | $.00^{\downarrow 100.0\%}_{\pm.000}$ | $.97^{\uparrow 11.5\%}_{\pm.021}$ | $.93^{\uparrow 6.9\%}_{\pm.049}$ | $.00^{\downarrow 100.0\%}_{\pm.000}$ | $.97^{\uparrow 7.8\%}_{\pm.013}$ | $.97^{\uparrow 7.8\%}_{\pm.008}$ | $.00^{\downarrow 100.0\%}_{\pm.000}$ |
| | Backbone | $.79^{\uparrow 3.9\%}_{\pm.008}$ | $.70^{\downarrow 7.9\%}_{\pm.014}$ | $.73^{\downarrow 3.9\%}_{\pm.013}$ | $.91^{\uparrow 4.6\%}_{\pm.034}$ | $.62^{\downarrow 28.7\%}_{\pm.087}$ | $.60^{\downarrow 31.0\%}_{\pm.107}$ | $.92^{\uparrow 2.2\%}_{\pm.020}$ | $.75^{\downarrow 16.7\%}_{\pm.044}$ | $.71^{\downarrow 21.1\%}_{\pm.018}$ |
| | Foldseek | | $.91^{\uparrow 19.7\%}_{\pm.046}$ | | | $.97^{\uparrow 11.5\%}_{\pm.014}$ | | | $.96^{\uparrow 6.7\%}_{\pm.015}$ | |
| | TM-Align | | $.76_{\pm.015}$ | | | $.87_{\pm.063}$ | | | $.90_{\pm.014}$ | |

## G  ABLATION STUDY

This section analyzes the contribution of the two plan-assessor components: the local-motif loss (LML) and the weight-correction term (WC) derived from the diagonal kernel. The combined ROC-AUC and LMS results across seven protein backbones and three tasks show two clear trends.

First, both LML and WC improve PLASMA's alignment quality. Adding LML yields consistently higher ROC-AUC, confirming that it helps the model concentrate alignment mass on the task-relevant functional substructures it is trained to detect. We also observe that LML can slightly reduce performance on `test_extra`, indicating a mild trade-off between specialization and generalization.

Second, WC is essential for ensuring stable alignment behavior, especially for the parameter-free PLASMA-PF variant. Removing WC causes a substantial performance drop on several backbones (notably ESM2 and ProtBERT), demonstrating that continuity weighting is crucial for suppressing fragmented correspondences and producing coherent alignment plans.

Overall, these results show that LML shapes the model toward identifying the desired functional motifs, while WC is indispensable for robust and stable alignment across architectures, particularly in the parameter-free setting.

Table 10: Ablation study results. Here we ablate two cases: not using the Label Matching Loss (w/o LML) and not using weight correction (w/o WC).

| Task | Method | PROSTT5 | PROTT5 | ANKH | ESM2 | PROTSSN | TM-VEC | PROTBERT |
|------|--------|---------|--------|------|------|---------|--------|----------|
| | | **ROC-AUC** | | | | | | |
| Motif | PLASMA | $.97_{\pm.002}$ | $.97_{\pm.002}$ | $.95_{\pm.002}$ | $.96_{\pm.002}$ | $.96_{\pm.001}$ | $.92_{\pm.004}$ | $.87_{\pm.004}$ |
| | PLASMA-PF | $.94_{\pm.003}$ | $.96_{\pm.002}$ | $.95_{\pm.003}$ | $.93_{\pm.004}$ | $.91_{\pm.003}$ | $.87_{\pm.001}$ | $.85_{\pm.004}$ |
| | PLASMA (w/o LML) | $.95_{\pm.008}$ | $.95_{\pm.006}$ | $.93_{\pm.004}$ | $.91_{\pm.022}$ | $.89_{\pm.018}$ | $.89_{\pm.033}$ | $.85_{\pm.004}$ |
| | PLASMA (w/o WC) | $.91_{\pm.005}$ | $.95_{\pm.004}$ | $.91_{\pm.004}$ | $.87_{\pm.019}$ | $.84_{\pm.003}$ | $.86_{\pm.006}$ | $.73_{\pm.009}$ |
| | PLASMA-PF (w/o WC) | $.74_{\pm.004}$ | $.87_{\pm.002}$ | $.85_{\pm.006}$ | $.44_{\pm.009}$ | $.75_{\pm.009}$ | $.84_{\pm.007}$ | $.60_{\pm.012}$ |
| Binding Site | PLASMA | $.99_{\pm.001}$ | $.99_{\pm.000}$ | $.99_{\pm.000}$ | $.99_{\pm.001}$ | $.99_{\pm.001}$ | $.96_{\pm.003}$ | $.98_{\pm.001}$ |
| | PLASMA-PF | $.99_{\pm.001}$ | $.99_{\pm.001}$ | $.99_{\pm.000}$ | $.96_{\pm.003}$ | $.97_{\pm.001}$ | $.92_{\pm.004}$ | $.90_{\pm.003}$ |
| | PLASMA (w/o LML) | $.99_{\pm.002}$ | $.99_{\pm.001}$ | $.99_{\pm.002}$ | $.97_{\pm.001}$ | $.99_{\pm.000}$ | $.98_{\pm.001}$ | $.98_{\pm.001}$ |
| | PLASMA (w/o WC) | $.99_{\pm.002}$ | $.99_{\pm.001}$ | $.99_{\pm.001}$ | $.98_{\pm.001}$ | $.92_{\pm.008}$ | $.97_{\pm.002}$ | $.77_{\pm.004}$ |
| | PLASMA-PF (w/o WC) | $.91_{\pm.002}$ | $.97_{\pm.001}$ | $.97_{\pm.002}$ | $.49_{\pm.003}$ | $.85_{\pm.006}$ | $.96_{\pm.003}$ | $.67_{\pm.006}$ |
| Active Site | PLASMA | $.99_{\pm.001}$ | $.99_{\pm.001}$ | $.99_{\pm.001}$ | $.99_{\pm.001}$ | $.99_{\pm.002}$ | $.99_{\pm.003}$ | $.99_{\pm.004}$ |
| | PLASMA-PF | $.99_{\pm.002}$ | $.99_{\pm.003}$ | $.99_{\pm.003}$ | $.96_{\pm.002}$ | $.98_{\pm.002}$ | $.98_{\pm.003}$ | $.94_{\pm.006}$ |
| | PLASMA (w/o LML) | $.99_{\pm.001}$ | $.99_{\pm.000}$ | $.99_{\pm.001}$ | $.99_{\pm.005}$ | $.99_{\pm.000}$ | $.98_{\pm.009}$ | $.99_{\pm.000}$ |
| | PLASMA (w/o WC) | $.99_{\pm.001}$ | $.99_{\pm.001}$ | $.99_{\pm.001}$ | $.98_{\pm.000}$ | $.93_{\pm.008}$ | $.99_{\pm.001}$ | $.81_{\pm.033}$ |
| | PLASMA-PF (w/o WC) | $.95_{\pm.002}$ | $.97_{\pm.001}$ | $.98_{\pm.001}$ | $.55_{\pm.008}$ | $.87_{\pm.005}$ | $.99_{\pm.001}$ | $.67_{\pm.009}$ |
| | | **LMS** | | | | | | |
| Motif | PLASMA | $.91_{\pm.007}$ | $.92_{\pm.001}$ | $.92_{\pm.002}$ | $.92_{\pm.005}$ | $.73_{\pm.013}$ | $.76_{\pm.005}$ | $.71_{\pm.007}$ |
| | PLASMA (w/o LML) | $.66_{\pm.135}$ | $.65_{\pm.142}$ | $.92_{\pm.012}$ | $.77_{\pm.170}$ | $.48_{\pm.136}$ | $.68_{\pm.167}$ | $.74_{\pm.012}$ |
| Binding Site | PLASMA | $.93_{\pm.002}$ | $.93_{\pm.003}$ | $.93_{\pm.004}$ | $.93_{\pm.003}$ | $.85_{\pm.006}$ | $.86_{\pm.002}$ | $.84_{\pm.003}$ |
| | PLASMA (w/o LML) | $.87_{\pm.080}$ | $.84_{\pm.110}$ | $.79_{\pm.081}$ | $.49_{\pm.004}$ | $.88_{\pm.000}$ | $.90_{\pm.011}$ | $.89_{\pm.012}$ |
| Active Site | PLASMA | $.97_{\pm.004}$ | $.97_{\pm.004}$ | $.97_{\pm.003}$ | $.97_{\pm.004}$ | $.89_{\pm.016}$ | $.93_{\pm.006}$ | $.89_{\pm.008}$ |
| | PLASMA (w/o LML) | $.89_{\pm.080}$ | $.84_{\pm.131}$ | $.91_{\pm.065}$ | $.79_{\pm.187}$ | $.90_{\pm.000}$ | $.79_{\pm.143}$ | $.89_{\pm.007}$ |

# H CASE STUDY

To provide a clearer view of the residue-level alignment patterns, we include enlarged versions of the alignment matrices corresponding to Figure 5 in the main text. These zoomed-in visualizations highlight how PLASMA identifies coherent local structural motifs across proteins with different folds, lengths, and sequence identities.

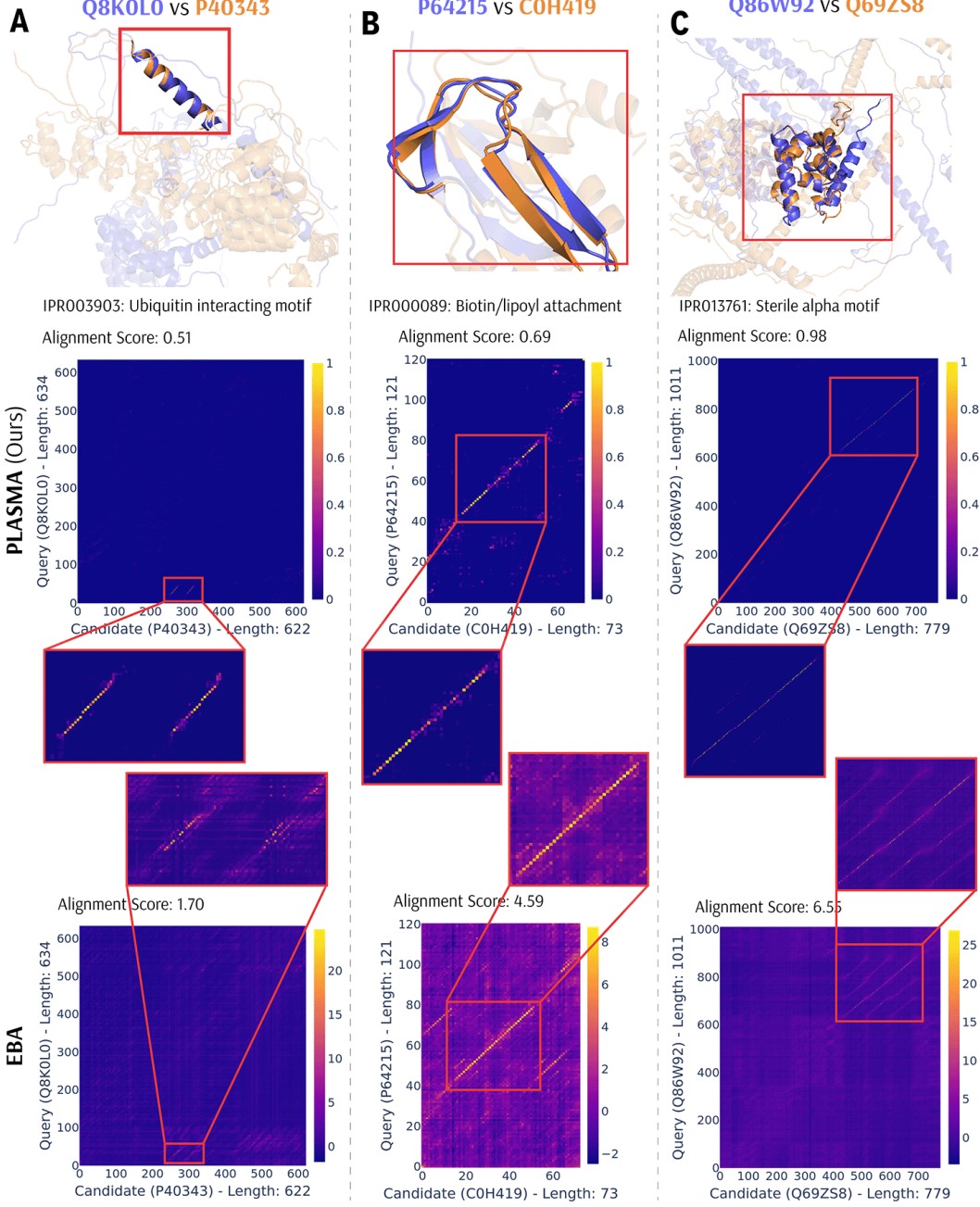

Figure 6: Representative alignment examples across three protein pairs. **A**, P40343 vs Q8K0L0. **B**, P64215 vs C0H419. **C**, Q69ZS8 vs Q86W92. Left: 3D structures with highlighted aligned regions. Center and right: alignment matrices from PLASMA and EBA with zoomed insets. This figure is the higher resolution version of Figure 5.

# I  ALIGNMENT MATRIX VISUALIZATIONS

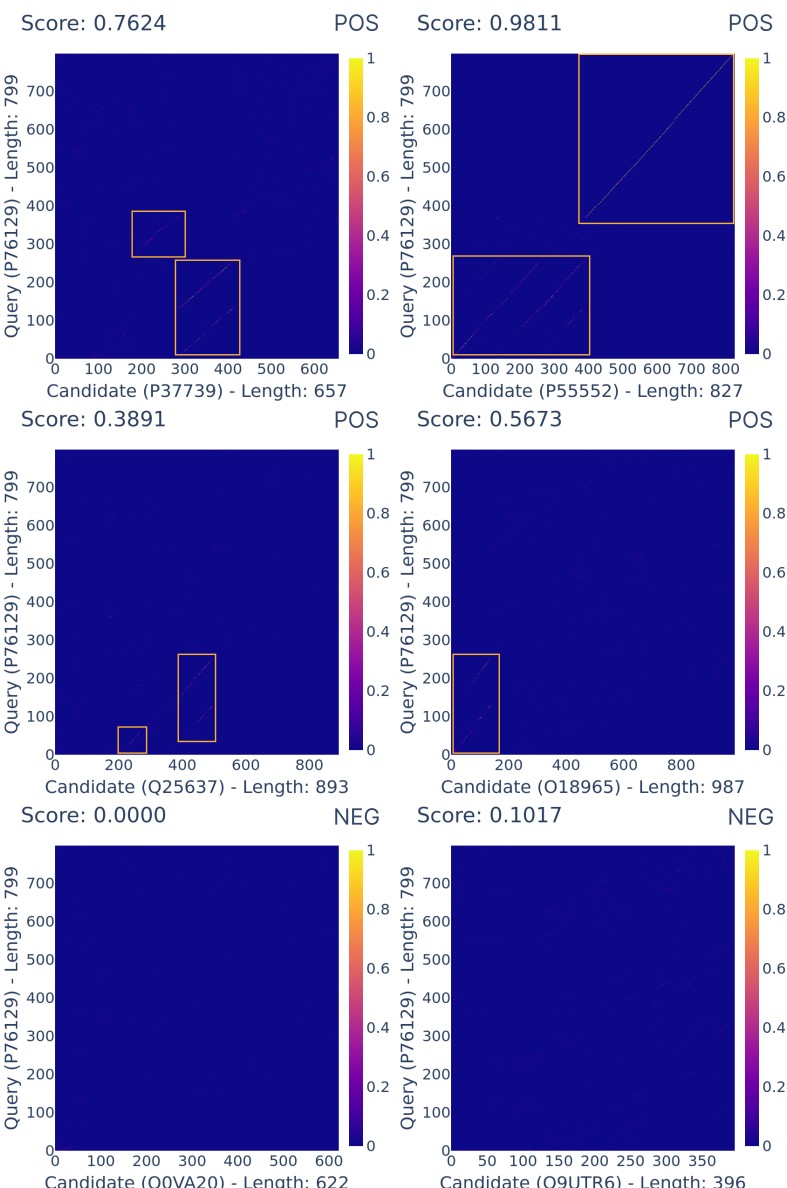

Figure 7: Representative alignment matrices comparing query protein P76129 against six candidate proteins. The visualization shows four positive pairs (POS) with shared substructures and two negative pairs (NEG) without substructure similarity. Orange regions highlight aligned substructures.

Figure 7 demonstrate PLASMA's interpretability by showing clear patterns that correspond to different levels of substructure similarity. The matrices were generated by comparing a single query protein (InterPro ID: P76129) against six different candidate proteins, including four positive pairs sharing functional substructures and two negative pairs without similar functional substructures. The orange-highlighted regions indicate aligned substructures, where larger and more intensely colored blocks correspond to stronger and more extensive alignments. Notably, positive pairs exhibit prominent diagonal patterns reflecting substructure correspondences, while negative pairs show minimal coherent structures and low alignment scores. This visualization validates that PLASMA's alignment scores accurately reflect the underlying biological relationships between protein substructures.

## J    Temperature Parameter Analysis

Figure 8: Effect of Sinkhorn temperature parameter $\tau$ on alignment matrix and score for both PLASMA and PLASMA-PF variants.

Figure 8 illustrates how the Sinkhorn temperature parameter $\tau$ impacts the alignment matrix in both PLASMA variants. The supervised PLASMA variant demonstrates greater stability and maintains meaningful alignment patterns across a wider range of temperature settings compared to PLASMA-PF, highlighting the robustness benefits of end-to-end training.

## K    Performance Evaluation at Different Structural Similarity Threshold

We report the detailed values of the performance at different TM-score thresholds visualized in Figure 3. PLASMA consistently outperforms other baseline methods, especially in low similarity settings (*e.g.*, TM-score $\leq 0.5$ and TM-score $\leq 0.3$).

Table 11: Numerical results of the ROC-AUC Performance at different TM-Align thresholds.

| Task | TM Score | PLASMA | PLASMA-PF | EBA | CosineSim | TM-Align | Foldseek |
|---|---|---|---|---|---|---|---|
| Motif | $\leq 1.0$ | $\mathbf{.96}_{\pm.002}$ | $.95_{\pm.002}$ | $.87_{\pm.003}$ | $.84_{\pm.004}$ | $.78_{\pm.004}$ | $.83_{\pm.004}$ |
| | $\leq 0.7$ | $\mathbf{.95}_{\pm.002}$ | $.94_{\pm.002}$ | $.84_{\pm.004}$ | $.81_{\pm.004}$ | $.73_{\pm.005}$ | $.81_{\pm.004}$ |
| | $\leq 0.5$ | $\mathbf{.93}_{\pm.003}$ | $.93_{\pm.003}$ | $.81_{\pm.005}$ | $.78_{\pm.005}$ | $.66_{\pm.006}$ | $.79_{\pm.005}$ |
| | $\leq 0.3$ | $\mathbf{.92}_{\pm.004}$ | $.91_{\pm.004}$ | $.74_{\pm.006}$ | $.73_{\pm.006}$ | $.58_{\pm.007}$ | $.74_{\pm.006}$ |
| Binding Site | $\leq 1.0$ | $\mathbf{.99}_{\pm.001}$ | $.99_{\pm.001}$ | $.97_{\pm.002}$ | $.96_{\pm.002}$ | $.87_{\pm.003}$ | $.90_{\pm.003}$ |
| | $\leq 0.7$ | $\mathbf{.99}_{\pm.001}$ | $.98_{\pm.002}$ | $.95_{\pm.003}$ | $.93_{\pm.003}$ | $.76_{\pm.006}$ | $.88_{\pm.004}$ |
| | $\leq 0.5$ | $\mathbf{.98}_{\pm.002}$ | $.97_{\pm.003}$ | $.93_{\pm.004}$ | $.91_{\pm.004}$ | $.62_{\pm.007}$ | $.85_{\pm.006}$ |
| | $\leq 0.3$ | $\mathbf{.97}_{\pm.004}$ | $.96_{\pm.004}$ | $.89_{\pm.007}$ | $.88_{\pm.007}$ | $.45_{\pm.010}$ | $.80_{\pm.009}$ |
| Active Site | $\leq 1.0$ | $\mathbf{.99}_{\pm.001}$ | $.99_{\pm.001}$ | $.99_{\pm.001}$ | $.97_{\pm.001}$ | $.94_{\pm.002}$ | $.92_{\pm.003}$ |
| | $\leq 0.7$ | $\mathbf{.99}_{\pm.001}$ | $.98_{\pm.002}$ | $.97_{\pm.002}$ | $.95_{\pm.003}$ | $.88_{\pm.005}$ | $.91_{\pm.004}$ |
| | $\leq 0.5$ | $\mathbf{.98}_{\pm.003}$ | $.96_{\pm.004}$ | $.95_{\pm.004}$ | $.92_{\pm.005}$ | $.76_{\pm.008}$ | $.89_{\pm.006}$ |
| | $\leq 0.3$ | $\mathbf{.96}_{\pm.007}$ | $.90_{\pm.010}$ | $.89_{\pm.011}$ | $.83_{\pm.013}$ | $.59_{\pm.016}$ | $.84_{\pm.013}$ |

## L    SEQUENCE SIMILARITY ANALYSIS

To further examine whether PLASMA's alignment performance is influenced by unintended global similarity, we analyze how PLASMA's alignment score relates to the sequence similarity of the aligned residues. Same as before, we define sequence similarity as the percentage of aligned residue pairs that share the same amino acid type.

Figure 9 presents the distribution of alignment scores and sequence-similarity values across all test pairs. The results show that high alignment scores typically coincide with high alignment coverage rather than high sequence similarity. Many correctly aligned substructures exhibit low sequence similarity despite high PLASMA scores, indicating that the method is driven by shared local 3D geometry rather than residue identity. For negative test pairs, the sequence-similarity values appear highly dispersed, which arises from their extremely low alignment coverage; with very few aligned residue pairs, the resulting sequence-similarity statistic becomes unstable and effectively uninformative. The upper-right region of the plot remains sparse, reflecting our dataset construction protocol that limits the global sequence identity of all test proteins to below 50%.

Overall, this analysis demonstrates that strong PLASMA alignment scores do not depend on high sequence similarity. The method therefore does not rely on global homology signals and is not affected by unintended data leakage.

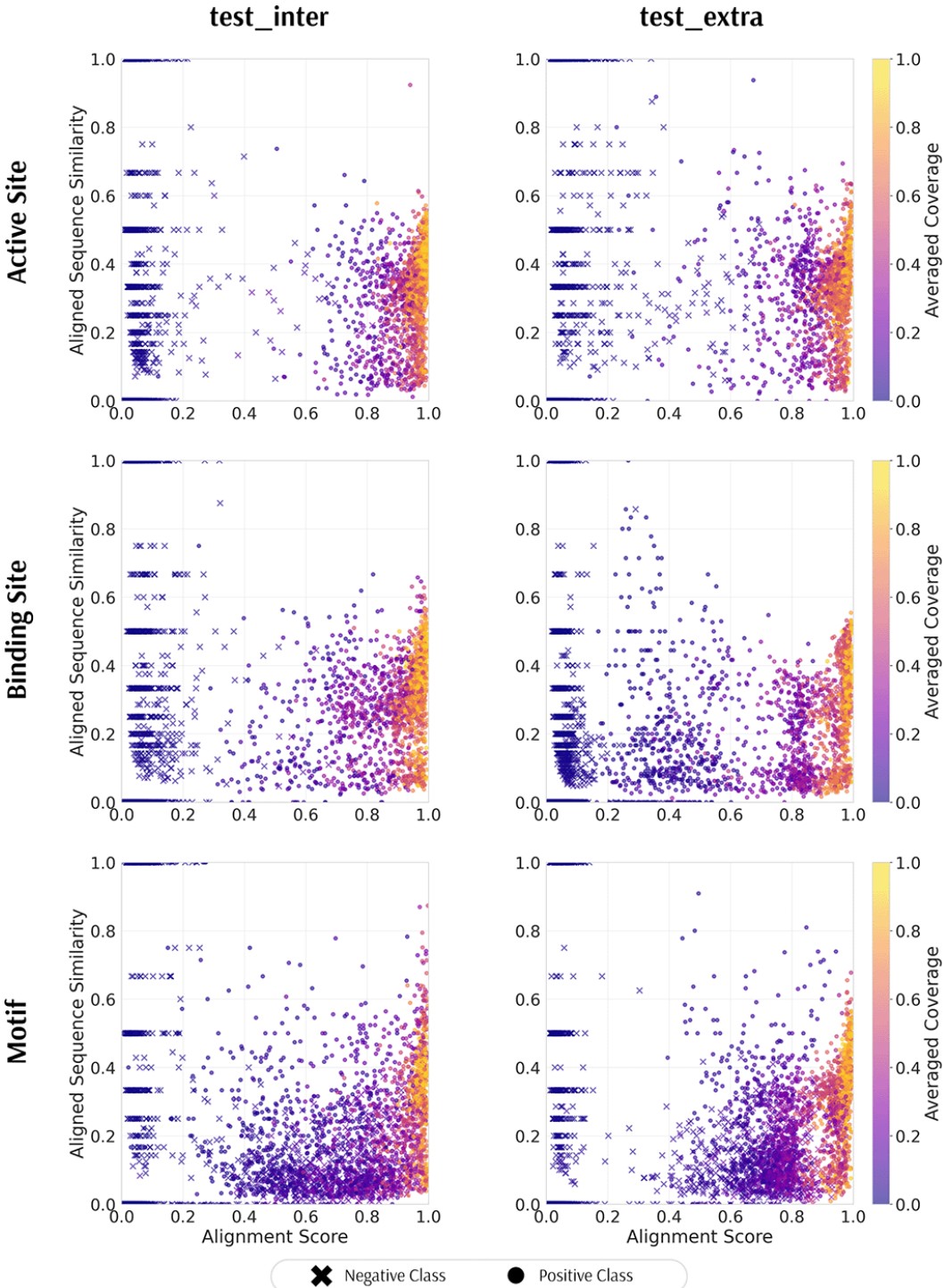

Figure 9: Sequence-similarity patterns of aligned substructures. Each panel shows how the aligned-sequence similarity varies with PLASMA's alignment score, colored by the averaged coverage between the query and candidate proteins. These plots illustrate that high alignment scores do not simply arise from high sequence similarity; the alignment quality is driven by structural correspondence rather than sequence identity. All results use embeddings from Ankh.

# M  HYPERPARAMETER ANALYSIS

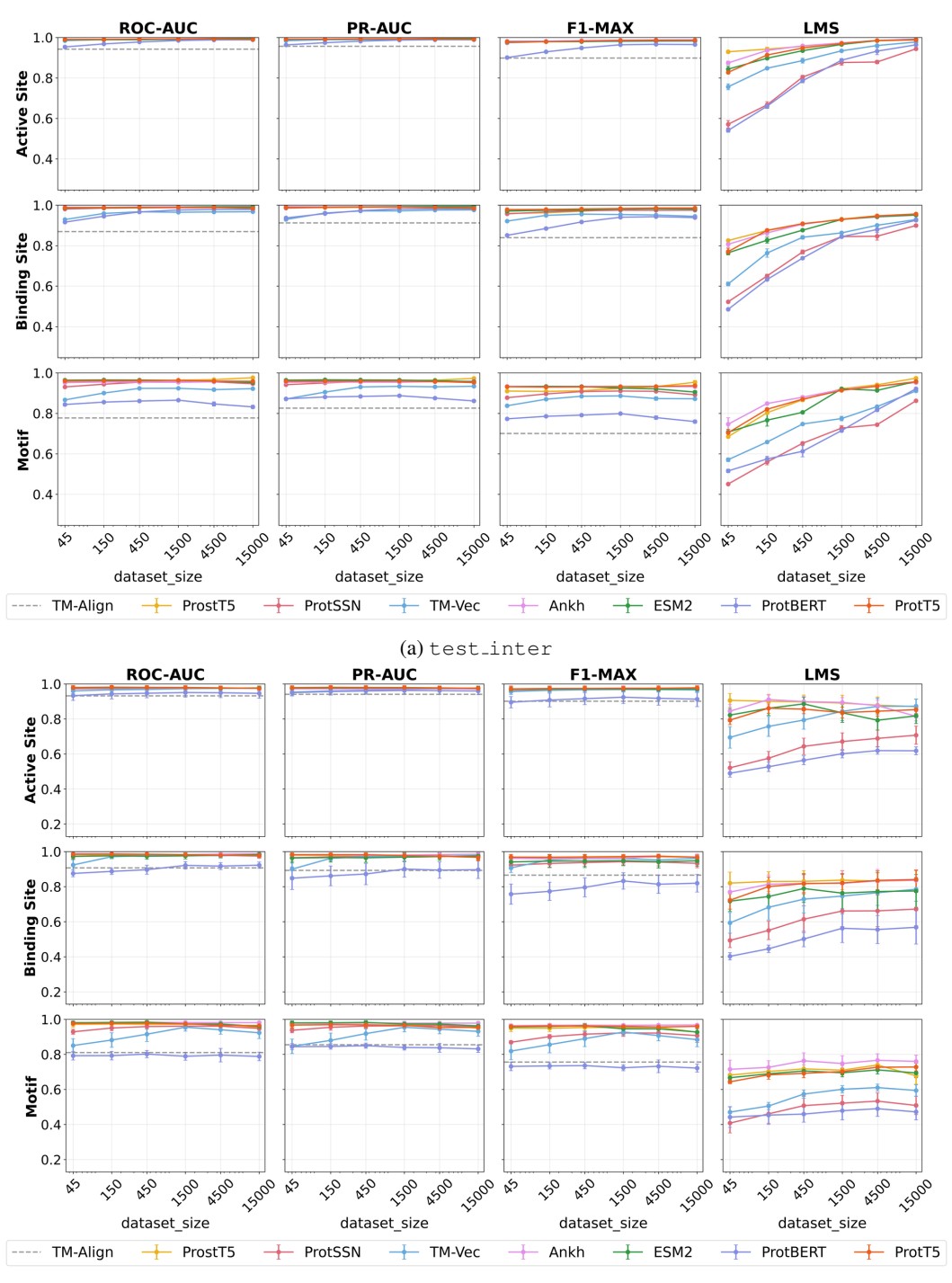

Figure 10: Performance vs dataset fraction. PLASMA demonstrates high performance in predicting the existence of substructure similarities even with minimal training data (45 samples), and, in most cases, this ability remains stable when the dataset size increases. However, the LMS of PLASMA noticeably improves as dataset size increases, indicating that training is important for predicting the precise local of similar substructures.

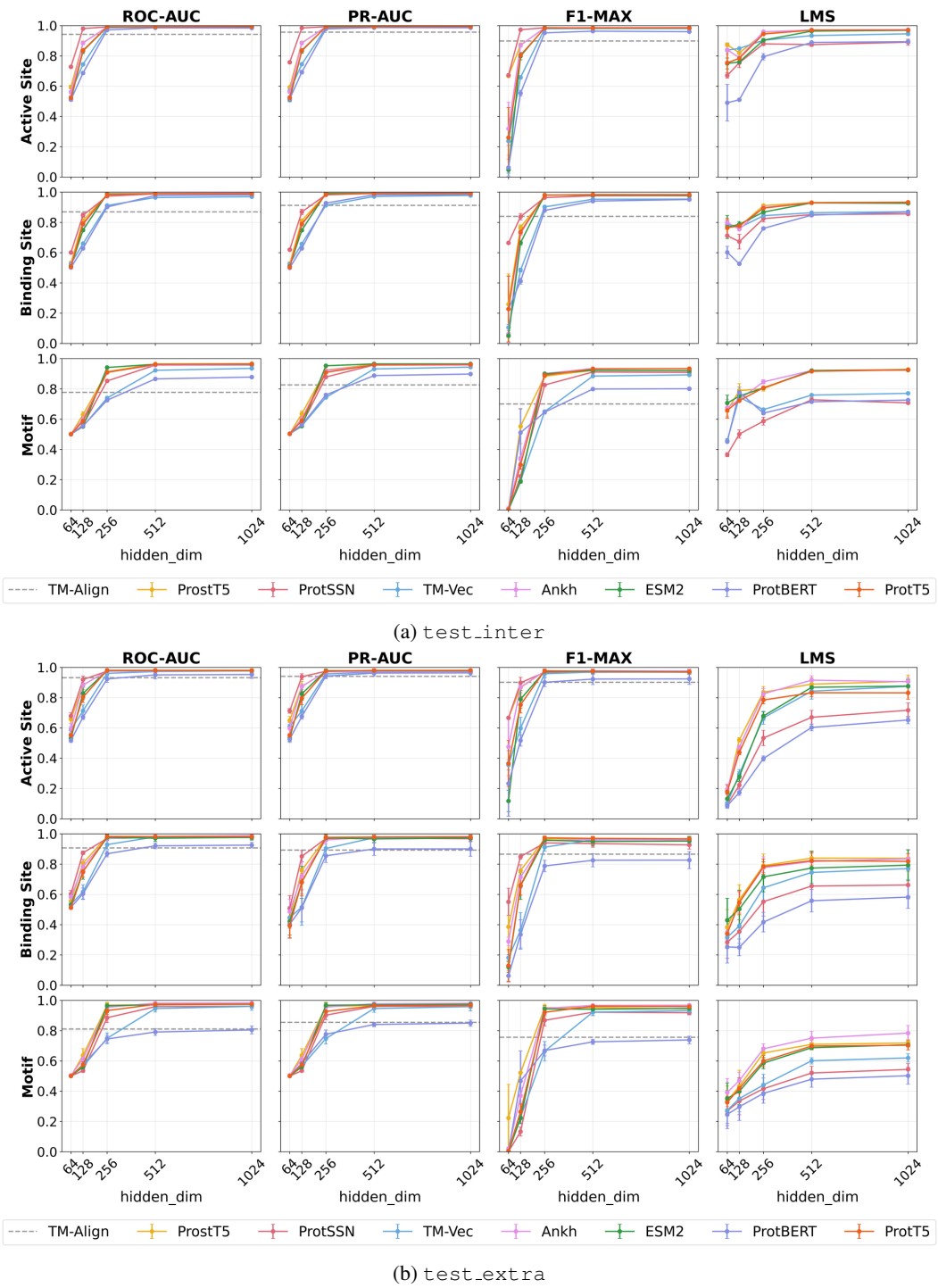

Figure 11: Performance vs hidden dimension size of the siamese network. While PLASMA's performance remains stable when the hidden dimension size is greater than $256$, it would significantly drop when the hidden dimension size is less than this number.

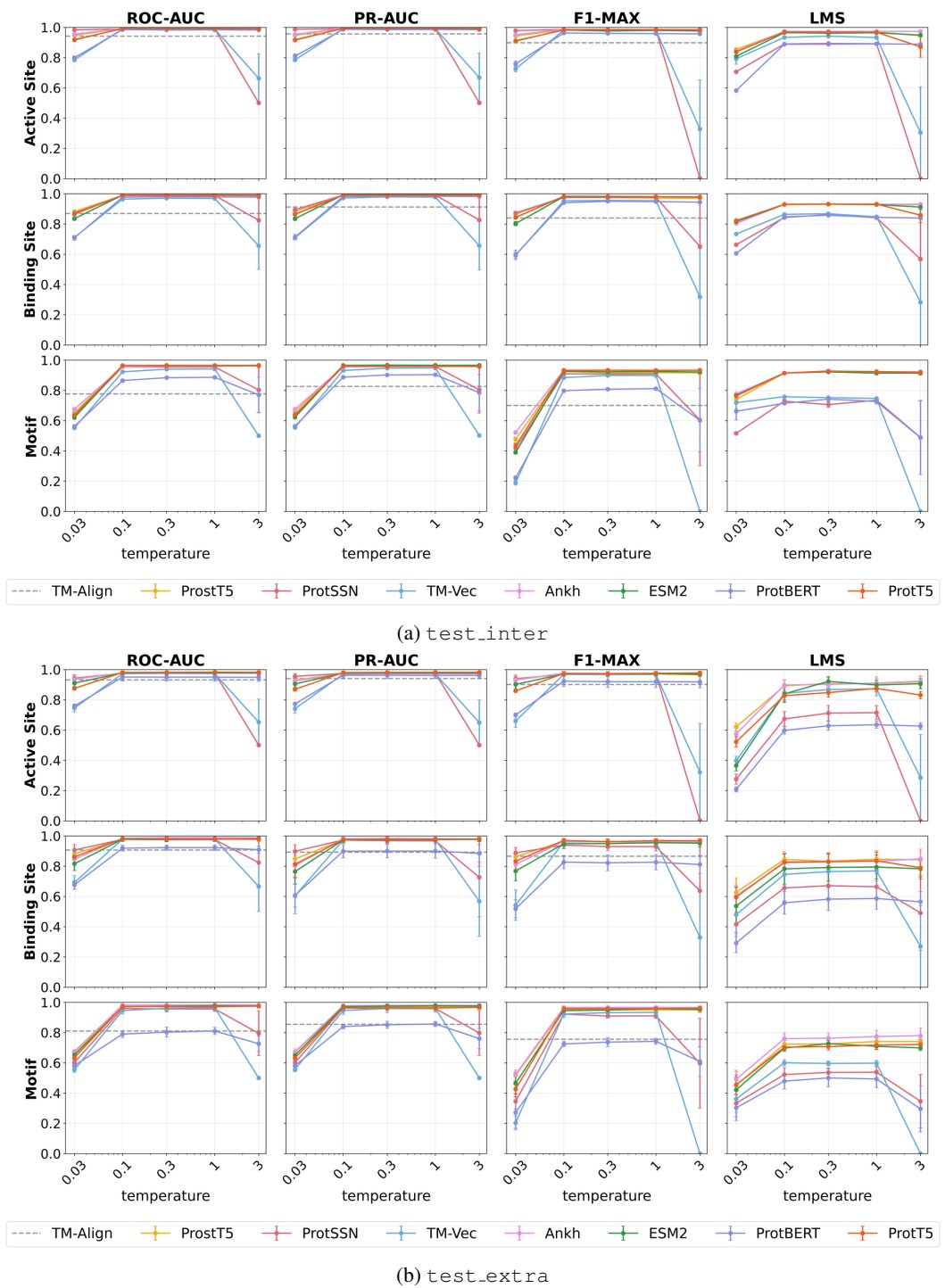

Figure 12: Performance vs Sinkhorn temperature ($\tau$). PLASMA's performance remains stably high within the 0.1–1 range, but when out of this range, PLASMA's performance noticeably drops.

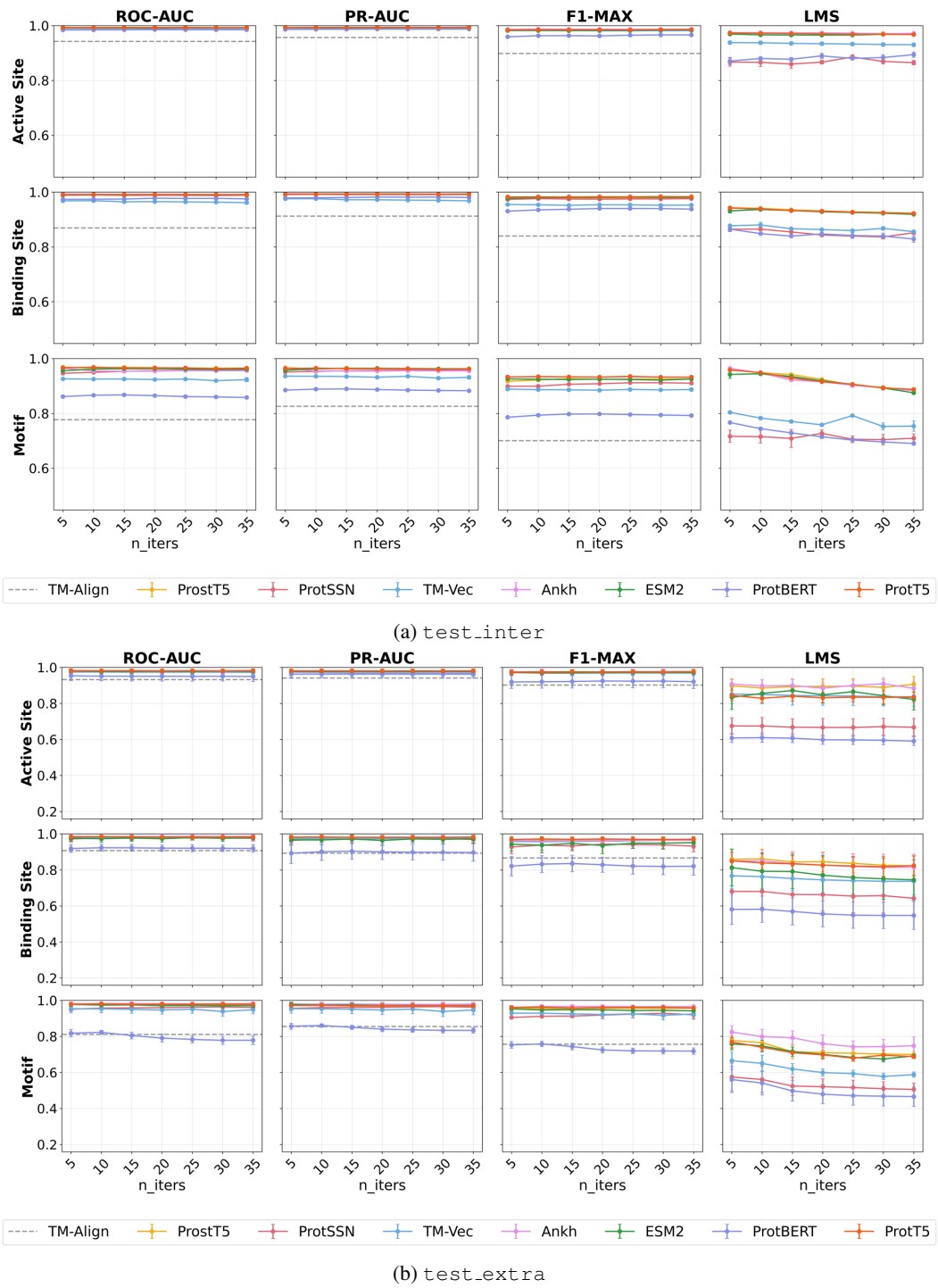

Figure 13: Performance vs number of Sinkhorn iterations $T$. In most cases, PLASMA's performance is insensitive of the setting of $T$, but for analyzing motifs, we can see a subtle decreasing trend as the number of iteration increases.

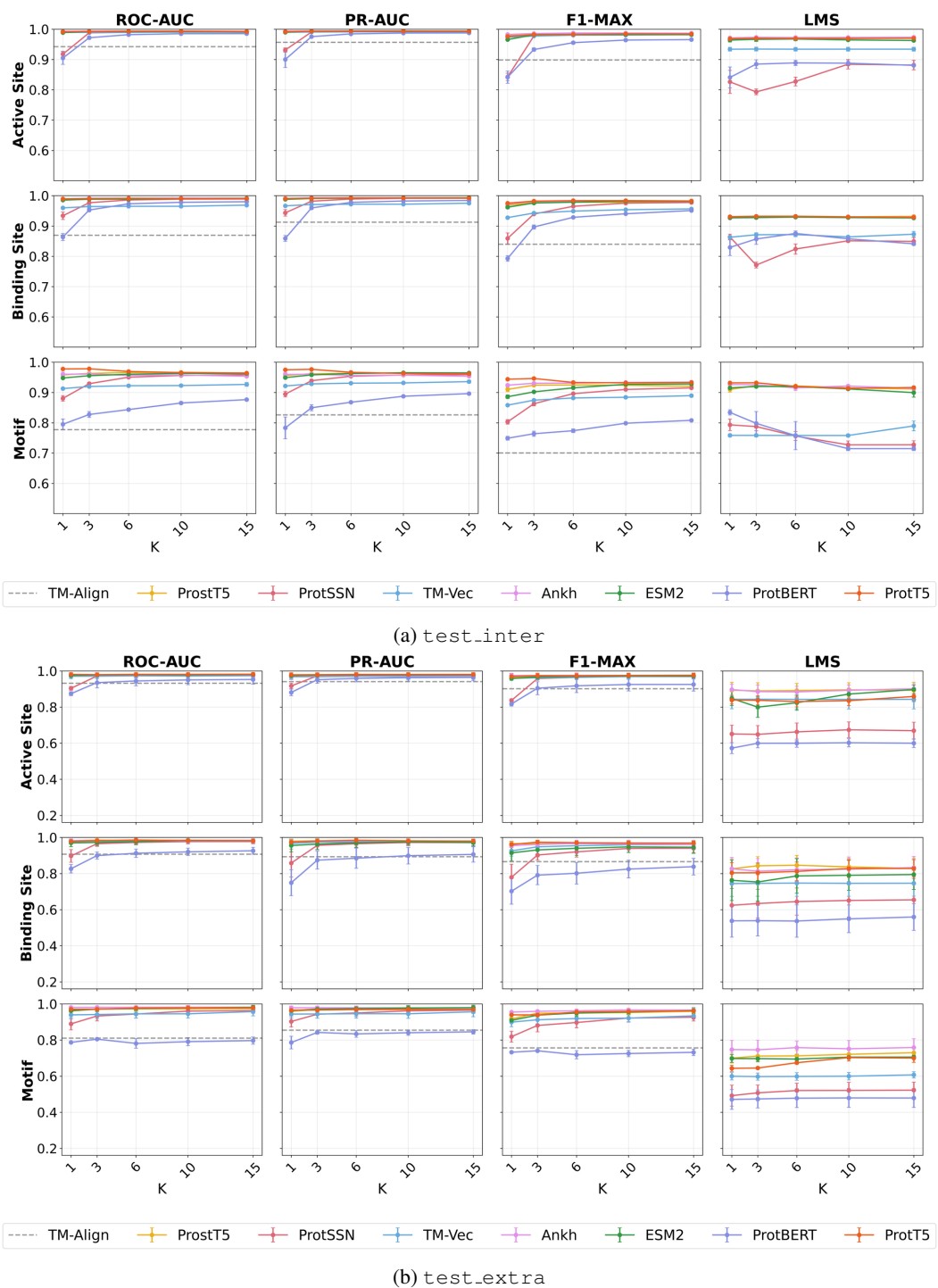

Figure 14: Performance vs the kernel size of the diagonal convolution ($k$). For interpolation tasks and in particular when using PROTSSN, PROTBERT, or TM-VEC as the backbone, there is a trade-off between detecting the existence of substructure similarities and predicting the precise location of similar regions—the former prefers higher $k$ while the latter prefers lower $k$. However, for other cases, PLASMA demonstrates stable performance regardless the choice of $k$.

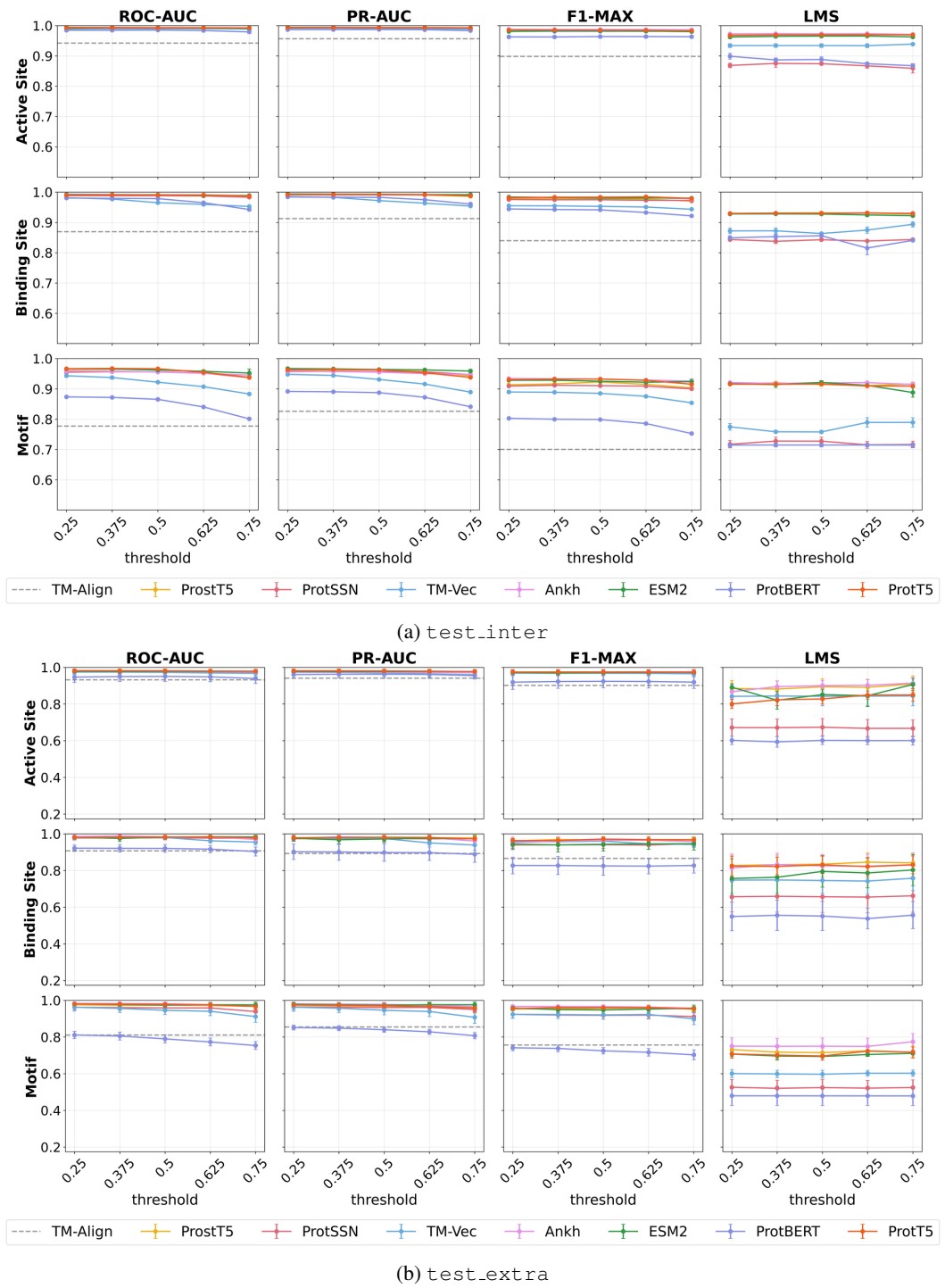

Figure 15: Performance vs residue matching threshold ($\rho$). PLASMA's performance remains stable overall when choosing different $\rho$ values, but for some backbone choices, such as TM-VEC and PROTBERT, PLASMA shows a slight preference over lower $\rho$ values.

# N FURTHER ALIGNMENT MATRIX VISUALIZATIONS

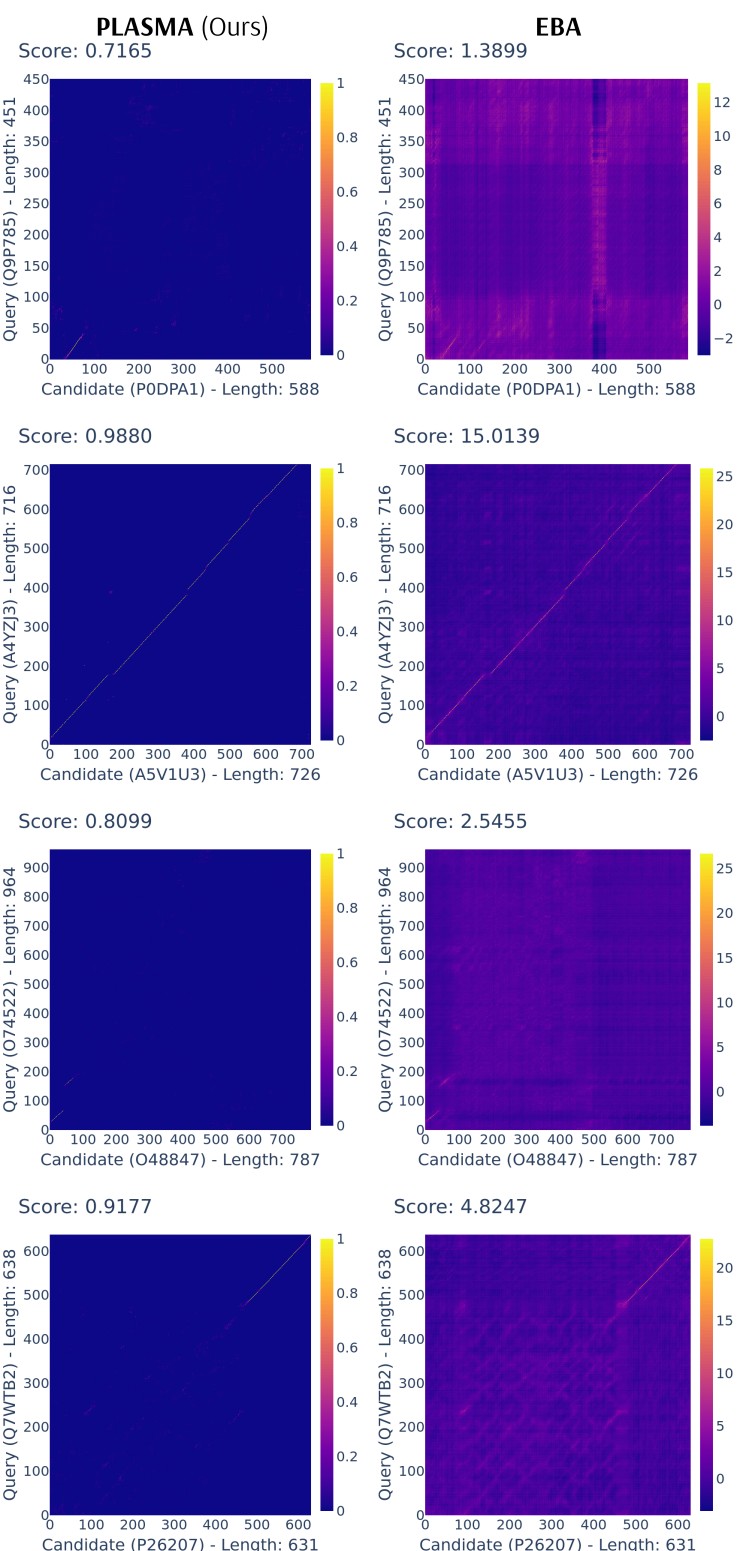

Figure 16: Alignment matrix visualizations of random positive pairs from test_inter. (Part 1)

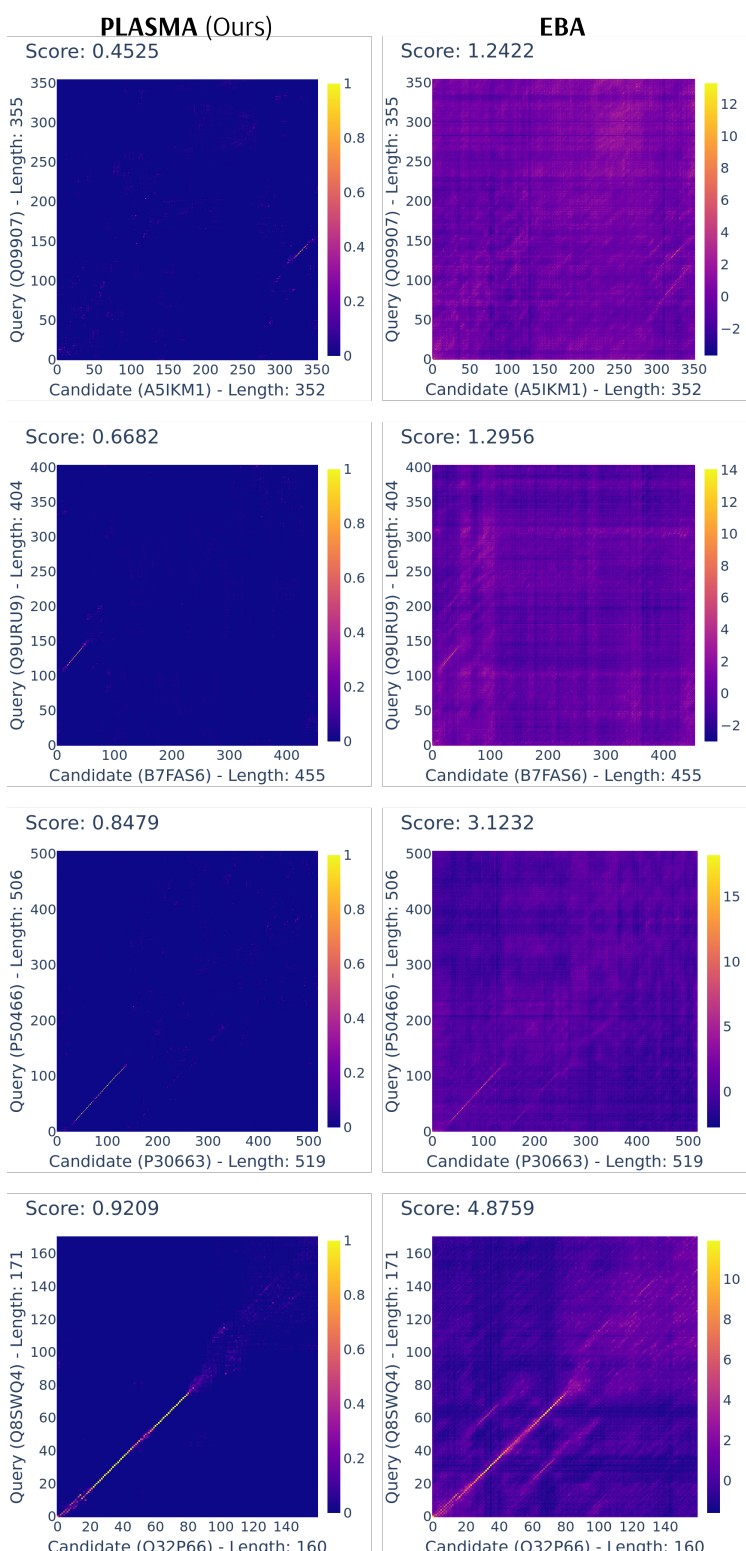

Figure 17: Alignment matrix visualizations of random positive pairs from test_inter. (Part 2)

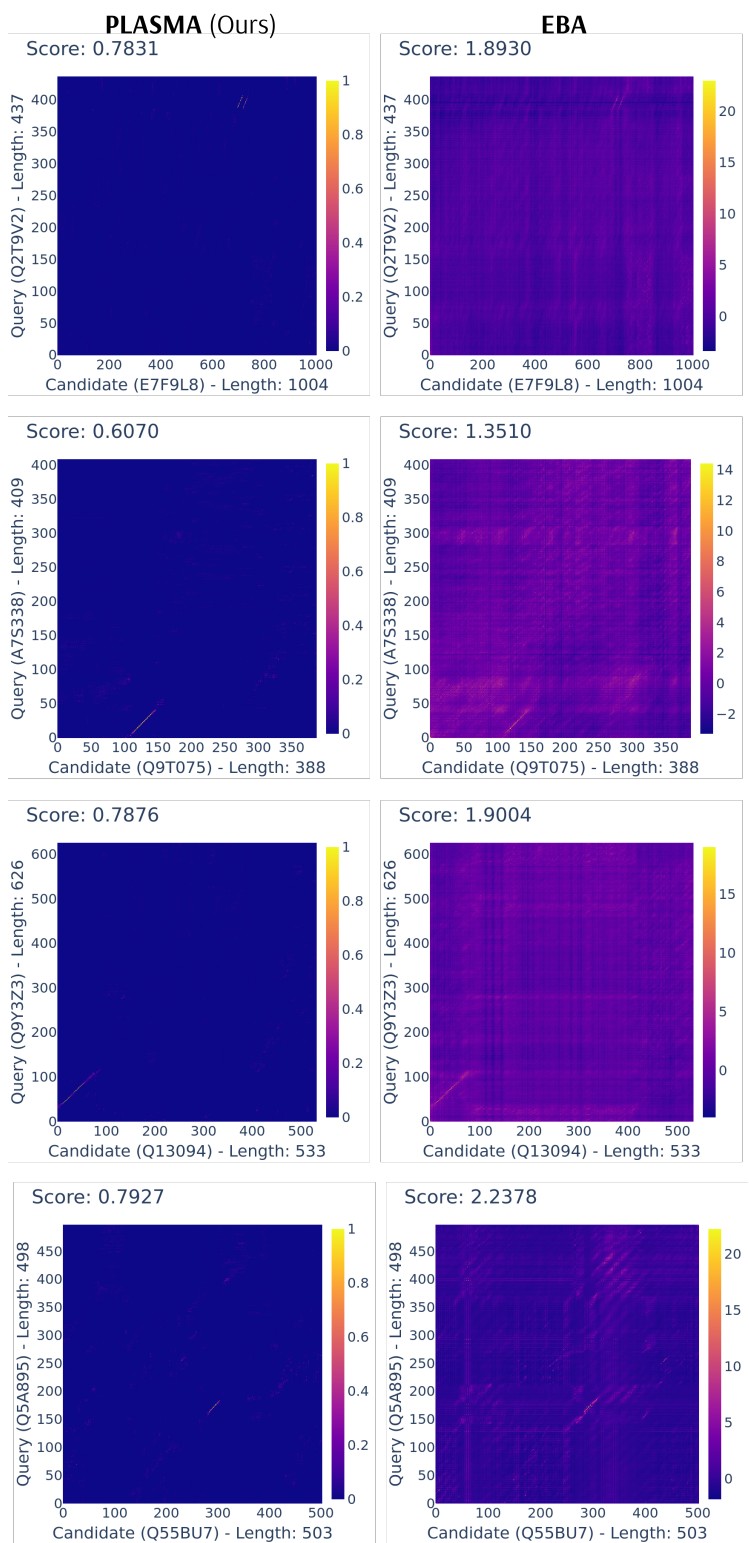

Figure 18: Alignment matrix visualizations of random positive pairs from test_inter. (Part 3)