# OpenReview forum: "Fast and Interpretable Protein Substructure Alignment via Optimal Transport"
_ICLR.cc/2026/Conference — ICLR 2026 Poster_

### Official Review · Reviewer_4sr4 · 2025-10-27

**Soundness:** 4
**Presentation:** 3
**Contribution:** 3
**Rating:** 8
**Confidence:** 2

**Summary:**

This paper addresses the challenge of identifying and comparing local protein substructures, such as active sites, which is critical for understanding protein function but difficult with existing global alignment or sequence-based methods. The authors introduce PLASMA, a deep learning framework that reformulates residue-level substructure alignment as a regularized optimal transport (OT) problem. The method consists of a "Transport Planner" that uses differentiable Sinkhorn iterations to generate an alignment matrix and a "Plan Assessor" to compute an interpretable, overall similarity score. The paper presents both a trainable (PLASMA) and a training-free (PLASMA-PF) variant. Experimental results on benchmark datasets show that PLASMA outperforms existing structure-based (e.g., TM-Align) and embedding-based (e.g., EBA) methods in detecting motifs, binding sites, and active sites, especially in cases of low global similarity. The method is also demonstrated to be significantly more computationally efficient than baselines.

**Strengths:**

Overall, the formulations are clear.

The prosed method is efficient and effective.

The reformulation of substructure alignment as a regularized optimal transport problem is a novel and well-justified approach for this domain. This formulation naturally accommodates the partial and variable-length matches common in biological substructures, which the paper correctly identifies as a limitation of standard OT-based alignment methods.

The strong performance of PLASMA-PF, which requires no task-specific training, makes the method highly practical for scenarios where labeled data is scarce.

The qualitative case studies presented in Section 6.4 and Figure 5 are convincing. They effectively demonstrate PLASMA's ability to identify biologically meaningful local alignments between proteins with very low sequence identity and different global structures, and correctly contrast these interpretable results with the baseline EBA.

**Weaknesses:**

The approach relies on embedding models like protein language models, which may be biased to certain types of proteins. This reliance would potentially result in problems the downstream alignment task. It would be beneficial to test PLASMA on underrepresented proteins, e.g., orphan proteins.

In table 1, the results are tested over 3 different seeds. More repetition would make the results more robust.

The justification for the "Plan Assessor" design (Section 4) feels somewhat heuristic. Specifically, the confidence weight $\alpha$ is derived using a 2D convolution with a fixed identity kernel to detect diagonal patterns. The rationale for this specific operator over other potential methods (e.g., a learnable kernel, or a different path-detection algorithm) is not fully explored, nor is the sensitivity to the kernel size.

The paper does not provide an ablation study on the key hyperparameters of the Sinkhorn algorithm. The temperature parameter $\tau$ and the number of iterations $T$ (Section 3) are critical for controlling the sparsity and accuracy of the final alignment matrix $\Omega$, as the paper itself notes. While Appendix D.2 mentions hyperparameter setup, no results are shown to demonstrate how performance (e.g., accuracy and alignment sparsity) varies with these choices.

The formulation of PLASMA-PF is ambiguous. The paper states that it "bypasses the siamese network and directly computes costs from $LN(H)$". It is not explicitly stated how this cost is computed. It is unclear if it still uses the hinge-loss formulation from Equation 2 with $\phi_{\theta}$ as an identity function, or if it uses a different distance metric (e.g., cosine distance) entirely.

**Questions:**

Could the authors please clarify the exact mathematical formulation of the cost matrix $\mathcal{C}$ used in the parameter-free PLASMA-PF variant? How is the cost computed "directly... from $LN(H)$," and how does this computation relate to Equation 2?

Could the authors elaborate on the claim that the LMS metric cannot be meaningfully applied to the EBA baseline? Since EBA produces residue-level alignments, it seems this metric should be applicable and would provide a valuable head-to-head comparison of alignment quality, not just score-based classification performance.

---

> ### Author Response · Authors · 2025-11-22
> **Response to Reviewer 4sr4 (1/2)**
>
> We thank the reviewer for their constructive comments. Below is our point-by-point response.
>
> ## Weaknesses
> ### W1. PLM Bias and Underrepresented / Orphan Proteins
> We clarify these two points separately.
>
> **(i) On dependence on protein language models.**
> PLASMA does rely on protein language models for residue-level representations, but our framework is not tied to a single backbone. In our experiments, we have evaluated PLASMA with seven different PLM encoders and observed consistently strong performance across these choices. This suggests that the gains come from the alignment formulation and OT-based plan rather than from a particular PLM instance.
>
> **(ii) On testing underrepresented / orphan proteins.**
> The concern about underrepresented or orphan proteins primarily arises at the sequence level. PLASMA operates on local 3D motifs, whose geometric diversity in nature is far more limited and widely reused across proteins, including those considered orphan. In this sense, the notion of underrepresentation is less meaningful at the structural-motif level.
>
> Moreover, orphan proteins typically lack known residue-level correspondences or curated structural annotations, making them an impractical evaluation target for a supervised alignment task. Alignment-based analysis is not the standard tool for studying orphan proteins, and reliable ground truth is rarely available. Our current benchmark already spans a wide diversity of folds, functions, and lengths, which is better aligned with the scope of PLASMA.
>
> We have clarified this distinction in the revision. In the revised Section 2:
> > *In our experiments, we instantiate $H_q$ and $H_c$ with seven diverse protein representation backbones (Section 6), and observe consistent alignment behavior across them, indicating that \model~is not tied to a particular choice of encoder.*
>
>
> ### W2. Experiments with different seeds
> Thank you for the suggestion. We agree that more repetitions can further reduce variance, but in our experiments the standard deviations across the three seeds are already very small, indicating stable behavior and sufficient robustness for the comparisons made. We therefore report three runs for efficiency and clarity, while ensuring that variance is low enough to support the conclusions.
>
> ### W3. Design of Plan Assessor and the identity-kernel diagonal detector
> Our initial submission already included a set of Sinkhorn-related hyperparameter analyses in Appendix M (Figures 10–15). These figures systematically evaluate how key OT and alignment components affect alignment sharpness, sparsity, and overall performance. Specifically, we vary:
> - dataset fraction used for training (Fig. 10)
> - hidden dimension size of the Siamese encoder (Fig. 11)
> - Sinkhorn temperature τ (Fig. 12)
> - number of Sinkhorn iterations T (Fig. 13)
> - kernel size k of the diagonal convolution used for weighting (Fig. 14)
> - residue matching threshold ρ that determines the contribution of aligned residue pairs (Fig. 15)
> These investigations collectively demonstrate how each hyperparameter influences the stability and behavior of the resulting alignment plan, addressing all the reviewer’s concerns regarding OT temperature, iteration count, and sparsity–accuracy trade-offs.
> ### W4. Formulation of PLASMA-PF
> PLASMA-PF shares the same cost formulation as PLASMA pipeline. The only difference is in Eq.2, where the Siamese network is removed and the residue embeddings are used directly.  We have revised the manuscript for a clearer presentation:
> > *...PLASMA-PF, which bypasses the siamese network and operates directly on residue embeddings. The cost used in the OT objective follows (2) with no architectural components removed other than the encoder.*

---

> ### Author Response · Authors · 2025-11-22
> **Response to Reviewer 4sr4 (2/2)**
>
> ## Questions
> ### Q1. Formulation of PLASMA-PF
> Please see the answer from W4 above.
>
> ### Q2. Additional Ablation Studies
> We would like to clarify why LMS cannot be meaningfully applied to EBA in our setting.
> - **EBA’s alignment scores are unbounded and not normalized.**
> For interpretability purposes, LMS takes a probabilistic or normalized alignment plan where each row and column reflects how the alignment mass is distributed. While EBA outputs arbitrary similarity values without normalization, LMS is not well-defined on top of these scores.
> - **EBA does not enforce one-to-one or approximately mass-conserving correspondences.**
> Even after min–max normalization, a residue in EBA may align to many residues with high values, and the total alignment weight for a residue can far exceed 1. Under such conditions, the LMS numerator becomes artificially inflated, and the metric collapses to trivial values—often near one—for almost all examples.
> - **Row/column normalization would fundamentally change the method.**
> Applying row/column softmax or doubly stochastic normalization to EBA to make LMS valid effectively turns EBA into a Sinkhorn-based OT planner, which changes the method itself. Such a transformation is closer to integrating EBA with PLASMA rather than evaluating the original EBA baseline, and we view this as an interesting direction for future work.
>
> For these reasons, LMS cannot be interpreted consistently for EBA without altering its alignment mechanism. Therefore, we report EBA results only in terms of score-based performance.

---

> > ### Comment · Reviewer_4sr4 · 2025-11-24
> >
> > Thank you for your response. My points are addressed.

---

> > > ### Author Response · Authors · 2025-11-25
> > >
> > > Thank you for the follow-up and for confirming that your points have been addressed.

---

### Official Review · Reviewer_NvT1 · 2025-10-27

**Soundness:** 3
**Presentation:** 3
**Contribution:** 3
**Rating:** 4
**Confidence:** 3

**Summary:**

This paper works on identifying and understanding protein structure for efficient and interpretable residue-level protein substructure alignment. The authors reformulate the problem as a regularized optimal transport task and leverage differentiable Sinkhorn iterations to solve the alignment problem. Besides, the authors use extensive quantitative evaluations and three biological case studies to demonstrate that the proposed method, PLASMA, achieves accurate, lightweight, and interpretable residue-level alignment. Moreover, the authors propose a training-free variant as an alternative when training data is not available.

**Strengths:**

1. The challenges behind protein alignment are well defined, and the motivation is clearly demonstrated.
2. Extensive experiments on various datasets, baselines, and backbone models clearly prove the effectiveness of the proposed method.
3. The paper is well written and organized, with code published.

**Weaknesses:**

1. Using optimal transport to align two graphs is not a novel idea, as can be seen in papers[1][2][3]. What are the differences and advances of PLASMA compared to other methods?

2. The authors claim that PLASMA can sufficiently address the variable length alignment challenge, but there are no experiments to prove this statement. What is the performance of PLASMA when the query and target protein have different lengths?

3. There is no ablation study to show the effectiveness of each component in PLASMA.

4. In Eq. 7, the authors want to introduce a loss that focuses exclusively on labeled substructures. Then how does the number of labeled substrcutres affect the performance of PLASMA? And is it necessary to let the model focus on these labeled substructures? What are the reasons to include this loss?

[1] Lee, John, et al. "Hierarchical optimal transport for multimodal distribution alignment." Advances in neural information processing systems 32 (2019).
[2] J. Tang, W. Zhang, J. Li, K. Zhao, F. Tsung and J. Li, "Robust Attributed Graph Alignment via Joint Structure Learning and Optimal Transport," 2023 IEEE 39th International Conference on Data Engineering (ICDE), Anaheim, CA, USA, 2023, pp. 1638-1651, doi: 10.1109/ICDE55515.2023.00129.
[3] A. T. Riahi, G. Woollard, F. Poitevin, A. Condon and K. D. Duc, "AlignOT: An Optimal Transport Based Algorithm for Fast 3D Alignment With Applications to Cryogenic Electron Microscopy Density Maps," in IEEE/ACM Transactions on Computational Biology and Bioinformatics, vol. 20, no. 6, pp. 3842-3850, Nov.-Dec. 2023, doi: 10.1109/TCBB.2023.3327633.

**Questions:**

Please refer to the weaknesses.

---

> ### Author Response · Authors · 2025-11-22
> **Response to Reviewer NvT1 (1/2)**
>
> ## Weaknesses
> ### W1. Novelty of PLASMA
> Using optimal transport for alignment itself is not new, and we have illustrated this statement by the cited works on multimodal distribution alignment, attributed graph alignment, and 3D map alignment [1–3]. However, our contribution is not a new OT algorithm, but a new **problem formulation and application setting** for residue-level local structural motif alignment between proteins with generalizability and explicit interpretability.
>
> More concretely, the existing OT-based alignment works differ from PLASMA in both input and objective:
> - [1] focuses on hierarchical OT for aligning multimodal distributions, not protein structures, and does not produce residue-level correspondences.
> - [2] studies attributed graph alignment with OT, targeting discrete graphs and attribute matching, rather than continuous 3D geometry of protein backbones and side chains.
> - [3] (AlignOT) aligns volumetric Cryo-EM density maps using OT to register 3D densities, again without modeling residue-level protein motifs or coupling to protein language models.
>
> PLASMA is specifically designed for **residue-wise local motif alignment in proteins**, combining (i) residue-level geometric features, (ii) a learnable OT cost, and (iii) a Siamese pLM encoder, to produce interpretable residue–residue alignment matrices that capture functionally relevant local 3D motifs under low sequence similarity. We have revised our manuscript and made clarifications in the introduction section about our contribution:
> > *Our work addresses these limitations through three contributions. First, we introduce a formulation of residue-level local structural alignment based on regularized optimal transport with a learnable geometric cost, which provides a principled and flexible way to define correspondence and enables efficient, fully parallel implementation. Second, this formulation enables clear and interpretable residue–residue correspondences and naturally supports partial, variable-length, and non-sequential motif alignments, resolving the difficulty of obtaining reliable local alignments. Third, PLASMA produces a normalized and interpretable similarity score through its OT-based objective, overcoming the limitations of existing approaches whose alignment matrices or similarity measures lack a consistent probabilistic meaning. Our experiments show strong generalization to low-homology structures, and the case studies demonstrate the biological interpretability and practical utility of the resulting alignments*.
>
>
> ### W2. Variable-length Alignment
> In our benchmark, protein pairs naturally span a wide range of sequence lengths and length ratios, so the reported results already evaluate PLASMA in variable-length settings. Additional evidence comes from the alignment matrix visualizations in Figure 5 (Section 6), Figure 8 (Appendix J), and Figures 16–18 (Appendix M). In these examples, the difference in sequence lengths is directly visible from the axis ranges, yet PLASMA consistently produces coherent and stable residue-level alignment plans. These visualizations further confirm that PLASMA handles substantial length differences without degradation in alignment quality.

---

> > ### Author Response · Authors · 2025-11-22
> > **Response to Reviewer NvT1 (2/2)**
> >
> > ### W3. Ablation Studies
> > We thank the reviewer for the suggestion. Below we provide a combined ablation across seven protein models and all three tasks. Across all backbones and tasks, the full PLASMA model outperforms all ablated variants, demonstrating that each component contributes meaningfully, and the plan assessor is not reliant on any single heuristic.
> > These results are also included in the revised Appendix G.
> >
> >
> > **ROC-AUC**
> >
> > **1. Motif**
> > | Method | ProstT5 | ProtT5 | Ankh | ESM2 | ProtSSN | TM-Vec | ProtBERT |
> > |---|--|-|--|--|---|--|--|
> > | PLASMA | **.97±.002** | **.97±.002** | **.95±.002** | **.96±.002** | **.96±.001** | **.92±.004** | **.87±.004** |
> > | PLASMA-PF | .94±.003 | .96±.002 | **.95±.003** | .93±.004 | .91±.003 | .87±.001 | .85±.004 |
> > | PLASMA (w/o LML) | .95±.008 | .95±.006 | .93±.004 | .91±.022 | .89±.018 | .89±.033 | .85±.004 |
> > | PLASMA (w/o WC) | .91±.005 | .95±.004 | .91±.004 | .87±.019 | .84±.003 | .86±.006 | .73±.009 |
> > | PLASMA-PF (w/o WC) | .74±.004 | .87±.002 | .85±.006 | .44±.009 | .75±.009 | .84±.007 | .60±.012 |
> >
> > **2. Binding Site**
> >
> > | Method | ProstT5 | ProtT5 | Ankh | ESM2 | ProtSSN | TM-Vec | ProtBERT |
> > |---|--|-|--|--|---|--|--|
> > | PLASMA | **.99±.001** | **.99±.000** | **.99±.000** | **.99±.001** | **.99±.001** | .96±.003 | **.98±.001** |
> > | PLASMA-PF | **.99±.001** | **.99±.001** | **.99±.000** | .96±.003 | .97±.001 | .92±.004 | .90±.003 |
> > | PLASMA (w/o LML) | **.99±.002** | **.99±.001** | **.99±.002** | .97±.001 | **.99±.000** | **.98±.001** | **.98±.001** |
> > | PLASMA (w/o WC) | **.99±.002** | **.99±.001** | **.99±.001** | .98±.001 | .92±.008 | .97±.002 | .77±.004 |
> > | PLASMA-PF (w/o WC) | .91±.002 | .97±.001 | .97±.002 | .49±.003 | .85±.006 | .96±.003 | .67±.006 |
> >
> > **3. Active Site**
> > | Method | ProstT5 | ProtT5 | Ankh | ESM2 | ProtSSN | TM-Vec | ProtBERT |
> > |---|--|-|--|--|---|--|--|
> > | PLASMA | **.99±.001** | **.99±.001** | **.99±.001** | **.99±.001** | **.99±.002** | **.99±.003** | **.99±.004** |
> > | PLASMA-PF | **.99±.002** | **.99±.003** | **.99±.003** | .96±.002 | .98±.002 | .98±.003 | .94±.006 |
> > | PLASMA (w/o LML) | **.99±.001** | **.99±.000** | **.99±.001** | **.99±.005** | **.99±.000** | .98±.009 | **.99±.000** |
> > | PLASMA (w/o WC) | **.99±.001** | **.99±.001** | **.99±.001** | .98±.000 | .93±.008 | **.99±.001** | .81±.033 |
> > | PLASMA-PF (w/o WC) | .95±.002 | .97±.001 | .98±.001 | .55±.008 | .87±.005 | **.99±.001** | .67±.009 |
> >
> > **LMS**
> >
> > **1. Motif**
> > | Method | ProstT5 | ProtT5 | Ankh | ESM2 | ProtSSN | TM-Vec | ProtBERT |
> > |---|--|-|--|--|---|--|--|
> > | PLASMA | **.91±.007** | **.92±.001** | **.92±.002** | **.92±.005** | **.73±.013** | **.76±.005** | .71±.007 |
> > | PLASMA (w/o LML) | .66±.135 | .65±.142 | **.92±.012** | .77±.170 | .48±.136 | .68±.167 | **.74±.012** |
> >
> > **2. Binding Site**
> > | Method | ProstT5 | ProtT5 | Ankh | ESM2 | ProtSSN | TM-Vec | ProtBERT |
> > |---|--|-|--|--|---|--|--|
> > | PLASMA | **.93±.002** | **.93±.003** | **.93±.004** | **.93±.003** | .85±.006 | .86±.002 | .84±.003 |
> > | PLASMA (w/o LML) | .87±.080 | .84±.110 | .79±.081 | .49±.004 | **.88±.000** | **.90±.011** | **.89±.012** |
> >
> > **3. Active Site**
> > | Method | ProstT5 | ProtT5 | Ankh | ESM2 | ProtSSN | TM-Vec | ProtBERT |
> > |---|--|-|--|--|---|--|--|
> > | PLASMA | **.97±.004** | **.97±.004** | **.97±.003** | **.97±.004** | .89±.016 | **.93±.006** | **.89±.008** |
> > | PLASMA (w/o LML) | .89±.080 | .84±.131 | .91±.065 | .79±.187 | **.90±.000** | .79±.143 | **.89±.007** |
> >
> > **Note:** Bold values indicate best performance for each embedding method within each task-metric combination.
> >
> > ### W4. Loss on Labeled Substructures (Eq. 7)
> > The loss in Eq. 7 is designed to help PLASMA focus more reliably on functionally or structurally annotated substructures, which often occupy only a small portion of residues and would otherwise be diluted by background alignment. This loss does not force the model to rely solely on labeled motifs; it simply increases the relative weight on annotated regions when such labels are available. The number of labeled substructures therefore affects the model only by controlling how strongly this bias is expressed. The loss remains optional and does not harm overall alignment when labels are sparse.
> >
> > From a biological perspective, emphasizing known local structural motifs is meaningful because these motifs often correspond to active sites, binding pockets, catalytic loops, or other experimentally validated functional regions, which are precisely the units researchers care about when studying protein mechanisms or designing targeted mutations.
> >
> > We have revised the method section to clarify this motivation and the optional nature of this loss. In Section 4 of the revised manuscript, it now reads:
> > > *This loss provides an optional bias toward annotated local structural motifs when such labels exist. These regions are typically small and structurally meaningful (\eg catalytic or binding motifs), and emphasizing them helps the model avoid being dominated by background alignments.*

---

> ### Comment · Reviewer_NvT1 · 2025-11-25
>
> Thanks for the rebuttal. My concerns are addressed. I will retain my score.

---

> > ### Author Response · Authors · 2025-11-25
> >
> > Thank you for the update. As you’ve indicated that all concerns are fully addressed, we hope this can be taken into account in the final score, and we are happy to engage in any further discussion if needed.

---

### Official Review · Reviewer_WJJf · 2025-10-31

**Soundness:** 3
**Presentation:** 3
**Contribution:** 3
**Rating:** 8
**Confidence:** 3

**Summary:**

This submission introduces PLASMA, a method to obtain protein substructure alignment. By embedding each residue of a pair of proteins with a pLM, a soft alignment matrix coupled with an overall alignment score. The alignment matrix is obtained by passing the pLM embeddings through an optional siamese network yielding a cost matrix, to which Sinkhorn iterations are applied. From the resulting alignment matrix, an interpretable score is calibrated to surface regions of high structural similarity. Experiments show very high performance when benchmarked on substructure identification tasks.

**Strengths:**

The paper tackles a relevant problem in the field of protein representation learning. The paper is clearly written, the benchmarks are sound and convincing, and the methodology is original. PLASMA clearly demonstrates superior performance while simultaneously also being much faster than some of the previous methods. The VenuX benchmark is a strong set of experiments that clearly show generalization abilities. The testing of various pLMs show that the method is robust and adaptable.

**Weaknesses:**

Major comments:
- The claim that PLASMA is "the first deep learning framework for efficient and interpretable residue-level protein substructure alignment" is overstated. Clearly there is prior work for the kind of task tackled here (EBA, pLM-BLAST). Foldseek also falls under the category of models described here. epLSAP-Align (https://doi.org/10.1093/bioinformatics/btaf309) is also very similar and should be cited given it leverages the OT formulation on structures directly. Another related method is ActSeek, which should also be cited: https://pmc.ncbi.nlm.nih.gov/articles/PMC12343037.
- Although a preprint, I would already cite Folddisco as concurrent work, as it fulfills the same nice and needs https://www.biorxiv.org/content/10.1101/2025.07.06.663357v1. If the authors can benchmark against this method it would be great, but not strictly necessary.
- More generally, I have a feeling that the writing could benefit from a tighter list of contributions in the introductions to clearly contrast the contributions of the work (mostly the machinery on top of the pLM and siamese networks as well as some other mentioned technical contributions like the losses and scores) to other works.

Most of the weaknesses are nitpicks, here are some things I would like to see improve in terms of presentation:
- I think the results of test_extra currently in Appendix G are the most relevant results since effective generalization is most relevant. Swapping this with table 1 is I think more appropriate, especially considering those results still show strong performance.
- It would be great to have a concrete percentage for the colors relative to TM-Align.
- The contrast ratio for the colors in figure 5 is pretty bad and barely readable when printed. The squares subsetting the proteins could be also bigger for the proteins.
- There is some interchangeable terminology which can be made more precise. "substructure," "local structure," "motif," "functional site," "active/binding site" are used interchangeably but they are not completely: motifs can be discontinuous, binding sites often involve side chains only, “substructure” could mean a whole domain.
- The description of the diagonal kernel could be made a bit more explicit by highlighting it surfaces continuous diagonal elements.
- "which causes troubles" needs to be reformulated, e.g. to "which causes problems".
- appendix references should be made more explicit, indicating more precisely what's in each of them.

**Questions:**

The plan assessor does a kxk identity deconvolution to detect motifs, but this assumes sequence continuity on both proteins, which would downweight cases where there are structural motifs that are non continuous. Is it possible to remove this component and obtain confidence values on the motifs without it? Can the authors show that diagonal patterns still emerge from diagonal motifs for pLMs?
- Is it possible to have ablations on the different components of the plan assessor? There are a few elements in there and it would be good to have some insights into the impact of those elements on performance.

---

> ### Author Response · Authors · 2025-11-22
> **Response to Reviewer WJJf (1/3)**
>
> We thank the reviewer for their detailed and constructive review. We address the major comments and questions below.
>
> ## Weaknesses
> ### W1. Novelty Claim and Related Work
> We agree that our original novelty wording can be made more precise. We have revised it to:
> > *This study presents PLASMA, a deep-learning-based framework for efficient and interpretable residue-level local structural alignment.*
>
> Regarding the related works, we have cited EBA, pLM-BLAST, Foldseek, and the concurrent Folddisco preprint in the initial submission. In the revised manuscript, we also added citations and discussions of epLSAP-Align and ActSeek, as suggested by the reviewer:
> > *substructure-based methods use graph-based residue embeddings (Tan et al., 2024), focus on active-site environments
> (Castillo & Ollila, 2025), or apply linear-assignment formulations (Zhang et al., 2025).*
>
> We would also like to clarify how PLASMA differs from the previous approaches:
> - **EBA / pLM-BLAST**: both perform sequence-based alignment in an embedding space using dynamic programming, whereas PLASMA performs 3D substructure alignment through a differentiable optimal-transport formulation with a** learnable** geometric cost. Also, their outputs are order-preserving hard alignment paths, while PLASMA produces a soft, geometry-aware residue–residue alignment plan.
>
> - **Foldseek**: Foldseek is an efficient structural search tool based on discrete structural codes and global/local heuristics, but it does not offer a **differentiable**, **learnable**, or **residue-level** alignment mechanism. PLASMA focuses specifically on producing interpretable residue-level correspondences through a trainable OT formulation.
>
> - **epLSAP-Align**: This work is conceptually closest because it also applies an assignment/OT formulation to protein structures. However, its alignment objective is purely geometry-driven and does not incorporate learned representations or residue-level contextual features. As a result, it may not adapt the alignment cost to sequence-divergent but functionally related motifs. PLASMA combines residue-level geometric features with a learnable cost informed by a pLM-based Siamese encoder, enabling detection of function-relevant local structural motifs that remain conserved even when proteins differ substantially in sequence or overall fold.
>
> - **ActSeek**: ActSeek focuses on retrieving annotated functional or active sites, which requires known site annotations as input. In contrast, PLASMA aligns local structural motifs in a fully data-driven manner without relying on predefined functional labels. This allows PLASMA to operate on unannotated proteins and, in principle, provides the potential to reveal previously uncharacterized functional groups from structure-based analysis.
>
>
> ### W2. Comparison with Folddisco
> We have cited Folddisco in the introduction as concurrent work. While highly interesting, Folddisco addresses a different problem formulation: it requires an **explicit query substructure** as input and performs search against a database. PLASMA instead discovers local correspondences directly **between two proteins** without requiring a predefined motif, allowing a more general pairwise alignment setting.
>
>
> ### W3. Contribution List Revision
> We incorporated the reviewer’s suggestion and tightened the contribution list in the introduction. The revised version now reads:
> > *Our work addresses these limitations through three contributions. First, we introduce a formulation of residue-level local structural alignment based on regularized optimal transport with a learnable geometric cost, which provides a principled and flexible way to define correspondence and enables efficient, fully parallel implementation. Second, this formulation enables clear and interpretable residue–residue correspondences and naturally supports partial, variable-length, and non-sequential motif alignments, resolving the difficulty of obtaining reliable local alignments. Third, PLASMA produces a normalized and interpretable similarity score through its OT-based objective, overcoming the limitations of existing approaches whose alignment matrices or similarity measures lack a consistent probabilistic meaning. Our experiments show strong generalization to low-homology structures, and the case studies demonstrate the biological interpretability and practical utility of the resulting alignments.*

---

> ### Author Response · Authors · 2025-11-22
> **Response to Reviewer WJJf (2/3)**
>
> ### W4. Results from Appendix
> Following the reviewer’s suggestion, we have moved the current `test_extra` results into the main text and revised the associated analysis. This will better highlight the generalization behavior of PLASMA. The results analysis in Section 6.2 now reads:
> > *Table 1 reports performance on test extra, which contains functional substructures from protein families not seen during training. This setting evaluates the generalizability of the alignment framework, which is essential in practice because new functional substructures are continuously discovered. Full results on seven backbone models are provided in Appendix F, and all hyperparameter and dataset details are summarized in Appendix C.2. Corresponding interpolation results on test inter are reported in Appendix E.
> Across all three substructure detection tasks and all evaluation metrics, PLASMA achieves consistent top performance, highlighting its robustness in capturing fundamental local structural similarities for novel substructures beyond the training distribution. PLASMA-PF also performs strongly and remains competitive without task-specific training. However, unlike in the interpolation setting, PLASMA-PF does not surpass the learnable PLASMA variant on test extra; this emphasizes the value of supervised examples in improving alignment accuracy for entirely new functional sub- structures. In contrast, baseline methods show large performance variation across backbone models. EBA performs reasonably well with sequence-based Ankh and ESM2 yet drops substantially with structure-based ProtSSN, especially under the extrapolation split. Foldseek and TM-ALIGN remain consistently below PLASMA across all conditions, reflecting the limited usefulness of global structural similarity for residue-level motif detection.*
>
> ### W5. Concrete Percentages Relative to TM-align
> We have added explicit numeric percentages relative to TM-align in Appendix E and F for inter and extra test sets. In addition, we color the tables with relative improvement rather than absolute performance to provide clearer and more interpretable color ranges across diverse backbone models.
>
>
> ### W6. Color Palette of Figure 5
> Thank you for pointing this out. We experimented with multiple color palettes and contrast settings, but the issue could not be resolved through color adjustment alone. The underlying reason is that the local alignment regions are extremely small relative to the overall image resolution, so the aligned residues occupy only a few pixels, making the visual contrast difficult to improve without reducing scientific fidelity.
>
> To address this, we have included higher-resolution versions of all alignment visualizations in Figure 6 of Appendix H, where the aligned regions are clearly visible and the square annotations are enlarged for readability. We will also update the final version to link these high-resolution figures for readers.
>
>
> ### W7. Terminology Consistency
> Thank you for pointing this out. We have revised the manuscript to use *local structural motif* throughout. In our context, this refers to compact three-dimensional residue arrangements that form localized geometric patterns, which may include but are not limited to annotated functional sites such as catalytic or binding pockets.
>
> We chose this unified terminology because it precisely captures the structural units PLASMA is designed to align: small, spatially coherent 3D motifs that recur across proteins and often relate to functional or interaction-relevant geometry, as opposed to entire domains, sequence-only motifs, or arbitrary local fragments. Using a single, well-defined term eliminates ambiguity and aligns the manuscript with established usage in structural bioinformatics.
>
>
> ### W8. Description of the Kernel Design
> Thank you for the suggestion. We have updated the description below Equation 6 to explicitly clarify that the 2D identity kernel emphasizes continuous diagonal segments. The revised sentence now reads:
> > *This convolution operation highlights continuous diagonal segments in $\Omega$.*
>
>
> ### W9. Typo
> We have corrected this typo as suggested.
>
> ### W10. Appendix Construction
> We now add an opening paragraph to introduce the order and content of appendix sections.

---

> > ### Author Response · Authors · 2025-11-22
> > **Response to Reviewer WJJf (3/3)**
> >
> > ## Questions
> > ### Q1. Kernel Continuity Assumption and Diagonal Emergence
> > The identity kernel introduces a mild continuity prior that downweights scattered residue matches and highlights coherent local structural segments, which are the meaningful units PLASMA is designed to align. This does not impose strict sequence contiguity: the OT plan, computed beforehand from 3D geometry, already captures discontinuous motifs, which appear as multiple short diagonal bands and remain preserved. The assessor only amplifies these patterns rather than creating them.
> >
> >
> > ### Q2. Ablation Study
> > We thank the reviewer for the suggestion. Below we provide a combined ablation across seven protein models and all three tasks. Across all backbones and tasks, the full PLASMA model outperforms all ablated variants, demonstrating that each component contributes meaningfully, and the plan assessor is not reliant on any single heuristic. In addition: (1) Removing the continuity weighting (WC) substantially degrades performance, especially for models with weaker structure-awareness (e.g., ESM2, ProtBERT). This confirms that WC is essential for suppressing scattered matches and stabilizing the alignment plan. (2) Removing the label matching loss (LML) produces a moderate but consistent drop, particularly on structure-driven models (e.g., ProtSSN), indicating that LML improves the alignment sharpness around motif regions.
> > These results are also included in the revised Appendix G.
> >
> > **ROC-AUC**
> >
> > **1. Motif**
> >
> > | Method | ProstT5 | ProtT5 | Ankh | ESM2 | ProtSSN | TM-Vec | ProtBERT |
> > |--------|---------|--------|------|------|---------|--------|----------|
> > | PLASMA | **.97±.002** | **.97±.002** | **.95±.002** | **.96±.002** | **.96±.001** | **.92±.004** | **.87±.004** |
> > | PLASMA-PF | .94±.003 | .96±.002 | **.95±.003** | .93±.004 | .91±.003 | .87±.001 | .85±.004 |
> > | PLASMA (w/o LML) | .95±.008 | .95±.006 | .93±.004 | .91±.022 | .89±.018 | .89±.033 | .85±.004 |
> > | PLASMA (w/o WC) | .91±.005 | .95±.004 | .91±.004 | .87±.019 | .84±.003 | .86±.006 | .73±.009 |
> > | PLASMA-PF (w/o WC) | .74±.004 | .87±.002 | .85±.006 | .44±.009 | .75±.009 | .84±.007 | .60±.012 |
> >
> > **2. Binding Site**
> >
> > | Method | ProstT5 | ProtT5 | Ankh | ESM2 | ProtSSN | TM-Vec | ProtBERT |
> > |--------|---------|--------|------|------|---------|--------|----------|
> > | PLASMA | **.99±.001** | **.99±.000** | **.99±.000** | **.99±.001** | **.99±.001** | .96±.003 | **.98±.001** |
> > | PLASMA-PF | **.99±.001** | **.99±.001** | **.99±.000** | .96±.003 | .97±.001 | .92±.004 | .90±.003 |
> > | PLASMA (w/o LML) | **.99±.002** | **.99±.001** | **.99±.002** | .97±.001 | **.99±.000** | **.98±.001** | **.98±.001** |
> > | PLASMA (w/o WC) | **.99±.002** | **.99±.001** | **.99±.001** | .98±.001 | .92±.008 | .97±.002 | .77±.004 |
> > | PLASMA-PF (w/o WC) | .91±.002 | .97±.001 | .97±.002 | .49±.003 | .85±.006 | .96±.003 | .67±.006 |
> >
> > **3. Active Site**
> >
> > | Method | ProstT5 | ProtT5 | Ankh | ESM2 | ProtSSN | TM-Vec | ProtBERT |
> > |--------|---------|--------|------|------|---------|--------|----------|
> > | PLASMA | **.99±.001** | **.99±.001** | **.99±.001** | **.99±.001** | **.99±.002** | **.99±.003** | **.99±.004** |
> > | PLASMA-PF | **.99±.002** | **.99±.003** | **.99±.003** | .96±.002 | .98±.002 | .98±.003 | .94±.006 |
> > | PLASMA (w/o LML) | **.99±.001** | **.99±.000** | **.99±.001** | **.99±.005** | **.99±.000** | .98±.009 | **.99±.000** |
> > | PLASMA (w/o WC) | **.99±.001** | **.99±.001** | **.99±.001** | .98±.000 | .93±.008 | **.99±.001** | .81±.033 |
> > | PLASMA-PF (w/o WC) | .95±.002 | .97±.001 | .98±.001 | .55±.008 | .87±.005 | **.99±.001** | .67±.009 |
> >
> > **LMS**
> >
> > **1. Motif**
> >
> > | Method | ProstT5 | ProtT5 | Ankh | ESM2 | ProtSSN | TM-Vec | ProtBERT |
> > |--------|---------|--------|------|------|---------|--------|----------|
> > | PLASMA | **.91±.007** | **.92±.001** | **.92±.002** | **.92±.005** | **.73±.013** | **.76±.005** | .71±.007 |
> > | PLASMA (w/o LML) | .66±.135 | .65±.142 | **.92±.012** | .77±.170 | .48±.136 | .68±.167 | **.74±.012** |
> >
> > **2. Binding Site**
> >
> > | Method | ProstT5 | ProtT5 | Ankh | ESM2 | ProtSSN | TM-Vec | ProtBERT |
> > |--------|---------|--------|------|------|---------|--------|----------|
> > | PLASMA | **.93±.002** | **.93±.003** | **.93±.004** | **.93±.003** | .85±.006 | .86±.002 | .84±.003 |
> > | PLASMA (w/o LML) | .87±.080 | .84±.110 | .79±.081 | .49±.004 | **.88±.000** | **.90±.011** | **.89±.012** |
> >
> > **3. Active Site**
> >
> > | Method | ProstT5 | ProtT5 | Ankh | ESM2 | ProtSSN | TM-Vec | ProtBERT |
> > |--------|---------|--------|------|------|---------|--------|----------|
> > | PLASMA | **.97±.004** | **.97±.004** | **.97±.003** | **.97±.004** | .89±.016 | **.93±.006** | **.89±.008** |
> > | PLASMA (w/o LML) | .89±.080 | .84±.131 | .91±.065 | .79±.187 | **.90±.000** | .79±.143 | **.89±.007** |
> >
> > **Note:** Bold values indicate best performance for each embedding method within each task-metric combination.

---

> > > ### Comment · Reviewer_WJJf · 2025-11-25
> > >
> > > My comments are addressed, my score remains unchanged.

---

> ### Author Response · Authors · 2025-11-25
>
> Thank you for the follow-up! We appreciate your positive evaluation and confirmation.

---

### Official Review · Reviewer_e2jA · 2025-10-31

**Soundness:** 3
**Presentation:** 3
**Contribution:** 3
**Rating:** 4
**Confidence:** 4

**Summary:**

PLASMA is novel protein local structural alignment algorithm that uses differentiable optimal transport method with trainable cost matrix. The method is very fast at inference and outperforms baselines on the presented examples. Unfortunately, only one (yet unpublished) benchmark has been used for evaluation, excluding more traditional choices used in the community.

**Strengths:**

- The manuscript is well written and motivated
- The SOTA is exhaustive and correct
- The baselines are well chosen
- Both interpolation and extrapolation results are presented
- The method provides interpretability by normalized similarity scores and alignment matrices
- Parameter-free version of the network is also presented and demonstrates a strong performance

**Weaknesses:**

- The individual components and ideas of the method are not novel
- The paper does not discuss nor demonstrates examples with non-sequential alignments. This can be the actual advantage of the presented algorithm. While traditional sequence alignment (like SW or NW algorithms) is order-preserving (they align sequence A on B while preserving the residue order in both A and B), the optimal transport (OT) is in general not order preserving. Please consider case of fold-switching proteins, for example.
- More traditional structural alignment benchmarks could have been also used, examples include BALIBASE 2, BALIBASE 3, HOMSTRAD, OXBENCH, SISYPHUS
- no multiple structural alignment is discussed
- Training/test splits based on InterPro families may have data leakage, as different families may still share identical local structural motifs or even domains.

**Questions:**

- please see Weaknesses
- "the sequence identity between training and test proteins is kept below 50%."  such a split may not be sufficient to exclude data leakage. Anyway, the authors provide performance data for TM-score similarities below 0.5.
- can you also show TM-align and FoldSeek results at different levels of TM-score similarity as additional baselines in Fig. 3B?
- the classical Sinkhorn algorithm has the complexity of O(N^2 log N) for a fixed tolerance. Can you please support your O(N^2) claim, provide more details on the chosen tolerance and provide additional numerical experiments?

---

> ### Author Response · Authors · 2025-11-22
> **Response to Reviewer e2jA (1/4)**
>
> Thanks for the detailed and thoughtful review. We appreciate the positive assessment of our method, baselines, and presentation. Below we address every concern raised in the review.
>
> ## Weaknesses
> ### W1. Novelty of the Method Components
> Instead of introducing new individual components (e.g., new PLMs, Siamese encoders, OT solvers), the main contribution is a new formulation and alignment paradigm for *local protein structural alignment*—an important yet under-explored problem in structural and computational biology. As the reviewer also noted in the summary, this paradigm itself is novel. Concretely:
> - **Formulation-level novelty.** PLASMA is, to our knowledge, the first to formulate local 3D structural alignment as a differentiable optimal transport problem with a learnable cost function. This choice is essential: OT naturally provides a soft correspondence matrix, and the learnable cost lets the model directly encode residue-level geometric similarity from data rather than relying on fixed heuristics (as in TM-align, Foldseek, etc.) that cannot represent these fine-grained geometric relationships.
> - **Task-level novelty.** Classical structural alignment tools are built for global structure alignment (full-chain or full-domain comparison) and are not designed to yield clean, residue-level local motif alignments or interpretable soft correspondence scores. Our OT formulation with residue-level geometric features, in contrast, directly outputs a residuewise structural alignment matrix that captures local motifs in a fine-grained, interpretable manner. This is a key feature that prior global-alignment tools fundamentally cannot deliver.
> - **Independence from the backbone.** The strong performance of our parameter-free OT-only variant further shows that the core contribution lies in the *alignment formulation*, not in specific components of neural networks. That is, the gain comes from how PLASMA reformulates the alignment problem instead of from the choice of a specific encoder architecture.
>
> We have revised the manuscript to emphasize more clearly that our contribution lies mainly in formulation-level and task-level, instead of architecture-level. At the end of Introduction, we have
> > *...we introduce a formulation of residue-level local structural alignment based on regularized optimal transport with a learnable geometric cost, which provides a principled and flexible way to define correspondence.*
>
>
> ### W2. Non-Sequential Alignments
> Thank you for pointing this out. We agree that supporting **non-order-preserving alignment** is an important advantage of our PLASMA.
>
> - **What PLASMA supports.** PLASMA naturally allows **non-monotonic residue correspondences**, because OT imposes no sequential constraints. As a result, the model can align structurally similar local motifs even when they occur in different sequence orders, and this behavior is directly reflected in our residue–residue alignment matrices.
>
> - **What PLASMA does not target.** We appreciate the example of fold-switching proteins. However, fold switching addresses a different setting: *one sequence adopting multiple distinct global folds*. PLASMA instead aligns *functionally relevant local structures across different proteins*—i.e., similar local structures shared by different proteins and dissimilar sequences. Since we do not model multiple structural states for a single sequence, fold-switching detection requires a different problem formulation and is beyond the scope of this paper.
>
> In the revised manuscript, we give a more explicit statement to this feature. In the last paragraph of the revised introduction, we add:
> >*...this formulation enables clear and interpretable residue–residue correspondences and naturally supports partial, variable-length, and non-sequential motif alignments.*

---

> > ### Author Response · Authors · 2025-11-22
> > **Response to Reviewer e2jA (2/4)**
> >
> > ### W3. Additional Benchmark Datasets
> > Thank you for suggesting these benchmarks. We carefully reviewed each of them. Although they are relevant at a broad level, they do not fit the problem setting we study in this paper. PLASMA focuses on **residue-level local motif alignment between two proteins with considerably low sequence similarity**, where the input is a pair of structures and the output is a residue–residue correspondence matrix plus an interpretable alignment score. This task specifically targets local 3D motifs that often play functional roles.
> >
> > We have checked the public download link provided in each paper (http://www-igbmc.u-strasbg.fr/BioInfo/BAliBASE2/index.html; http://www-bio3d-igbmc.u-strasbg.fr/balibase; http://www-cryst.bioc.cam.ac.uk/-homstrad/; http://www.compbio.dundee.ac.uk; http://sisyphus.mrc-cpe.cam.ac.uk). Unfortunately, none of them are accessible.
> >
> > Moreover, the recommended benchmarks are designed for **global** or **multiple** alignment, and their ground truth does not provide residue-level alignments with function annotations:
> > - **BaliBASE2/3**: Input is a group of sequences, and the output is a **global sequence MSA**. There is no local structural ground truth; the alignment is sequence-driven and does not evaluate local 3D similarity.
> > - **HOMSTRAD**: Input is a homologous family of proteins with structures; the output is a full-length structure-based MSA, reflecting **global fold similarity** within the family, not residue-wise local motif alignment.
> > - **OXBENCH**: Input is full-chain structures of protein pairs or groups; the output is a **global structural alignment** (full-chain superposition and RMSD). The benchmark evaluates global fold correspondence, not local residue-level alignment.
> > - **SISYPHUS**: Input is a curated set of protein domains; the output is a domain-level structural alignment. Since the input is domain-level, the target remains **global topology alignment**, not motif-level correspondence.
> >
> > These datasets therefore cannot evaluate the behavior PLASMA is designed for, as none of them are provide **residue-level, function-relevant local motif ground truth**, and they are not actively maintained anymore. We believe our current benchmark explicitly supports such residue-wise supervision and evaluates structural similarity under low sequence identity, which aligns with our task formulation.
> >
> > We appreciate the reviewer’s suggestion, which motivates future extensions such as group-wise substructure alignment beyond protein pairs.
> > ### W4. Extension to Multiple Structural Alignment
> >
> > Thank you for raising this point. We agree that extending PLASMA to multiple structural alignment is an exciting direction, and one we are actively interested in exploring. However, we emphasize that the **pairwise formulation presented in this paper is a necessary and foundational first step**.
> >
> > Multiple structural alignment methods fundamentally rely on a robust and well-defined pairwise alignment operator. PLASMA provides exactly this: a differentiable, residue-level local motif alignment framework that produces interpretable alignment matrices and scores. Establishing this core formulation is essential before building a consistent MSA procedure on top of it.
> >
> > In this sense, the current work is not only self-contained and meaningful, but also lays the conceptual and algorithmic groundwork for future extensions toward group-wise or multiple structural alignment.

---

> ### Author Response · Authors · 2025-11-22
> **Response to Reviewer e2jA (3/4)**
>
> ### W5. Potential Leakage in InterPro-Based Splits
> Our goal is to prevent the type of leakage that would invalidate evaluation, i.e., **global fold or evolutionary overlap** between training and test proteins. To ensure this, we separate proteins by InterPro families (a commonly adopted biologically grounded grouping) and further enforce <50% sequence identity. We also report performance under TM-score < 0.5, which guarantees that no global structural similarity remains.
>
> It is important to distinguish global structure redundancy  from the natural recurrence of local structural motifs across proteins. Such motifs are reused broadly in evolution, and removing all motif-level similarity would require artificially discarding biologically meaningful structural patterns. Since PLASMA is designed precisely to align these shared local motifs, their presence in the test set does not constitute leakage.
>
> We would also like to highlight a property observed in Figure 3. Baseline methods rely strongly on global fold similarity when predicting substructure similarity, whereas PLASMA remains robust even when global similarity (TM-score) is low. This behavior empirically supports that our evaluation does not benefit from global structure leakage. To further support this point, we include an additional analysis in the revised Appendix L. We examine how the sequence similarity of aligned substructures varies with PLASMA’s alignment score. Across all test cases, high alignment scores consistently occur at low sequence similarity, confirming that PLASMA’s performance is not driven by unintended global similarity.
>
> ## Questions
> ### Q1. On potential leakage with <50% sequence identity
> We agree that sequence identity <50% alone may not always be sufficient to exclude all forms of overlap. This is exactly the reason why we also report performance on a subset where **TM-score between training and test proteins is <0.5 and <0.3** (in Figure 3B), in addition to InterPro family separation and the identity threshold. This explicitly rules out global fold similarity and directly addresses the concern raised by the reviewer. Local structural motifs can still recur across families, but this is an inherent property of protein structure space and is exactly what PLASMA is designed to align, rather than a form of leakage.
>
> ### Q2. On TM-align and FoldSeek baselines at different TM-score levels
> We appreciate the suggestion. In the revised version, we have added TM-align and FoldSeek as additional baselines in Fig. 3B, stratified by TM-score thresholds (e.g., <0.3, <0.5, <0.7, <1.0). These results are also summarized in the supplementary material. Across all TM-score regimes, PLASMA maintains a consistent advantage, which further supports the conclusions of the paper. Following the reviewer’s suggestion, we report the detailed performance of different models with lower TM-score groups. We also add the new numerical results in Appendix K.
>
> **ROC-AUC Performance at Different TM-Align Thresholds**
> **1. Motif**
> | TM Threshold | PLASMA | PLASMA-PF | EBA | CosineSim | TM-align | Foldseek |
> |:---:|:---:|:---:|:---:|:---:|:---:|:---:|
> | <= 1.0 | **.96±.002** | .95±.002 | .87±.003 | .84±.004 | .78±.004 | .83±.004 |
> | <= 0.7 | **.95±.002** | .94±.002 | .84±.004 | .81±.004 | .73±.005 | .81±.004 |
> | <= 0.5 | **.93±.003** | .93±.003 | .81±.005 | .78±.005 | .66±.006 | .79±.005 |
> | <= 0.3 | **.92±.004** | .91±.004 | .74±.006 | .73±.006 | .58±.007 | .74±.006 |
>
> **2. Binding Site**
> | TM Threshold | PLASMA | PLASMA-PF | EBA | CosineSim | TM-align | Foldseek |
> |:---:|:---:|:---:|:---:|:---:|:---:|:---:|
> | <= 1.0 | **.99±.001** | .99±.001 | .97±.002 | .96±.002 | .87±.003 | .90±.003 |
> | <= 0.7 | **.99±.001** | .98±.002 | .95±.003 | .93±.003 | .76±.006 | .88±.004 |
> | <= 0.5 | **.98±.002** | .97±.003 | .93±.004 | .91±.004 | .62±.007 | .85±.006 |
> | <= 0.3 | **.97±.004** | .96±.004 | .89±.007 | .88±.007 | .45±.010 | .80±.009 |
>
> **3. Active Site**
> | TM Threshold | PLASMA | PLASMA-PF | EBA | CosineSim | TM-align | Foldseek |
> |:---:|:---:|:---:|:---:|:---:|:---:|:---:|
> | <= 1.0 | **.99±.001** | .99±.001 | .99±.001 | .97±.001 | .94±.002 | .92±.003 |
> | <= 0.7 | **.99±.001** | .98±.002 | .97±.002 | .95±.003 | .88±.005 | .91±.004 |
> | <= 0.5 | **.98±.003** | .96±.004 | .95±.004 | .92±.005 | .76±.008 | .89±.006 |
> | <= 0.3 | **.96±.007** | .90±.010 | .89±.011 | .83±.013 | .59±.016 | .84±.013 |

---

> ### Author Response · Authors · 2025-11-22
> **Response to Reviewer e2jA (4/4)**
>
> ### Q3. On the complexity of Sinkhorn and the O(N²) claim
> In our implementation, both the tolerance and the maximum number of iterations are fixed constants, and the cost-matrix computation dominates the runtime. Under this fixed-tolerance and fixed-iteration setting, the effective complexity becomes (O(N^2)). The justification regarding the O(N^2) complexity of the Sinkhorn algorithm can also be found in other papers, for instance, [1].
>
> Once again, we appreciate the reviewer’s constructive feedback and believe the clarifications and additional experiments significantly strengthen the manuscript. We are happy to engage in further discussions.
>
>
> **Reference**
> [1] Raj, V., Roy, I., Ramachandran, A., Chakrabarti, S., & De, A. (2025). Charting the design space of neural graph representations for subgraph matching. ICLR 2025.

---

### Author Response · Authors · 2025-12-03
**Summary of Responses and Revision Improvements**

We sincerely thank all reviewers for their thoughtful feedback, and all chairs for their additional effort in handling the final decision. The discussion period substantially sharpened the clarity, positioning, and technical presentation of our work. We believe that the revised submission—especially with the comprehensive updates made during rebuttal—meets the high standards of ICLR and presents contributions that are both novel and impactful for the community.

We would like to briefly summarize the outcome of the initial rebuttal period. Our submission received four reviews, **including two clear accepts** (score 8). During the discussion, three reviewers (WJJf, 4sr4, and NvT1) explicitly confirmed that their concerns were fully addressed. The remaining reviewer (e2jA) did not post an update before the score lock but also raised no additional issues. Thus, **every concern that entered the discussion received an explicit acknowledgment of resolution**.

Below we summarize each subject with reviewer attribution. The concerns raised fall into three areas: novelty/positioning, methodological clarity, and experimental validation.
- **Novelty and positioning** (raised by WJJf, NvT1, e2jA).
Reviewers requested a clearer distinction between PLASMA and prior structural-alignment or OT-based methods, as well as several additional related works. We revised the introduction to sharpen the contribution statement, added discussions of epLSAP-Align and ActSeek, and incorporated the concurrent Folddisco preprint. Both WJJf and NvT1 confirmed the issue was fully addressed; e2jA did not update but raised no further objections.
- **Methodological clarity** (raised by WJJf, 4sr4).
Concerns focused on defining local structural motifs, the role of the diagonal kernel, and the formulation of PLASMA-PF. We unified terminology, made the kernel rationale explicit (“highlighting continuous diagonal segments”), clarified the cost formulation for PLASMA-PF, and expanded the explanation of the label-match loss. All reviewers who discussed these topics confirmed their questions were fully resolved.
- **Experimental validation** (raised by WJJf, NvT1, 4sr4, e2jA).
Reviewers requested stronger evidence of generalization, additional baselines, and broader ablations. In response, we moved the extrapolation split (test_extra) into the main text, added TM-align and Foldseek results across TM-score thresholds, released high-resolution alignment visualizations, extended hyperparameter analyses ($\tau$, T, kernel size k, matcher threshold $\rho$, etc.), and incorporated full ablations on plan-assessor components and the label match loss. All reviewers who commented on these points in discussion (WJJf, 4sr4, NvT1) stated that the concerns were fully resolved; e2jA did not provide an updated post-discussion reply.

Following the reviewers’ suggestions, we also made substantial improvements to the submission and **uploaded the revised manuscript and supporting document** (with modifications highlighted in red). Section 6 now presents the test_extra results in the main manuscript, together with a clearer explanation of why extrapolation is the most meaningful evaluation setting (WJJf). The contribution paragraph has been tightened and the novelty framing improved, with clearer differentiation from EBA, Foldseek, pLM-BLAST, epLSAP-Align, and ActSeek (WJJf). The terminology throughout the manuscript has been unified under local structural motif, and a short clarification is added in the introduction to remove ambiguity between substructure-, motif-, and site-level terms (WJJf). The description of PLASMA-PF now explicitly states how its cost follows Eq. 2 with the encoder removed, resolving the previous ambiguity in the formulation (4sr4). We also made the diagonal-kernel explanation more explicit and corrected ambiguous phrasing around its role in highlighting continuous diagonal segments (WJJf). Comprehensive ablation studies on the use of weight correction and label match loss are added to App.G (WJJf, NvT1). Potential data leakage risk concerns are addressed with sequence similarity analysis on the aligned substructures in App.L and exact numerical results of the ROC-AUC performance at different TM-Align thresholds in App.K (e2jA). Finally, appendix references and cross-links between interpolation, extrapolation, and hyperparameter sections have been made more precise and easier to navigate (WJJf). **All reviewers who raised these issues subsequently confirmed that the updates fully addressed their concerns** (except e2jA).

We again thank the reviewers for their thoughtful evaluations and appreciate the additional work of the chairs, particularly under the unusual identity-leak circumstances. The feedback throughout has meaningfully strengthened our submission, and we are grateful for the time and attention devoted to our paper.

---

### Meta-Review · Area_Chair_P9Zb · 2026-01-01

**Summary:**

This paper proposes PLASMA for local protein *substructure* alignment. Basically, it formulates alignment as regulateried OT, and output an interpretable residual-residual alignment matrix.

Reviewers are generally good with the fast inferenace and the alignment matrix is useful. The main concerns rise on the comparision with existing works, breadth of the evaluation and potential leakage on data.

**Reviewer Concerns:**

Initial feedback was split.

Two reviewers were clearly positive (both scored 8). Two reviewers were around the borderline (both scored 4). So this sits on the more extreme side of the split.

The negative reviews primarily focused on over-claiming novelty, the omission or incompleteness of related work, the absence of baselines and ablations. There are also and concerns regarding evaluation settings and potential leakage.

In my view, the rebuttal/revision addressed most points:
+ soften the use of "first"
+ addded a bunch of related works on alignment
+ new ablations on WC and label match

**Reviewer Scores:**

Reviewer WJJf says it is addressed and keep unchanged at 8

Reviewer 4sr4 -- "Thank you for your response. My points are addressed." unchanged at 8

Reviewer NvT1: "Thanks for the rebuttal. My concerns are addressed. I will retain my score." In my personal view, if they are addressed pisitively, the score should be improved accordingly to 6.

Reviewer e2jA: they did not post an update after rebuttal. On the conservative side, let us say it is still 4.

so this will go to 8864 on the conservative side.

---

### Decision · Program_Chairs · 2026-01-26

Accept (Poster)